# On the Lipschitz Continuity of Set Aggregation Functions and Neural Networks for Sets

**Giannis Nikolentzos**
Department of Informatics and Telecommunications
University of Peloponnese
Tripoli, 22131, Greece
`nikolentzos@uop.gr`

**Konstantinos Skianis**
Department of Computer Science & Engineering
University of Ioannina
Ioannina, 45110, Greece
`kskianis@cs.uoi.gr`

## Abstract

The Lipschitz constant of a neural network is connected to several important properties of the network such as its robustness and generalization. It is thus useful in many settings to estimate the Lipschitz constant of a model. Prior work has focused mainly on estimating the Lipschitz constant of multi-layer perceptrons and convolutional neural networks. Here we focus on data modeled as sets or multisets of vectors and on neural networks that can handle such data. These models typically apply some permutation invariant aggregation function, such as the sum, mean or max operator, to the input multisets to produce a single vector for each input sample. In this paper, we investigate whether these aggregation functions, along with an attention-based aggregation function, are Lipschitz continuous with respect to three distance functions for unordered multisets, and we compute their Lipschitz constants. In the general case, we find that each aggregation function is Lipschitz continuous with respect to only one of the three distance functions, while the attention-based function is not Lipschitz continuous with respect to any of them. Then, we build on these results to derive upper bounds on the Lipschitz constant of neural networks that can process multisets of vectors, while we also study their stability to perturbations and generalization under distribution shifts. To empirically verify our theoretical analysis, we conduct a series of experiments on datasets from different domains.

## 1 Introduction

In the past decade, deep neural networks have been applied with great success to several problems in different machine learning domains ranging from computer vision (Krizhevsky et al., 2012; He et al., 2016) to natural language processing (Vaswani et al., 2017; Peters et al., 2018). Owing to their recent success, these models are now ubiquitous in machine learning applications. However, deep neural networks can be very sensitive to their input. Indeed, it is well-known that if some specially designed small perturbation is applied to an image, it can cause a neural network model to make a false prediction even though the perturbed image looks "normal" to humans (Szegedy et al., 2014; Goodfellow et al., 2015).

A key metric for quantifying the robustness of neural networks to small perturbations is the Lipschitz constant. Training neural networks with bounded Lipschitz constant has been considered a promising direction for producing models robust to adversarial examples (Tsuzuku et al., 2018; Anil et al., 2019; Trockman & Kolter, 2021). However, if Lipschitz constraints are imposed, neural networks might lose a significant portion of their expressive power (Zhang et al., 2022a). Therefore, typically no constraints are imposed, and the neural network's Lipschitz constant is determined once the model is trained. Unfortunately, even for two-layer neural networks, exact computation of this quantity is NP-hard (Virmaux & Scaman, 2018). A recent line of work has thus focused on estimating the Lipschitz constant of neural networks, mainly by deriving upper bounds (Virmaux & Scaman, 2018; Fazlyab et al., 2019; Latorre et al., 2020; Combettes & Pesquet, 2020; Kim et al., 2021; Pabbaraju et al., 2021; Chuang & Jegelka, 2022). Efficiency is often sacrificed for the sake of tighter bounds (e. g., use of semidefinite programming) which underlines the need for accurate estimation of the Lipschitz constant.

Table 1: Summary of main results. Lipschitz constants of the different aggregation functions with respect to the three considered distance functions. $d$ denotes the dimension of the vectors. The "–" symbol denotes that the function is not Lipschitz continuous with respect to a given distance function. †: all multisets have equal cardinalities ($= M$).

|  | SUM | MEAN | MAX |
|---|---|---|---|
| EMD | $^\dagger L = M$ | $L = 1$ | $^\dagger L = M$ |
| HAUSDORFF DIST. | – | – | $L = \sqrt{d}$ |
| MATCHING DIST. | $L = 1$ | $^\dagger L = 1/M$ | $^\dagger L = 1$ |

Prior work estimates mainly the Lipschitz constant of architectures composed of fully-connected and convolutional layers. However, in several domains, input data might correspond to complex objects which consist of other simpler objects. We typically model these complex objects as sets or multisets (i.e., a generalization of a set). For instance, in computer vision, a point cloud is a set of data points in the 3-dimensional space. Likewise, in natural language processing, documents may be represented by multisets of word embeddings. Neural networks for sets typically consist of a series of fully-connected layers followed by an aggregation function which produces a representation for the entire multiset (Zaheer et al., 2017; Qi et al., 2017a). For the model to be invariant to permutations of the multiset's elements, a permutation invariant aggregation function needs to be employed. Common functions include the sum, mean and max operators. While previous work has studied the expressive power of neural networks that employ the aforementioned aggregation functions (Wagstaff et al., 2019; 2022), their Lipschitz continuity and stability to perturbations remains underexplored. Besides those standard aggregation functions, other methods for embedding multisets have been proposed recently such as the Fourier Sliced-Wasserstein embedding which is bi-Lipschitz with respect to the Wasserstein distance (Amir & Dym, 2025).

In this paper, we consider three distance functions between multisets of vectors and investigate whether the three commonly-employed aggregation functions (i.e., sum, mean, max), along with an attention-based function, are Lipschitz continuous with respect those functions. We show that for multisets of arbitrary size, each aggregation function is Lipschitz continuous with respect to only a single distance function, while the attention-based function is not Lipschitz continuous with respect to any of them. On the other hand, if all multisets have equal cardinalities, each aggregation function is Lipschitz continuous with respect to other distance functions as well. Our results are summarized in Table 1. We also study the Lipschitz constant of neural networks for sets which employ the aforementioned aggregation functions. We find that for multisets of arbitrary size, the models that employ the mean and max operators are Lipschitz continuous with respect to a single metric and we provide upper bounds on their Lipschitz constants. Strikingly, we also find that there exist models that employ the sum operator which are not Lipschitz continuous with respect to any of the considered functions. We also relate the Lipschitz constant of those networks to their generalization performance under distribution shifts. We verify our theoretical results empirically on real-world datasets from different domains.

## 2 PRELIMINARIES

### 2.1 NOTATION

Let $\mathbb{N}$ denote the set of natural numbers. Then, $[n] = \{1, \ldots, n\} \subset \mathbb{N}$ for $n \geq 1$. Let also $\{\!\{\}\!\}$ denote a *multiset*, i.e., a generalized concept of a set that allows multiple instances for its elements. Since a set is also a multiset, in what follows we use the term "multiset" to refer to both sets and multisets. Here we focus on finite multisets whose elements are $d$-dimensional real vectors. Let $M \in \mathbb{N} \setminus \{1\}$. We denote by $\mathcal{S}_{\leq M}(\mathbb{R}^d)$ and by $\mathcal{S}_M(\mathbb{R}^d)$ the set of all those multisets that consist of at most $M$ and of exactly $M$ elements, respectively. We drop the subscript when it is clear from context. The elements of a multiset do not have an inherent ordering. Therefore, the two multisets $X = \{\!\{\mathbf{v}_1, \mathbf{v}_2, \mathbf{v}_2\}\!\}$ and $Y = \{\!\{\mathbf{v}_2, \mathbf{v}_1, \mathbf{v}_2\}\!\}$ are equal to each other, i.e., $X = Y$. The cardinality $|X|$ of a multiset $X$ is equal to the number of elements of $X$. Vectors are denoted by boldface lowercase letters (e.g., $\mathbf{v}$ and $\mathbf{u}$) and matrices by boldface uppercase letters (e.g., $\mathbf{A}$ and $\mathbf{M}$). Given some vector $\mathbf{v}$, we denote by $[\mathbf{v}]_i$ the $i$-th element of the vector. Likewise, given some matrix $\mathbf{M}$, we denote by $[\mathbf{M}]_{ij}$ the element in the $i$-th row and $j$-th column of the matrix.

## 2.2 LIPSCHITZ CONTINUOUS FUNCTIONS

**Definition 2.1.** *Given two metric spaces $(\mathcal{X}, d_{\mathcal{X}})$ and $(\mathcal{Y}, d_{\mathcal{Y}})$, a function $f \colon \mathcal{X} \to \mathcal{Y}$ is called Lipschitz continuous if there exists a real constant $L \geq 0$ such that, for all $x_1, x_2 \in \mathcal{X}$, we have that*

$$d_{\mathcal{Y}}\big(f(x_1), f(x_2)\big) \leq L\, d_{\mathcal{X}}(x_1, x_2)$$

*The smallest such $L$ is called the Lipschitz constant of $f$.*

In this paper, we focus on functions $f \colon \mathcal{S}(\mathbb{R}^d) \to \mathbb{R}^{d'}$ that map multisets of $d$-dimensional vectors to $d'$-dimensional vectors. Therefore, $\mathcal{X} = \mathcal{S}(\mathbb{R}^d)$, while $\mathcal{Y} = \mathbb{R}^{d'}$. For $d_{\mathcal{X}}$, we consider three distance functions for multisets of vectors (presented in subsection 2.4 below), while $d_{\mathcal{Y}}$ is induced by the $\ell_2$-norm, i.e., $d_{\mathcal{Y}}\big(f(x_1), f(x_2)\big) = \|f(x_1) - f(x_2)\|_2$.

## 2.3 AGGREGATION FUNCTIONS

As already discussed, we consider three permutation invariant aggregation functions which are commonly employed in deep learning architectures, namely the SUM, MEAN and MAX operators.

| SUM | MEAN | MAX |
|---|---|---|
| $f_{\text{SUM}}(X) = \sum_{\mathbf{v} \in X} \mathbf{v}$ | $f_{\text{MEAN}}(X) = \dfrac{1}{\|X\|} \sum_{\mathbf{v} \in X} \mathbf{v}$ | $\big[f_{\text{MAX}}(X)\big]_i = \max\big(\{[\mathbf{v}]_i : \mathbf{v} \in X\}\big), \quad \forall i \in [d]$ |

The SUM aggregator can represent a strictly larger class of functions over sets than the MEAN and MAX aggregators. If the elements of the input sets come from a countable set $\mathcal{X}$, then for an appropriate $f \colon \mathcal{X} \to \mathbb{R}$, the function defined as $g(\{x_1, \ldots, x_n\}) = \sum_{i=1}^{n} f(x_i)$ maps the input sets injectively to $\mathbb{R}$ (Zaheer et al., 2017). Notably, it is also shown that injectivity is sufficient for approximation. On the other hand, the MEAN and MAX functions are not injective set functions. These results have been also generalized to multisets (Xu et al., 2019). Note, however, that it has been empirically observed that MEAN and MAX aggregators can outperform the SUM aggregator in certain applications (Zaheer et al., 2017; Cappart et al., 2023).

## 2.4 DISTANCE FUNCTIONS FOR UNORDERED MULTISETS

We next present the three considered functions for comparing multisets to each other. Let $X = \{\!\!\{\mathbf{v}_1, \ldots, \mathbf{v}_m\}\!\!\}$ and $Y = \{\!\!\{\mathbf{u}_1, \ldots, \mathbf{u}_n\}\!\!\}$ denote two multisets of vectors, i.e., $X, Y \in \mathcal{S}(\mathbb{R}^d)$. The three functions require to compute the distance between each element of the first multiset and every element of the second multiset. We use the distance induced by the $\ell_2$-norm (i.e., Euclidean distance) to that end. Note also that all three functions can be computed in polynomial time in the number of elements of the input multisets.

**Earth Mover's Distance.** The *earth mover's distance* (EMD) is a measure of dissimilarity between two distributions (Rubner et al., 2000). Roughly speaking, given two distributions, the output of EMD is proportional to the minimum amount of work required to change one distribution into the other. Over probability distributions, EMD is also known as the Wasserstein metric with $p = 1$ ($\mathcal{W}_1$). We use the formulation of the EMD where the total weights of the signatures are equal to each other which is known to be a metric on the space of sets of vectors (Rubner et al., 2000) and a pseudometric on $\mathcal{S}(\mathbb{R}^d)$:

$$d_{\text{EMD}}(X, Y) = \min_{\mathbf{F}} \sum_{i=1}^{m} \sum_{j=1}^{n} [\mathbf{F}]_{ij} \, \|\mathbf{v}_i - \mathbf{u}_j\|_2$$

$$\text{subject to} \quad [\mathbf{F}]_{ij} \geq 0, \qquad 1 \leq i \leq m, \quad 1 \leq j \leq n$$

$$\sum_{j=1}^{n} [\mathbf{F}]_{ij} = \frac{1}{m}, \qquad 1 \leq i \leq m$$

$$\sum_{i=1}^{m} [\mathbf{F}]_{ij} = \frac{1}{n}, \qquad 1 \leq j \leq n$$

**Hausdorff distance.** The *Hausdorff distance* is another measure of dissimilarity between two multisets of vectors (Rockafellar & Wets, 1998). It represents the maximum distance of a multiset to the nearest point in the other multiset, and is defined as follows:

$$h(X,Y) = \max_{i \in [m]} \min_{j \in [n]} \|\mathbf{v}_i - \mathbf{u}_j\|_2$$

The above distance function is not symmetric and thus it is not a metric. The bidirectional Hausdorff distance between $X$ and $Y$ is then defined as:

$$d_H(X,Y) = \max\big(h(X,Y), h(Y,X)\big)$$

The bidirectional Hausdorff distance is a metric on the space of sets of vectors and a pseudometric on $\mathcal{S}(\mathbb{R}^d)$. Roughly speaking, its value is small if every point of either set is close to some point of the other set.

**Matching Distance.** We also define a distance function for multisets of vectors, so-called *matching distance*, where elements of one multiset are assigned to elements of the other. If one of the multisets is larger than the other, some elements of the former are left unassigned. The assignments are determined by a permutation of the elements of the larger multiset. Let $\mathfrak{S}_n$ denote the set of all permutations of a multiset with $n$ elements. The matching distance between $X$ and $Y$ is defined as:

$$d_M(X,Y) = \begin{cases} M(X,Y) & \text{if } m \geq n \\ M(Y,X) & \text{otherwise.} \end{cases}$$

$$\text{where} \qquad M(X,Y) = \min_{\pi \in \mathfrak{S}_m} \left[ \sum_{i=1}^{n} \|\mathbf{v}_{\pi(i)} - \mathbf{u}_i\|_2 + \sum_{i=n+1}^{m} \|\mathbf{v}_{\pi(i)}\|_2 \right]$$

and $M(Y,X)$ is defined analogously. Variants of this distance function have been introduced in prior work (Chuang & Jegelka, 2022; Davidson & Dym, 2024). If the elements of the input multisets do not contain the zero vector, the matching distance is a metric.

**Proposition 2.2** (Proof in Appendix B.1). *The matching distance is a metric on $\mathcal{S}(\mathbb{R}^d \setminus \{\mathbf{0}\})$ where $d \in \mathbb{N}$ and $\mathbf{0}$ is the zero vector. It is a pseudometric on $\mathcal{S}(\mathbb{R}^d)$.*

For multisets of the same size, the matching distance is related to EMD.

**Proposition 2.3** (Proof in Appendix B.2). *Let $X, Y \in \mathcal{S}(\mathbb{R}^d)$ denote two multisets of the same size, i.e., $|X| = |Y| = M$. Then, we have that $d_M(X,Y) = M d_{EMD}(X,Y)$.*

## 3 LIPSCHITZ CONTINUITY OF SET AGGREGATION FUNCTIONS AND NEURAL NETWORKS

### 3.1 LIPSCHITZ CONTINUITY OF AGGREGATION FUNCTIONS

We first investigate whether the three aggregation functions which are key components in several neural network architectures are Lipschitz continuous with respect to the three considered distance functions for unordered multisets.

**Theorem 3.1** (Proof in Appendix B.3).

1. *The MEAN function defined on $\mathcal{S}_{\leq M}(\mathbb{R}^d)$ is Lipschitz continuous with respect to EMD and its Lipschitz constant is $L = 1$, but is not Lipschitz continuous with respect to the Hausdorff distance and with respect to the matching distance.*

2. *The SUM function defined on $\mathcal{S}_{\leq M}(\mathbb{R}^d)$ is Lipschitz continuous with respect to the matching distance and its Lipschitz constant is $L = 1$, but is not Lipschitz continuous with respect to EMD and with respect to the Hausdorff distance.*

3. *The MAX function defined on $\mathcal{S}_{\leq M}(\mathbb{R}^d)$ is Lipschitz continuous with respect to the Hausdorff distance and its Lipschitz constant is $L = \sqrt{d}$, but is not Lipschitz continuous with respect to EMD and with respect to the matching distance.*

The above theoretical result suggests that there is some correspondence between the three aggregation functions and the three distance functions for unordered multisets. In fact, each aggregation function seems to be closely related to a single metric. We also observe that while the Lipschitz constants of the SUM and MEAN functions with respect to the matching distance and to EMD, respectively, is constant (equal to 1), the Lipschitz constant of the MAX function with respect to the Hausdorff distance depends on the dimension $d$ of the vectors (which is typically larger than 1). A direct consequence of the above Theorem is that no two of the considered distance functions are bi-Lipschitz equivalent.

As discussed above, the SUM function is theoretically more expressive than the rest of the functions. However, in practice it has been observed that in certain tasks MEAN and MAX aggregators lead to higher levels of performance (Zaheer et al., 2017; Cappart et al., 2023). While this discrepancy between theory and empirical evidence may be attributed to the training procedure rather than to expressivity, it suggests that there is no single aggregation function that provides a superior performance under all possible circumstances and motivates the study of all three of them.

Our previous result showed that each aggregation function is Lipschitz continuous only with respect to a single distance function for multisets. It turns out that if the multisets have fixed size, then the aggregation functions are Lipschitz continuous also with respect to other functions. This is not surprising given Proposition 2.3.

**Lemma 3.2** (Proof in Appendix B.4).

1. *The MEAN function defined on $\mathcal{S}_M(\mathbb{R}^d)$ is Lipschitz continuous with respect to the matching distance and its Lipschitz constant is $L = \frac{1}{M}$, but is not Lipschitz continuous with respect to the Hausdorff distance.*

2. *The SUM function defined on $\mathcal{S}_M(\mathbb{R}^d)$ is Lipschitz continuous with respect to EMD and its Lipschitz constant is $L = M$, but is not Lipschitz continuous with respect to the Hausdorff distance.*

3. *The MAX function defined on $\mathcal{S}_M(\mathbb{R}^d)$ is Lipschitz continuous with respect to EMD and its Lipschitz constant is $L = M$, and it is also Lipschitz continuous with respect to the matching distance and its Lipschitz constant is $L = 1$.*

According to the above Lemma, the MAX function is Lipschitz continuous with respect to all three distance functions when all multisets have the same cardinality. This suggests that in such a setting, if any of the three distance functions is insensitive to some perturbation applied to the input data, this perturbation will not result in a large variation in the output of the MAX function.

In addition to the standard aggregation functions described above, recent work has also explored neural-based approaches for aggregating multisets of vectors. Here, we consider an *attention mechanism*, which is commonly employed to produce a vector from a multiset of vectors, and has achieved significant success in the fields of natural language processing (Yang et al., 2016; Nikolentzos et al., 2020) and graph learning (Veličković et al., 2018; Brody et al., 2022). Given a multiset $X = \{\!\{\mathbf{v}_1, \ldots, \mathbf{v}_m\}\!\} \in \mathcal{S}(\mathbb{R}^d)$, the attention mechanism is defined as follows:

$$f_{\text{ATT}}(X) = \sum_{i=1}^{m} \alpha_i \mathbf{v}_i \quad \text{where} \quad \alpha_i = \frac{\exp\left(\mathbf{q}^\top g(\mathbf{W}\,\mathbf{v}_i)\right)}{\sum_{j=1}^{m} \exp\left(\mathbf{q}^\top g(\mathbf{W}\,\mathbf{v}_j)\right)}$$

where $\mathbf{W} \in \mathbb{R}^{d' \times d}$ and $\mathbf{q} \in \mathbb{R}^{d'}$ denote a trainable matrix and a trainable vector, respectively, while $g$ denotes some activation function. The output of the mechanism is a convex combination of the multiset's elements.

We next investigate whether the attention mechanism $f_{\text{ATT}}(X)$ is Lipschitz continuous with respect to the three considered distance functions for multisets of vectors.

**Proposition 3.3** (Proof in Appendix B.5). *There exist instances of $f_{\text{ATT}}(X)$ defined on $\mathcal{S}_M(\mathbb{R}^d)$ which are not Lipschitz continuous with respect to any of the three considered distance functions.*

Our result aligns with the finding of Kim et al. (2021) who showed that the standard self-attention mechanism is not Lipschitz. Note, however, that the definition of the considered attention mechanism differs from that of self-attention. Kim et al. (2021) also proposed an alternative $\ell_2$ self-attention that is Lipschitz. Incorporating $\ell_2$ attention into the definition of $f_{\text{ATT}}(X)$, unfortunately, does not make it Lipschitz (more details in Appendix B.6).

## 3.2 Upper Bounds of Lipschitz Constants of Neural Networks for Sets

Neural networks that are designed for multisets typically consist of a series of fully-connected layers (i. e., a multi-layer perceptron (MLP)) followed by an aggregation function which is then potentially followed by further fully-connected layers. Exact computation of the Lipschitz constant of MLPs is NP-hard (Virmaux & Scaman, 2018). However, as already discussed, there exist several approximation algorithms which can compute tight upper bounds for MLPs (Virmaux & Scaman, 2018; Fazlyab et al., 2019; Combettes & Pesquet, 2020). An MLP that consists of $K$ layers is actually a function $f_{\mathrm{MLP}} \colon \mathbb{R}^d \to \mathbb{R}^{d'}$ defined as: $f_{\mathrm{MLP}}(\mathbf{v}) = T_K \circ \rho_{K-1} \circ \ldots \circ \rho_1 \circ T_1(\mathbf{v})$ where $T_i \colon \mathbf{v} \mapsto \mathbf{W}_i \mathbf{v} + \mathbf{b}_i$ is an affine function and $\rho_i$ is a non-linear activation function for all $i \in [K]$. Let $\mathrm{Lip}(f_{\mathrm{MLP}})$ denote the Lipschitz constant of the MLP. Note that the Lipschitz constant depends on the choice of norm, and here we assume $\ell_2$-norms for the domain and codomain of $f_{\mathrm{MLP}}$.

Given a multiset of vectors $X = \{\!\{\mathbf{v}_1, \ldots, \mathbf{v}_m\}\!\}$, let $\mathrm{NN}_g$ denote a neural network model which computes its output as $\mathrm{NN}_g(X) = f_{\mathrm{MLP}_2}\Big(g\big(\{\!\{f_{\mathrm{MLP}_1}(\mathbf{v}_1), \ldots, f_{\mathrm{MLP}_1}(\mathbf{v}_m)\}\!\}\big)\Big)$ where $g$ denotes the employed aggregation function (i. e., MEAN, SUM or MAX).

Here we investigate whether $\mathrm{NN}_g$ is Lipschitz continuous with respect to the three considered distance functions for multisets of vectors. In fact, the next Theorem utilizes the Lipschitz constants for MEAN and MAX from Theorem 3.1 to upper bound the Lipschitz constants of those neural networks.

**Theorem 3.4** (Proof in Appendix B.7)**.**

1. $\mathrm{NN}_{\mathrm{MEAN}}$ *defined on* $\mathcal{S}_{\leq M}(\mathbb{R}^d)$ *is Lipschitz continuous with respect to EMD and its Lipschitz constant is upper bounded by* $\mathrm{Lip}(f_{MLP_2}) \cdot \mathrm{Lip}(f_{MLP_1})$.

2. *There exist instances of* $\mathrm{NN}_{\mathrm{SUM}}$ *defined on* $\mathcal{S}_{\leq M}(\mathbb{R}^d)$ *which are not Lipschitz continuous with respect to the matching distance.*

3. $\mathrm{NN}_{\mathrm{MAX}}$ *defined on* $\mathcal{S}_{\leq M}(\mathbb{R}^d)$ *is Lipschitz continuous with respect to the Hausdorff distance and its Lipschitz constant is upper bounded by* $\sqrt{d} \cdot \mathrm{Lip}(f_{MLP_2}) \cdot \mathrm{Lip}(f_{MLP_1})$.

The above result suggests that if the Lipschitz constants of the MLPs are small, then the Lipschitz constant of the $\mathrm{NN}_{\mathrm{MEAN}}$ and $\mathrm{NN}_{\mathrm{MAX}}$ models with respect to EMD and Hausdorff distance, respectively, will also be small. Therefore, if proper weights are learned (or a method is employed that restricts the Lipschitz constant of the MLPs), we can obtain models stable under perturbations of the input multisets with respect to EMD or Hausdorff distance. On the other hand, $\mathrm{NN}_{\mathrm{SUM}}$ is not necessarily Lipschitz continuous with respect to the matching distance. This is due to the *bias parameters* of $f_{\mathrm{MLP}_1}$. Interestingly, if we omit the bias terms of that layer, $\mathrm{NN}_{\mathrm{SUM}}$ also becomes Lipschitz continuous with respect to the matching distance.

If the input multisets have fixed size, then we can derive upper bounds for the Lipschitz constant of $\mathrm{NN}_{\mathrm{SUM}}$, but also of $\mathrm{NN}_{\mathrm{MEAN}}$ and $\mathrm{NN}_{\mathrm{MAX}}$ with respect to other metrics.

**Lemma 3.5** (Proof in Appendix B.8)**.**

1. $\mathrm{NN}_{\mathrm{MEAN}}$ *defined on* $\mathcal{S}_M(\mathbb{R}^d)$ *is Lipschitz continuous with respect to the matching distance and its Lipschitz constant is upper bounded by* $\frac{1}{M} \cdot \mathrm{Lip}(f_{MLP_2}) \cdot \mathrm{Lip}(f_{MLP_1})$.

2. $\mathrm{NN}_{\mathrm{SUM}}$ *defined on* $\mathcal{S}_M(\mathbb{R}^d)$ *is Lipschitz continuous with respect to the matching distance and its Lipschitz constant is upper bounded by* $\mathrm{Lip}(f_{MLP_2}) \cdot \mathrm{Lip}(f_{MLP_1})$, *and is also Lipschitz continuous with respect to EMD and its Lipschitz constant is upper bounded by* $M \cdot \mathrm{Lip}(f_{MLP_2}) \cdot \mathrm{Lip}(f_{MLP_1})$.

3. $\mathrm{NN}_{\mathrm{MAX}}$ *defined on* $\mathcal{S}_M(\mathbb{R}^d)$ *is Lipschitz continuous with respect to EMD and its Lipschitz constant is upper bounded by* $M \cdot \mathrm{Lip}(f_{MLP_2}) \cdot \mathrm{Lip}(f_{MLP_1})$, *and it is also Lipschitz continuous with respect to the matching distance and its Lipschitz constant is upper bounded by* $\mathrm{Lip}(f_{MLP_2}) \cdot \mathrm{Lip}(f_{MLP_1})$.

## 3.3 Stability of Neural Networks for Sets under Perturbations

The Lipschitz constant is a well-established tool for assessing the stability of neural networks to small perturbations. Due to space limitations, we only present a single perturbation, namely element addition. Other types of perturbations (e. g., element disruption) are provided in Appendix C.

Theorem 3.4 implies that the output variation of $\text{NN}_{\text{MEAN}}$ and $\text{NN}_{\text{MAX}}$ under perturbations of the elements of an input multiset can be bounded via the EMD and Hausdorff distance between the input and perturbed multisets, respectively. It can be combined with the following result to determine the robustness of $\text{NN}_{\text{MEAN}}$ and $\text{NN}_{\text{MAX}}$ to the addition of a single element to a multiset.

**Proposition 3.6** (Proof in Appendix B.9). *Given a multiset of vectors $X = \{\!\{\mathbf{v}_1, \ldots, \mathbf{v}_n\}\!\} \in \mathcal{S}_{\leq M}(\mathbb{R}^d)$, let $X' = \{\!\{\mathbf{v}_1, \ldots, \mathbf{v}_n, \mathbf{v}_{n+1}\}\!\} \in \mathcal{S}_{\leq M}(\mathbb{R}^d)$ be the multiset where element $\mathbf{v}_{n+1}$ has been added to $X$, where $n + 1 \leq M$. Then,*

1. *The EMD between $X$ and $X'$ is bounded as $d_{EMD}(X, X') \leq \frac{1}{n(n+1)} \sum_{i=1}^{n} \|\mathbf{v}_i - \mathbf{v}_{n+1}\|$*

2. *The Hausdorff distance between $X$ and $X'$ is equal to $d_H(X, X') = \min_{i \in [n]} \|\mathbf{v}_i - \mathbf{v}_{n+1}\|$*

### 3.4 GENERALIZATION OF NEURAL NETWORKS FOR SETS UNDER DISTRIBUTION SHIFTS

Finally, we capitalize on a prior result (Shen et al., 2018), and bound the generalization error of neural networks for sets under distribution shifts. Let $\mathcal{X}$ denote the set of input data and $\mathcal{Y}$ the output space. Here we focus on binary classification tasks, i.e., $\mathcal{Y} = \{0, 1\}$. Let $\mu_S$ and $\mu_T$ denote the distribution of *source* and *target* instances, respectively. In domain adaptation, a single labeling function $f \colon \mathcal{X} \to [0, 1]$ is associated with both the source and target domains. A hypothesis class $\mathcal{H}$ is a set of predictor functions, i.e., $\forall h \in \mathcal{H}$, $h \colon \mathcal{X} \to \mathcal{Y}$. To estimate the adaptability of a hypothesis $h$, i.e., its generalization to the target distribution, the objective is to bound the target error (a.k.a. risk) $\epsilon_T(h) = \mathbb{E}_{x \sim \mu_T}[|h(x) - f(x)|]$ with respect to the source error $\epsilon_S(h) = \mathbb{E}_{x \sim \mu_S}[|h(x) - f(x)|]$ (Ben-David et al., 2010). Shen et al. (2018) show that if the hypothesis class is Lipschitz continuous, then the target error can be bounded by the Wasserstein distance with $p = 1$ for empirical measures on the source and target domain samples.

**Theorem 3.7** ((Shen et al., 2018)). *For all hypotheses $h \in \mathcal{H}$, the target error is bounded as:*

$$\epsilon_T(h) \leq \epsilon_S(h) + 2 L \mathcal{W}_1(\mu_S, \mu_T) + \lambda$$

*where $L$ is the Lipschitz constant of $h$ and $\lambda$ is the combined error of the ideal hypothesis $h^*$ that minimizes the combined error $\epsilon_S(h) + \epsilon_T(h)$.*

This bound can be applied to neural networks for sets that are Lipschitz continuous with respect to a given metric. Since $\text{NN}_{\text{MEAN}}$ and $\text{NN}_{\text{MAX}}$ are Lipschitz continuous for arbitrary multisets, EMD and Hausdorff distance can serve as ground metrics for these models. Specifically, for the two aforementioned models, the domain discrepancy $\mathcal{W}_1(\mu_S, \mu_T)$ is defined as $\mathcal{W}_1(\mu_S, \mu_T) = \inf_{\pi \in \Pi(\mu_S, \mu_T)} \int d(X, Y) d\pi(X, Y)$ where $d(X, Y)$ is EMD or the Hausdorff distance, respectively.

## 4 NUMERICAL EXPERIMENTS

We experiment with two datasets from different domains: (i) *ModelNet40*: it contains 12,311 3D CAD models that belong to 40 object categories (Wu et al., 2015); and (ii) *Polarity*: it contains 10,662 positive and negative labeled movie review snippets from Rotten Tomatoes (Pang & Lee, 2004). Note that the samples of both datasets can be thought of as multisets of vectors. Each sample of ModelNet40 is a multiset of 3-dimensional vectors. Polarity consists of textual documents, and each document is represented as a multiset of word vectors. The word vectors are obtained from a publicly available pre-trained model (Mikolov et al., 2013).

### 4.1 LIPSCHITZ CONSTANT OF AGGREGATION FUNCTIONS

In the first set of experiments, we empirically validate the results of Theorem 3.1 and Lemma 3.2. To obtain a collection of multisets of vectors, we train three different neural network models on the ModelNet40 and Polarity datasets. The difference between the three models lies in the employed aggregation function: MEAN, SUM or MAX. More details about the different layers of those models are given in Appendix D. Note that the multisets used to verify the Lipschitz constants of the aggregation functions could, in principle, be generated by any means. We use those models to create multisets since the objective is to investigate how these functions behave in comparison to the derived bounds when the inputs are sampled from real distributions, rather than artificially generated data that do not occur in practice. Once the models are trained, we feed the test samples to them.

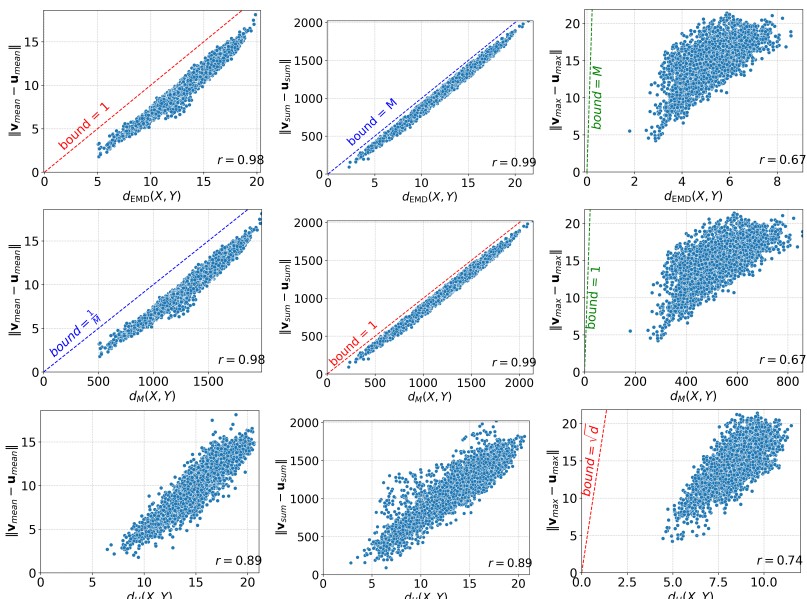

Figure 1: Each dot corresponds to a pair of point clouds from the test set of ModelNet40. Each subfigure compares the distance between the latent representations of pairs of point clouds computed by a distance function for multisets (i. e., EMD, Hausdorff distance or matching distance) with the Euclidean distance between the representations of the pairs obtained after applying an aggregation function (i. e., MEAN, SUM or MAX). The correlation between the two distances is also computed and visualized. The Lipschitz bounds are illustrated with dashed lines.

For each test sample, we store the multiset of vectors produced by the layer of the model that precedes the aggregation function, and we also store the output of the aggregation function (a vector for each multiset). We then randomly choose 100 test samples, and for each pair of those samples, we compute the EMD, Hausdorff distance and matching distance of their multisets of vectors, and also the Euclidean distance of their vector representations produced by the aggregation function. This gives rise to 9 combinations of distance functions and aggregation functions in total. Due to limited available space, we only show results for ModelNet40 in Figure 1. The results for Polarity can be found in Appendix E.1. Note that there are $\binom{100}{2} = 4,950$ distinct pairs in total. Therefore, 4,950 dots are visualized in each subfigure. To quantify the relationship between the output of the distance functions for multisets and the Euclidean distances of their vector representations, we compute and report the Pearson correlation coefficient. We observe from Figure 1 that the Lipschitz bounds (dash lines) successfully upper bound the Euclidean distance of the outputs of the aggregation functions. Note that all point clouds contained in the ModelNet40 dataset have equal cardinalities. Therefore, the conclusions of both Theorem 3.1 and Lemma 3.2 apply to this case, and thus we can derive Lipschitz constants for 7 out of the 9 combinations of distance functions for multisets and aggregation functions. We can see that the bounds that are associated with the MEAN and SUM functions are tight, while those associated with the MAX function are relatively loose. We also observe that the distances of the representations produced by the MEAN and SUM functions are very correlated with the distances produced by all three considered distance functions for multisets. On the other hand, the MAX function gives rise to representations that are less correlated with the produced distances.

## 4.2 UPPER BOUNDS OF LIPSCHITZ CONSTANTS OF NEURAL NETWORKS FOR SETS

In the second set of experiments, we empirically validate the results of Theorem 3.4 and Lemma 3.5 on the ModelNet40 and Polarity datasets. We build neural networks that consist of three layers: (i) a fully-connected layer; (ii) an aggregation function; and (iii) a second fully-connected layer. Therefore, those models first transform the elements of the input multisets using an affine function, then aggregate the representations of the elements of each multiset and finally they transform the aggregated representations using a another affine function. Note that the Lipschitz constant of an affine function is equal to the largest singular value of the associated weight matrix, and can be

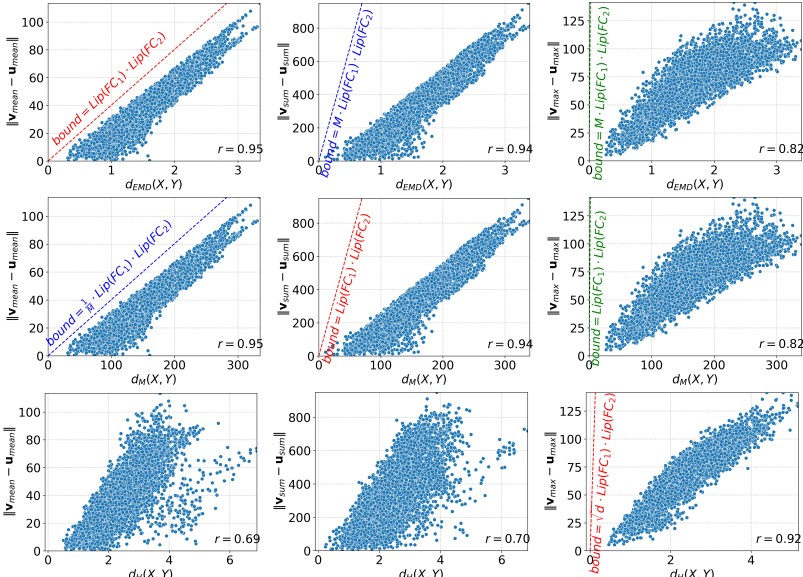

Figure 2: Each dot corresponds to a pair of point clouds from the test set of ModelNet40. Each subfigure compares the distance between the pairs of point clouds computed by EMD, Hausdorff distance or matching distance with the Euclidean distance between the representations of the pairs that emerge at the second-to-last layer of $NN_{MEAN}$, $NN_{SUM}$ or $NN_{MAX}$.

exactly computed in polynomial time. We thus denote by $Lip(FC_1)$ and $Lip(FC_2)$ the Lipschitz constants of the two fully-connected layers, respectively. Note also that the Lipschitz constant of most activation functions (e. g., ReLU, LeakyReLU, Tanh) is equal to 1. Therefore, we can compute an upper bound of the Lipschitz constant of some models using Theorem 3.4 and Lemma 3.5. To train the models, we add a final layer to them which transforms the vector representations of the multisets into class probabilities. We use the same experimental protocol as in subsection 4.1 above (i. e., we randomly choose 100 test samples). We only provide results for ModelNet40 in Figure 2, while the results for Polarity can be found in Appendix E.2. Since all point clouds contained in the ModelNet40 dataset have equal cardinalities, the conclusions of both Theorem 3.4 and Lemma 3.5 apply to this setting, and thus, once again, we can derive upper bounds on the Lipschitz constants for 7 out of the 9 combinations of distance functions for multisets and aggregation functions. We observe that the dash lines (Lipschitz upper bounds from Theorem 3.4 and Lemma 3.5) indeed upper bound the Euclidean distance of the outputs of the aggregation functions. We can also see that the bounds that are associated with the MEAN function are tight, while those associated with the SUM and MAX functions are relatively loose and very loose, respectively. We also observe that the distances of the representations produced by the MEAN and SUM functions are very correlated with the distances produced by EMD and matching distance. On the other hand, the MAX function gives rise to representations whose distances are less correlated with the distances produced by the distance functions for multisets.

## 4.3 STABILITY UNDER PERTURBATIONS OF INPUT MULTISETS

We now empirically study the stability of the two Lipschitz continuous models ($NN_{MEAN}$ and $NN_{MAX}$) under perturbations of the input multisets. Our objective is to apply small perturbations to test samples such that the models misclassify the perturbed samples. We consider two different perturbations, Pert. #1 and Pert. #2. Both perturbations are applied to test samples once each model has been trained. We then examine whether the perturbation leads to a decrease in the accuracy achieved on the test set. Pert. #1 is the perturbation described in Proposition 3.6 and is applied to the multisets of ModelNet40. Specifically, we add to each test sample a single element. We choose to add the element that has the highest norm across the elements of all samples. Pert. #2 is applied to the multisets of Polarity. It adds random noise to each element of each multiset of the test set. Specifically, a random vector is sampled from $\mathcal{U}(0, 0.2)^d$

Table 2: Average drop in accuracy of $NN_{MEAN}$ and $NN_{MAX}$ after perturbations Pert. #1 and Pert. #2 are applied to the multisets of the test set.

| Model | ModelNet40 Pert. #1 | Polarity Pert. #2 |
|---|---|---|
| $NN_{MEAN}$ | 2.0 ($\pm$ 1.3) | 13.6 ($\pm$ 7.1) |
| $NN_{MAX}$ | 20.1 ($\pm$ 1.8) | 4.8 ($\pm$ 3.7) |

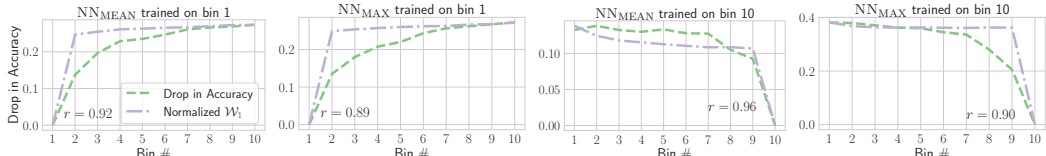

Figure 3: Size Generalization of $NN_{MEAN}$ and $NN_{MAX}$ models. For illustration purposes, the Wasserstein distances $\mathcal{W}_1$ are normalized to make the maximal distance equal to the greatest performance drops. The models in the left plots are trained on the first bucket, while those in the right plots are trained on the last bucket.

and is added to each element of each test sample. We selected this distribution because its mean and standard deviation closely match the empirical mean and standard deviation of the dataset's word vectors. The results are provided in Table 2. $NN_{MEAN}$ appears to be insensitive to Pert. #1, while $NN_{MAX}$ is insensitive to Pert. #2. The results indicate that $NN_{MEAN}$ is more robust than $NN_{MAX}$ to a larger perturbation that is associated with a single or a few elements of the multiset. On the other hand, $NN_{MAX}$ is more robust to smaller perturbations applied to all elements of the multiset.

### 4.4 Generalization under Distribution Shifts

Finally, we investigate whether neural networks for multisets can generalize to multisets of different cardinalities. We randomly sample $2,000$ documents from the Polarity dataset, and we represent them as multisets of word vectors. We then sort the multisets of word vectors based on their cardinality, and construct 10 bins, each containing 200 multisets. The $i$-th bin contains multisets $X_{(200 \cdot i)+1}, X_{(200 \cdot i)+2}, \ldots, X_{(200 \cdot i)+200}$ from the sorted list of multisets. We then train $NN_{MEAN}$ and $NN_{MAX}$ (which are Lipschitz continuous) on the first and the last bin and once the models are trained, we compute their accuracy on all 10 bins. We also compute the Wasserstein distance with $p = 1$ between domain distributions (i. e., between the first bin and the rest of the bins, and also between the last bin and the rest of the bins). We then aim to validate Theorem 3.7 which states that the error on different domains can be bounded by the Wasserstein distance between the data distributions. We thus compute the correlation between the accuracy drop and the Wasserstein distance between the two distributions. The results for $NN_{MEAN}$ and $NN_{MAX}$ are illustrated in Figure 3. The results are averaged over 10 runs. We observe that the Wasserstein distance between the data distributions using EMD (for $NN_{MEAN}$) and Hausdorff distance (for $NN_{MAX}$) as ground metrics highly correlates with the accuracy drop both in the case where the $NN_{MEAN}$ and $NN_{MAX}$ models are trained on small multisets and tested on larger multisets ($r = 0.92$ and $r = 0.90$, respectively) and also in the case where the models are trained on large multisets and tested on smaller multisets ($r = 0.94$ and $r = 0.90$, respectively). The correlation is slightly weaker in the case of the $NN_{MAX}$ model. Our results suggest that the drop in accuracy is indeed related to the Wasserstein distance between the data distributions, and that it can provide insights into the generalization performance of the models.

## 5 Conclusion and Discussion

In this paper, we studied the Lipschitz continuity of multiset aggregation functions with respect to three distance functions. We also explored the Lipschitz constants of neural networks that process multisets of vectors. As a general guideline, one should choose the aggregation function that is Lipschitz continuous with respect to the distance function that best captures the distances between the multisets in the considered dataset or problem. For example, in problems where the shape of the input object matters (e. g., shapes extracted from medical images or 3D scans), Hausdorff distance is preferable to EMD and the matching distance since we would like to detect whether any part of one shape is far away from the other shape, even if the rest of the shapes are well-aligned. However, in some cases, not a single distance function is suitable for a single problem. For instance, consider the problem of text categorization, where documents are represented as multisets of word vectors. If two documents are considered similar when they contain similar terms, regardless of their length, the EMD is likely to best capture the distance between them. On the other hand, if similarity is determined by the presence of just one or a few extreme shared words, the Hausdorff distance is more appropriate. This illustrates that selecting an aggregation function typically requires some domain knowledge. In the absence of such knowledge, choosing an aggregation function can be challenging, except in special cases, such as when multisets have the same cardinality where our results indicate that the max function is Lipschitz continuous with respect to all distance functions.

ACKNOWLEDGEMENTS

We thank the anonymous reviewers for their helpful comments and suggestions. This work has been partially supported by project MIS 5154714 of the National Recovery and Resilience Plan Greece 2.0 funded by the European Union under the NextGenerationEU Program.

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

## A    RELATED WORK

In recent years, there has been an increasing interest in applying machine learning algorithms to set-structured data. Since sets and multisets are inherently unordered, these models need to be invariant to permutations of the elements in the input set. It has been shown that, otherwise, the ordering of inputs strongly affects performance (Vinyals et al., 2016).

The seminal work of Zaheer et al. (2017) introduced DeepSets, a model that uses the sum aggregator to produce permutation-invariant set representations. They showed that when the elements of the input sets come from a countable set $\mathcal{X}$, then for an appropriate $f \colon \mathcal{X} \to \mathbb{R}$, the function defined as $g(\{x_1, \ldots, x_n\}) = \sum_{i=1}^{n} f(x_i)$ maps the input sets injectively to $\mathbb{R}$. This result was later extended to multisets (Xu et al., 2019). In the countable case, an embedding dimension of 1 already suffices for injectivity. When $\mathcal{X} = \mathbb{R}$, multiset cardinalities are bounded by $m$ and $f$ is continuous, an embedding dimension of at least $m$ is both necessary and sufficient for injectivity (Wagstaff et al., 2019; 2022). For $\mathcal{X} = \mathbb{R}^d$, an embedding dimension of at least $md$ is necessary (Joshi et al., 2023). Most of these results rely on polynomial constructions to build injective multiset functions, after which the universal approximation theorem is invoked to argue that MLPs can approximate such polynomials. Amir et al. (2023) investigate whether MLP-based multiset functions are actually injective. They show that injectivity depends on the activation function. Specifically, analytic non-polynomial activation functions yield injective models, while networks with piecewise linear activation functions are injective only when $\mathcal{X}$ is finite or corresponds to certain irregular, countably infinite sets.

Besides DeepSets, several other architectures and aggregation functions have been proposed recently. PointNet is another important architecture, primarily designed for point cloud data (Qi et al., 2017a). It consists of the same components as DeepSets, but instead of a sum aggregation function, it employs a max aggregator. To allow PointNet to capture local structures at different scales, Qi et al. (2017b) proposed PointNet++, a hierarchical model which applies PointNet recursively to nested partitions of the input set. Janossy pooling applies a neural network to all permutations of the input data and averages their outputs (Murphy et al., 2019). Since computing all permutations is generally intractable, the authors also propose some practical approximations. Set Transformer is a variant of the Transformer architecture designed for sets (Lee et al., 2019). Due to its attention mechanism, the model can capture interactions between elements in the input set. RepSet is another model designed for set-structured data which generates set representations by comparing the input set against learnable latent sets using a network flow algorithm (Skianis et al., 2020). FSPool sorts each feature across the elements of the set, and then computes a weighted sum of the elements where different weights can be learned for each feature dimension (Zhang et al., 2020). Pellegrini et al. (2021) propose a learnable aggregation function which can approximate common aggregation

functions (e. g., mean, sum, max), but also more complex functions. Kimura et al. (2024) introduce the Hölder's Power DeepSets, a model tha generalized DeepSets by employing function known as power-mean (a.k.a. Hölder mean), controlled by an exponent $p$.

Recently, a line of works has studied the Lipschitz continuity of aggregation functions and has developed embeddings that are bi-Lipschitz. Amir et al. (2023) showed that although DeepSets models that use analytic non-polynomial activation functions are injective, they are not bi-Lipschitz with respect to the 2-Wasserstein distance. Davidson & Dym (2024) investigated the Hölder continuity of neural networks for sets, a relaxation of Lipschitz continuity. They relied on a probabilistic framework of Hölder stability in expectation and showed that DeepSets with ReLU activation functions have an expected lower-Hölder exponent of $3/2$, whereas smooth activation functions yield a much worse expected lower-Hölder exponent. Balan et al. (2025) presented an embedding scheme based on sorting random projections of the multiset elements. The embedding is shown to be injective and bi-Lipschitz. The Fourier Sliced–Wasserstein (FSW) embedding is another theoretically grounded method for learning representations of sets (Amir & Dym, 2025). It computes random projections of the input data, and for each projection it samples the cosine transform of the corresponding quantile function. From a theoretical standpoint, the FSW embedding has a significant advantage over most previous methods, as it is proven to be both injective and bi-Lipschitz.

Neural network models that can handle set-structured data have been applied across diverse domains, including biology (Clarke et al., 2024), chemistry (Boulougouri et al., 2024) and materials science (Zhang et al., 2022b). In some applications, domain knowledge is incorporated into those models. For instance, Lim et al. (2023) introduce neural architectures specifically designed for eigenvector-based inputs, which can be viewed as variants of DeepSets, while explicitly accounting for the symmetries inherent in eigenvectors. For a recent overview of neural network models for set-structured data, we refer the reader to the survey by Xie & Tong (2025).

## B  PROOFS

We next provide the proofs of the theoretical claims made in the main paper.

### B.1  PROOF OF PROPOSITION 2.2

We will show that the matching distance is a metric on $\mathcal{S}(\mathbb{R}^d \setminus \{\mathbf{0}\})$. Let $X, Y \in \mathcal{S}(\mathbb{R}^d \setminus \{\mathbf{0}\})$. Non-negativity and symmetry hold trivially in all cases. Furthermore, $d_M(X, X) = 0$, while the distance between two distinct points is always positive. Suppose that $m = |X| > |Y| = n$. Then, we have that:

$$d_M(X, Y) = \min_{\pi \in \mathfrak{S}_m} \left[ \sum_{i=1}^{n} \|\mathbf{v}_{\pi(i)} - \mathbf{u}_i\| + \sum_{i=n+1}^{m} \|\mathbf{v}_{\pi(i)}\| \right] \geq \sum_{i=n+1}^{m} \|\mathbf{v}_{\pi(i)}\| > 0$$

since $\|\mathbf{v}\| > 0$ for all $\mathbf{v} \in \mathbb{R}^d \setminus \{\mathbf{0}\}$. If $|X| = |Y| = m$, since the two multisets are different from each other, there exists at least one vector $\mathbf{u}_i$ with $i \in [m]$ such that $\mathbf{u}_i \in Y$, but $\mathbf{u}_i \notin X$. Let $\pi^* \in \mathfrak{S}_m$ denote a permutation associated with $d_M(X, Y)$. Then, we have that:

$$d_M(X, Y) = \|\mathbf{v}_{\pi^*(1)} - \mathbf{u}_1\| + \ldots + \|\mathbf{v}_{\pi^*(i)} - \mathbf{u}_i\| + \ldots + \|\mathbf{v}_{\pi^*(m)} - \mathbf{u}_m\| \geq \|\mathbf{v}_{\pi^*(i)} - \mathbf{u}_i\| > 0$$

Thus, we only need to prove that the triangle inequality holds. Let $Z \in \mathcal{S}(\mathbb{R}^d \setminus \{\mathbf{0}\})$. The three multisets can have different cardinalities. Let $|X| = m$, $|Y| = n$ and $|Z| = k$. Then, $X = \{\!\{\mathbf{v}_1, \mathbf{v}_2, \ldots, \mathbf{v}_m\}\!\}$, $Y = \{\!\{\mathbf{u}_1, \mathbf{u}_2, \ldots, \mathbf{u}_n\}\!\}$ and $Z = \{\!\{\mathbf{z}_1, \mathbf{z}_2, \ldots, \mathbf{z}_k\}\!\}$. There are 6 different cases. But it suffices to show that the triangle inequality holds when $|X| \geq |Y| \geq |Z|$, when $|Z| \geq |Y| \geq |X|$ and when $|X| \geq |Z| \geq |Y|$. Proofs for the rest of the cases are similar.

**Case 1**: Suppose $|X| \geq |Y| \geq |Z|$. Let $\pi_1^*$ denote the matching produced by the solution of the matching distance function $d_M(X, Z)$:

$$\pi_1^* = \arg\min_{\pi \in \mathfrak{S}_m} \left[ \sum_{i=1}^{k} \|\mathbf{v}_{\pi(i)} - \mathbf{z}_i\| + \sum_{i=k+1}^{m} \|\mathbf{v}_{\pi(i)}\| \right]$$

Likewise, let $\pi_2^*$ denote the matching produced by the solution of the matching distance function $d_M(Z, Y)$:

$$\pi_2^* = \arg\min_{\pi \in \mathfrak{S}_n} \left[ \sum_{i=1}^{k} \|\mathbf{u}_{\pi(i)} - \mathbf{z}_i\| + \sum_{i=k+1}^{n} \|\mathbf{u}_{\pi(i)}\| \right]$$

Then, we have:

$$d_M(X, Z) + d_M(Z, Y) = \sum_{i=1}^{k} \|\mathbf{v}_{\pi_1^*(i)} - \mathbf{z}_i\| + \sum_{i=k+1}^{m} \|\mathbf{v}_{\pi_1^*(i)}\| + \sum_{i=1}^{k} \|\mathbf{z}_i - \mathbf{u}_{\pi_2^*(i)}\| + \sum_{i=k+1}^{n} \|\mathbf{u}_{\pi_2^*(i)}\|$$

$$= \sum_{i=1}^{k} \left[ \|\mathbf{v}_{\pi_1^*(i)} - \mathbf{z}_i\| + \|\mathbf{z}_i - \mathbf{u}_{\pi_2^*(i)}\| \right] + \sum_{i=k+1}^{n} \left[ \|\mathbf{v}_{\pi_1^*(i)}\| + \| - \mathbf{u}_{\pi_2^*(i)}\| \right] + \sum_{i=n+1}^{m} \|\mathbf{v}_{\pi_1^*(i)}\|$$

$$\geq \sum_{i=1}^{k} \|\mathbf{v}_{\pi_1^*(i)} - \mathbf{u}_{\pi_2^*(i)}\| + \sum_{i=k+1}^{n} \|\mathbf{v}_{\pi_1^*(i)} - \mathbf{u}_{\pi_2^*(i)}\| + \sum_{i=n+1}^{m} \|\mathbf{v}_{\pi_1^*(i)}\|$$

$$= \sum_{i=1}^{n} \|\mathbf{v}_{\pi_1^*(i)} - \mathbf{u}_{\pi_2^*(i)}\| + \sum_{i=n+1}^{m} \|\mathbf{v}_{\pi_1^*(i)}\|$$

$$\geq \min_{\pi \in \mathfrak{S}_m} \left[ \sum_{i=1}^{n} \|\mathbf{v}_{\pi(i)} - \mathbf{u}_i\| + \sum_{i=n+1}^{m} \|\mathbf{v}_{\pi(i)}\| \right]$$

$$= d_M(X, Y)$$

**Case 2**: Suppose $|X| \geq |Z| \geq |Y|$. Let $\pi_1^*$ denote the matching produced by the solution of the matching distance function $d_M(X, Z)$:

$$\pi_1^* = \arg\min_{\pi \in \mathfrak{S}_m} \left[ \sum_{i=1}^{k} \|\mathbf{v}_{\pi(i)} - \mathbf{z}_i\| + \sum_{i=k+1}^{m} \|\mathbf{v}_{\pi(i)}\| \right]$$

Likewise, let $\pi_2^*$ denote the matching produced by the solution of the matching distance function $d_M(Z, Y)$:

$$\pi_2^* = \arg\min_{\pi \in \mathfrak{S}_k} \left[ \sum_{i=1}^{n} \|\mathbf{z}_{\pi(i)} - \mathbf{u}_i\| + \sum_{i=n+1}^{k} \|\mathbf{z}_{\pi(i)}\| \right]$$

Then, we have:

$$d_M(X, Z) + d_M(Z, Y) = \sum_{i=1}^{k} \|\mathbf{v}_{\pi_1^*(i)} - \mathbf{z}_i\| + \sum_{i=k+1}^{m} \|\mathbf{v}_{\pi_1^*(i)}\| + \sum_{i=1}^{n} \|\mathbf{z}_{\pi_2^*(i)} - \mathbf{u}_i\| + \sum_{i=n+1}^{k} \|\mathbf{z}_{\pi_2^*(i)}\|$$

$$= \sum_{i=1}^{n} \left[ \|\mathbf{v}_{\pi_1^*(\pi_2^*(i))} - \mathbf{z}_{\pi_2^*(i)}\| + \|\mathbf{z}_{\pi_2^*(i)} - \mathbf{u}_i\| \right]$$

$$+ \sum_{i=n+1}^{k} \left[ \|\mathbf{v}_{\pi_1^*(\pi_2^*(i))} - \mathbf{z}_{\pi_2^*(i)}\| + \|\mathbf{z}_{\pi_2^*(i)}\| \right] + \sum_{i=k+1}^{m} \|\mathbf{v}_{\pi_1^*(i)}\|$$

$$\geq \sum_{i=1}^{n} \|\mathbf{v}_{\pi_1^*(\pi_2^*(i))} - \mathbf{u}_i\| + \sum_{i=n+1}^{k} \|\mathbf{v}_{\pi_1^*(\pi_2^*(i))}\| + \sum_{i=k+1}^{m} \|\mathbf{v}_{\pi_1^*(i)}\|$$

$$\geq \min_{\pi \in \mathfrak{S}_m} \left[ \sum_{i=1}^{n} \|\mathbf{v}_{\pi(i)} - \mathbf{u}_i\| + \sum_{i=n+1}^{m} \|\mathbf{v}_{\pi(i)}\| \right]$$

$$= d_M(X, Y)$$

**Case 3**: Suppose $|Z| \geq |Y| \geq |X|$. Let $\pi_1^*$ denote the matching produced by the solution of the matching distance function $d_M(X, Z)$:

$$\pi_1^* = \arg\min_{\pi \in \mathfrak{S}_k} \sum_{i=1}^{m} \left[ \|\mathbf{v}_i - \mathbf{z}_{\pi(i)}\| + \sum_{i=m+1}^{k} \|\mathbf{z}_{\pi(i)}\| \right]$$

Likewise, let $\pi_2^*$ denote the matching produced by the solution of the matching distance function $d_M(Z, Y)$:

$$\pi_2^* = \arg\min_{\pi \in \mathfrak{S}_k} \left[ \sum_{i=1}^{n} \|\mathbf{u}_i - \mathbf{z}_{\pi(i)}\| + \sum_{i=n+1}^{k} \|\mathbf{z}_{\pi(i)}\| \right]$$

Then, we have:

$$d_M(X, Z) + d_M(Z, Y) = \sum_{i=1}^{m} \|\mathbf{v}_i - \mathbf{z}_{\pi_1^*(i)}\| + \sum_{i=m+1}^{k} \|\mathbf{z}_{\pi_1^*(i)}\| + \sum_{i=1}^{n} \|\mathbf{z}_{\pi_2^*(i)} - \mathbf{u}_i\| + \sum_{i=n+1}^{k} \|\mathbf{z}_{\pi_2^*(i)}\|$$

For each $i \in [k]$, there exists a single $j \in [k]$ such that $\pi_1^*(i) = \pi_2^*(j)$. For each $i, j \in [k]$ with $\pi_1^*(i) = \pi_2^*(j)$ one of the following holds:

1. $\|\mathbf{v}_i - \mathbf{z}_{\pi_1^*(i)}\| + \|\mathbf{z}_{\pi_2^*(j)} - \mathbf{u}_j\| \geq \|\mathbf{v}_i - \mathbf{u}_j\|$ if $i \leq m$ and $j \leq n$
2. $\|\mathbf{v}_i - \mathbf{z}_{\pi_1^*(i)}\| + \|\mathbf{z}_{\pi_2^*(j)}\| \geq \|\mathbf{v}_i\|$ if $i \leq m$ and $j > n$
3. $\|\mathbf{z}_{\pi_1^*(i)}\| + \|\mathbf{z}_{\pi_2^*(j)} - \mathbf{u}_j\| \geq \|\mathbf{u}_j\|$ if $i > m$ and $j \leq n$
4. $\|\mathbf{z}_{\pi_1^*(i)}\| + \|\mathbf{z}_{\pi_2^*(j)}\| \geq 0$ if $i > m$ and $j > n$

Note that $d_M(X, Z) + d_M(Z, Y)$ can be written as a sum of $k$ terms, where each term corresponds to one of the above 4 sums of norms. If we take pairs of terms of types 2 and 3 and we sum them, we have that:

$$\|\mathbf{v}_i - \mathbf{z}_{\pi_1^*(i)}\| + \|\mathbf{z}_{\pi_2^*(j)}\| + \|\mathbf{z}_{\pi_1^*(i)}\| + \|\mathbf{z}_{\pi_2^*(j)} - \mathbf{u}_j\| \geq \|\mathbf{v}_i\| + \|\mathbf{u}_j\| = \|\mathbf{v}_i\| + \|-\mathbf{u}_j\| \geq \|\mathbf{v}_i - \mathbf{u}_j\|$$

Note also that type 2 occurs $m - n$ times more than type 3. Therefore, using the inequalities for the 4 types of sums of norms above, we have:

$$d_M(X, Z) + d_M(Z, Y) \geq \min_{\pi \in \mathfrak{S}_m} \left[ \sum_{i=1}^{n} \|\mathbf{v}_{\pi(i)} - \mathbf{u}_{\pi(i)}\| + \sum_{i=n+1}^{m} \|\mathbf{v}_{\pi(i)}\| \right]$$
$$= d_M(X, Y)$$

In case $\mathbf{0}$ can be an element of the multisets, there exist $X, Y \in \mathcal{S}(\mathbb{R}^d)$ with $X \neq Y$ such that $d_M(X, Y) = 0$, i.e., the distance between two distinct points can be equal to 0. The rest of the properties still hold, and thus the matching distance in a pseudometric on $\mathcal{S}(\mathbb{R}^d)$.

## B.2 PROOF OF PROPOSITION 2.3

If $|X| = |Y| = M$, the second and third constraints of the optimization problem that needs to be solved to compute $d_{\mathrm{EMD}}(X, Y)$ become as follows:

$$\sum_{j=1}^{M} [\mathbf{F}]_{ij} = \frac{1}{M}, \quad 1 \leq i \leq M \qquad \text{and} \qquad \sum_{i=1}^{M} [\mathbf{F}]_{ij} = \frac{1}{M}, \quad 1 \leq j \leq M$$

Therefore, matrix $\mathbf{F}$ is a doubly stochastic matrix. The Birkhoff-von Neumann Theorem states that the set of $M \times M$ doubly stochastic matrices forms a convex polytope whose vertices are the $M \times M$ permutation matrices. Furthermore, it is known that the optimal value of a linear objective in a nonempty polytope is attained at a vertex of the polytope (Bertsimas & Tsitsiklis, 1997). The optimal solution would thus be a permutation matrix $\mathbf{P} \in \mathbf{\Pi}_M$ scaled by $1/M$. Let also $\pi \in \mathfrak{S}_M$ denote the permutation that is associated with that matrix. Therefore, we have that:

$$M d_{\mathrm{EMD}}(X, Y) = M \min_{\mathbf{F} \in \mathcal{B}_M} \sum_{i=1}^{M} \sum_{j=1}^{M} [\mathbf{F}]_{ij} \|\mathbf{v}_i - \mathbf{u}_j\|_2$$
$$= M \min_{\mathbf{P} \in \mathbf{\Pi}_M} \sum_{i=1}^{M} \sum_{j=1}^{M} \frac{1}{M} [\mathbf{P}]_{ij} \|\mathbf{v}_i - \mathbf{u}_j\|_2$$
$$= \min_{\pi \in \mathfrak{S}_M} \sum_{i=1}^{M} \|\mathbf{v}_{\pi(i)} - \mathbf{u}_i\|_2$$
$$= d_M(X, Y)$$

### B.3 PROOF OF THEOREM 3.1

#### B.3.1 THE MEAN FUNCTION IS LIPSCHITZ CONTINUOUS WITH RESPECT TO EMD

Let $X = \{\!\{\mathbf{v}_1, \mathbf{v}_2, \ldots, \mathbf{v}_m\}\!\}$ and $Y = \{\!\{\mathbf{u}_1, \mathbf{u}_2, \ldots, \mathbf{u}_n\}\!\}$ be two multisets, consisting of $m$ and $n$ vectors of dimension $d$, respectively. Let also $\mathbf{F}^*$ denote the matrix that minimizes $d_{\text{EMD}}(X, Y)$. Then, we have that:

$$
\begin{aligned}
\left\| f_{\text{MEAN}}(X) - f_{\text{MEAN}}(Y) \right\| &= \left\| \frac{1}{m} \sum_{i=1}^{m} \mathbf{v}_i - \frac{1}{n} \sum_{j=1}^{n} \mathbf{u}_j \right\| \\
&= \left\| \sum_{i=1}^{m} \frac{1}{m} \mathbf{v}_i - \sum_{j=1}^{n} \frac{1}{n} \mathbf{u}_j \right\| \\
&= \left\| \sum_{i=1}^{m} \left( \sum_{j=1}^{n} [\mathbf{F}^*]_{ij} \right) \mathbf{v}_i - \sum_{j=1}^{n} \left( \sum_{i=1}^{m} [\mathbf{F}^*]_{ij} \right) \mathbf{u}_j \right\| \\
&= \left\| \sum_{i=1}^{m} \sum_{j=1}^{n} [\mathbf{F}^*]_{ij} (\mathbf{v}_i - \mathbf{u}_j) \right\| \\
&\leq \sum_{i=1}^{m} \sum_{j=1}^{n} \left\| [\mathbf{F}^*]_{ij} (\mathbf{v}_i - \mathbf{u}_j) \right\| \\
&= \sum_{i=1}^{m} \sum_{j=1}^{n} [\mathbf{F}^*]_{ij} \left\| (\mathbf{v}_i - \mathbf{u}_j) \right\| \\
&= d_{\text{EMD}}(X, Y)
\end{aligned}
$$

The MEAN function is thus Lipschitz continuous with respect to EMD and the Lipschitz constant is equal to $1$.

#### B.3.2 THE MEAN FUNCTION IS NOT LIPSCHITZ CONTINUOUS WITH RESPECT TO THE MATCHING DISTANCE

Suppose that the MEAN function is Lipschitz continuous with respect to the matching distance. Let $L > 0$ be given. Let also $\epsilon > 0$ and $c > (2L+1)\epsilon$. Let $X = \{\mathbf{v}_1, \mathbf{v}_2\}$, $Y = \{\mathbf{u}_1\}$ be two multisets, consisting of 2 and 1 vectors of dimension $d$, respectively. Then, we set $\mathbf{v}_1 = \mathbf{u}_1 = (c, c, \ldots, c)^\top$, and $\mathbf{v}_2 = (\epsilon, \epsilon, \ldots, \epsilon)^\top$. Clearly, we have that $d_M(X, Y) = \|\mathbf{v}_2\| = \sqrt{d}\epsilon$. We also have that:

$$
\begin{aligned}
\left\| f_{\text{MEAN}}(X) - f_{\text{MEAN}}(Y) \right\| &= \left\| \frac{1}{2} \sum_{i=1}^{2} \mathbf{v}_i - \mathbf{u}_1 \right\| \\
&= \left\| \frac{1}{2} \mathbf{v}_1 + \frac{1}{2} \mathbf{v}_2 - \mathbf{u}_1 \right\| \\
&= \left\| \frac{1}{2} (c, c, \ldots, c)^\top + \frac{1}{2} (\epsilon, \epsilon, \ldots, \epsilon)^\top - (c, c, \ldots, c)^\top \right\| \\
&= \left\| \left( \frac{\epsilon - c}{2}, \frac{\epsilon - c}{2}, \ldots, \frac{\epsilon - c}{2} \right)^\top \right\| \\
&= \frac{1}{2} \| (\epsilon - c, \epsilon - c, \ldots, \epsilon - c)^\top \| \\
&= \frac{1}{2} \sqrt{\underbrace{(\epsilon - c)^2 + (\epsilon - c)^2 + \ldots + (\epsilon - c)^2}_{d \text{ times}}} \\
&= \frac{1}{2} \sqrt{d}(c - \epsilon) \\
&> \frac{1}{2} \sqrt{d}\big((2L+1)\epsilon - \epsilon\big) \\
&= L\sqrt{d}\epsilon
\end{aligned}
$$

$$= L\, d_M(X, Y)$$

Therefore, for any $L > 0$, there exist $X, Y \in \mathcal{S}(\mathbb{R}^d)$ such that $\|f_{\text{MEAN}}(X) - f_{\text{MEAN}}(Y)\| > L\, d_M(X, Y)$, which is a contradiction. Thus, the MEAN function is not Lipschitz continuous with respect to the matching distance.

### B.3.3 THE MEAN FUNCTION IS NOT LIPSCHITZ CONTINUOUS WITH RESPECT TO THE HAUSDORFF DISTANCE

Suppose that the MEAN function is Lipschitz continuous with respect to the Hausdorff distance. Let $L > 0$ be given. Let also $\epsilon > 0$ and $c > 3L\epsilon$. Let $X = \{\mathbf{v}_1, \mathbf{v}_2, \mathbf{v}_3\}$, $Y = \{\mathbf{u}_1, \mathbf{u}_2\}$ be two multisets, consisting of 3 and 2 vectors of dimension $d$, respectively. Then, we set $\mathbf{v}_1 = \mathbf{u}_1 = (-c, -c, \ldots, -c)^\top$, $\mathbf{v}_2 = \mathbf{u}_2 = (c, c, \ldots, c)^\top$ and $\mathbf{v}_3 = (c+\epsilon, c+\epsilon, \ldots, c+\epsilon)^\top$. Clearly, we have that $d_H(X, Y) = \max_{i \in [3]} \min_{j \in [2]} \|\mathbf{v}_i - \mathbf{u}_j\| = \|\mathbf{v}_3 - \mathbf{u}_2\| = \sqrt{d}\epsilon$. We also have that:

$$
\begin{aligned}
\left\| f_{\text{MEAN}}(X) - f_{\text{MEAN}}(Y) \right\| &= \left\| \frac{1}{3} \sum_{i=1}^{3} \mathbf{v}_i - \frac{1}{2} \sum_{j=1}^{2} \mathbf{u}_j \right\| \\
&= \left\| \frac{1}{3}\mathbf{v}_1 + \frac{1}{3}\mathbf{v}_2 + \frac{1}{3}\mathbf{v}_3 - \frac{1}{2}\mathbf{u}_1 - \frac{1}{2}\mathbf{u}_2 \right\| \\
&= \left\| \frac{1}{3}\mathbf{v}_3 \right\| \\
&= \frac{1}{3}\sqrt{(c+\epsilon)^2 + (c+\epsilon)^2 + \ldots + (c+\epsilon)^2} \\
&= \frac{1}{3}\sqrt{d(c+\epsilon)^2} \\
&= \frac{1}{3}\sqrt{d}(c+\epsilon) \\
&> \frac{1}{3}\sqrt{d}(3L\epsilon + \epsilon) \\
&> L\sqrt{d}\epsilon \\
&= L\, d_H(X, Y)
\end{aligned}
$$

For any $L > 0$, there exist $X, Y \in \mathcal{S}(\mathbb{R}^d)$ such that $\|f_{\text{MEAN}}(X) - f_{\text{MEAN}}(Y)\| > L\, d_H(X, Y)$. We have thus reached a contradiction. Therefore, the MEAN function is not Lipschitz continuous with respect to the Hausdorff distance.

### B.3.4 THE SUM FUNCTION IS NOT LIPSCHITZ CONTINUOUS WITH RESPECT TO EMD

Suppose that the SUM function is Lipschitz continuous with respect to EMD. Let $L > 0$ be given. Then, let $m = \lfloor L + 1 \rfloor$. Let also $X = \{\!\{\mathbf{v}_1, \ldots, \mathbf{v}_m\}\!\}$, $Y = \{\!\{\mathbf{u}_1, \ldots, \mathbf{u}_m\}\!\}$ be two multisets, each consisting of $m$ vectors. We construct the two multisets such that $\mathbf{v}_1 = \mathbf{u}_1, \mathbf{v}_2 = \mathbf{u}_2, \ldots, \mathbf{v}_{m-1} = \mathbf{u}_{m-1}$, and such that $\mathbf{v}_1 + \ldots + \mathbf{v}_{m-1} = \mathbf{u}_1 + \ldots + \mathbf{u}_{m-1} = 0$. Let also $\|\mathbf{v}_m - \mathbf{u}_m\| = m$. We already showed in subsection B.3.1 that the distance between the mean vectors of two multisets of vectors is a lower bound on the EMD between them. Therefore, we have that:

$$
\begin{aligned}
\sum_{i=1}^{m} \sum_{j=1}^{m} [\mathbf{F}]_{ij} \|\mathbf{v}_i - \mathbf{u}_j\| &\geq \left\| \frac{1}{m} \sum_{i=1}^{m} \mathbf{v}_i - \frac{1}{m} \sum_{j=1}^{m} \mathbf{u}_j \right\| \\
&= \frac{1}{m} \|(\mathbf{v}_1 - \mathbf{u}_1) + (\mathbf{v}_2 - \mathbf{u}_2) + \ldots + (\mathbf{v}_m - \mathbf{u}_m)\| \\
&= \frac{1}{m} \|\mathbf{v}_m - \mathbf{u}_m\| \\
&= \frac{1}{m} m = 1
\end{aligned}
$$

We can achieve the lower bound if we set the values of $\mathbf{F}$ as follows:

$$
[\mathbf{F}^*]_{ij} = \begin{cases} \frac{1}{m} & \text{if } i = j \\ 0 & \text{if } i \neq j \end{cases}
$$

Therefore, the EMD between $X$ and $Y$ is equal to 1. Then, we have that:

$$\left\| f_{\text{SUM}}(X) - f_{\text{SUM}}(Y) \right\| = \left\| \sum_{i=1}^{m} \mathbf{v}_i - \sum_{j=1}^{m} \mathbf{u}_j \right\|$$
$$= \left\| (\mathbf{v}_1 - \mathbf{u}_1) + (\mathbf{v}_2 - \mathbf{u}_2) + \ldots + (\mathbf{v}_m - \mathbf{u}_m) \right\|$$
$$= \left\| \mathbf{v}_m - \mathbf{u}_m \right\|$$
$$= m \cdot 1$$
$$= m \sum_{i=1}^{m} \sum_{j=1}^{m} [\mathbf{F}^*]_{ij} \| \mathbf{v}_i - \mathbf{u}_j \|$$
$$> L \sum_{i=1}^{m} \sum_{j=1}^{m} [\mathbf{F}^*]_{ij} \| \mathbf{v}_i - \mathbf{u}_j \|$$
$$= L \, d_{\text{EMD}}(X, Y)$$

Therefore, for any $L > 0$, there exist $X, Y \in \mathcal{S}(\mathbb{R}^d)$ such that $\| f_{\text{SUM}}(X) - f_{\text{SUM}}(Y) \| > L \, d_{\text{EMD}}(X, Y)$. We have thus arrived at a contradiction, and the SUM function is not Lipschitz continuous with respect to EMD.

### B.3.5 THE SUM FUNCTION IS LIPSCHITZ CONTINUOUS WITH RESPECT TO THE MATCHING DISTANCE

Let $X = \{\!\{\mathbf{v}_1, \mathbf{v}_2, \ldots, \mathbf{v}_m\}\!\}$ and $Y = \{\!\{\mathbf{u}_1, \mathbf{u}_2, \ldots, \mathbf{u}_n\}\!\}$ be two multisets, consisting of $m$ and $n$ vectors of dimension $d$, respectively. Without loss of generality, we assume that $m > n$. Let $\pi^*$ denote the matching produced by the solution of the matching distance function:

$$\pi^* = \underset{\pi \in \mathfrak{S}_m}{\arg\min} \left[ \sum_{i=1}^{n} \| \mathbf{v}_{\pi(i)} - \mathbf{u}_i \| + \| \mathbf{v}_{\pi(n+1)} \| + \ldots + \| \mathbf{v}_{\pi(m)} \| \right]$$

Then, we have that:

$$\left\| f_{\text{SUM}}(X) - f_{\text{SUM}}(Y) \right\| = \left\| \sum_{i=1}^{m} \mathbf{v}_i - \sum_{j=1}^{n} \mathbf{u}_j \right\|$$
$$= \left\| (\mathbf{v}_{\pi^*(1)} - \mathbf{u}_1) + \ldots + (\mathbf{v}_{\pi^*(n)} - \mathbf{u}_n) + \mathbf{v}_{\pi^*(n+1)} + \ldots + \mathbf{v}_{\pi^*(m)} \right\|$$
$$\leq \| \mathbf{v}_{\pi^*(1)} - \mathbf{u}_1 \| + \ldots + \| \mathbf{v}_{\pi^*(n)} - \mathbf{u}_n \| + \| \mathbf{v}_{\pi^*(n+1)} \| + \ldots + \| \mathbf{v}_{\pi^*(m)} \|$$
$$= \sum_{i=1}^{n} \| \mathbf{v}_{\pi^*(i)} - \mathbf{u}_i \| + \| \mathbf{v}_{\pi^*(n+1)} \| + \ldots + \| \mathbf{v}_{\pi^*(m)} \|$$
$$= \min_{\pi \in \mathfrak{S}_m} \left[ \sum_{i=1}^{n} \| \mathbf{v}_{\pi(i)} - \mathbf{u}_i \| + \| \mathbf{v}_{\pi(n+1)} \| + \ldots + \| \mathbf{v}_{\pi(m)} \| \right]$$
$$= d_M(X, Y)$$

which concludes the proof. The SUM function is thus Lipschitz continuous with respect to the matching distance and the Lipschitz constant is equal to 1.

### B.3.6 THE SUM FUNCTION IS NOT LIPSCHITZ CONTINUOUS WITH RESPECT TO THE HAUSDORFF DISTANCE

Suppose that the SUM function is Lipschitz continuous with respect to the Hausdorff distance. Let $L > 0$ be given. Let also $\epsilon > 0$ and $c > L\epsilon$. Let $X = \{\mathbf{v}_1, \mathbf{v}_2\}$, $Y = \{\mathbf{u}_1\}$ be two multisets, consisting of 2 and 1 vectors of dimension $d$, respectively. Then, we set $\mathbf{v}_1 = \mathbf{u}_1 = (c, c, \ldots, c)^\top$ and $\mathbf{v}_2 = (c+\epsilon, c+\epsilon, \ldots, c+\epsilon)^\top$. Clearly, we have that $d_H(X, Y) = \max_{i \in [2]} \min_{j \in [1]} \| \mathbf{v}_i - \mathbf{u}_j \| = \| \mathbf{v}_2 - \mathbf{u}_1 \| = \sqrt{d}\epsilon$. We also have that:

$$\left\| f_{\text{SUM}}(X) - f_{\text{SUM}}(Y) \right\| = \left\| \sum_{i=1}^{2} \mathbf{v}_i - \mathbf{u}_1 \right\|$$

$$= \|\mathbf{v}_1 + \mathbf{v}_2 - \mathbf{u}_1\|$$
$$= \|\mathbf{v}_2\|$$
$$= \sqrt{(c + \epsilon)^2 + (c + \epsilon)^2 + \ldots + (c + \epsilon)^2}$$
$$= \sqrt{d(c + \epsilon)^2}$$
$$= \sqrt{d}(c + \epsilon)$$
$$> \sqrt{d}(L\epsilon + \epsilon)$$
$$> L\sqrt{d}\epsilon$$
$$= L\, d_H(X, Y)$$

Therefore, for any $L > 0$, there exist $X, Y \in \mathcal{S}(\mathbb{R}^d)$ such that $\|f_{\text{SUM}}(X) - f_{\text{SUM}}(Y)\| > L\, d_H(X, Y)$, which is a contradiction. Therefore, the SUM function is also not Lipschitz continuous with respect to the Hausdorff distance.

### B.3.7  THE MAX FUNCTION IS NOT LIPSCHITZ CONTINUOUS WITH RESPECT TO EMD

Suppose that the MAX function is Lipschitz continuous with respect to EMD. Let $L > 0$ be given. Then, let $m = \lfloor L + 1 \rfloor$. Let also $X = \{\!\{\mathbf{v}_1, \ldots, \mathbf{v}_m\}\!\}$, $Y = \{\!\{\mathbf{u}_1, \ldots, \mathbf{u}_m\}\!\}$ be two multisets, each consisting of $m$ $d$-dimensional vectors. We construct the two sets such that $\mathbf{v}_1 = \mathbf{u}_1, \mathbf{v}_2 = \mathbf{u}_2, \ldots, \mathbf{v}_{m-1} = \mathbf{u}_{m-1}$, and such that $\mathbf{v}_1 + \ldots + \mathbf{v}_{m-1} = \mathbf{u}_1 + \ldots + \mathbf{u}_{m-1} = 0$. Suppose that the elements of vectors $\mathbf{v}_m$, $\mathbf{u}_m$ are larger than those of all other vectors of $X$ and $Y$, respectively. Therefore, we have that $[\mathbf{v}_m]_k \geq [\mathbf{v}_i]_k$, $\forall i \in [m]$ and $k \in [d]$. We also have that $[\mathbf{u}_m]_k \geq [\mathbf{u}_j]_k$, $\forall j \in [m]$ and $k \in [d]$. Let also $\|\mathbf{v}_m - \mathbf{u}_m\| = 1$. We already showed in subsection B.3.1 that the distance between the mean vectors of two multisets of vectors is a lower bound on the EMD between them. Therefore, we have that:

$$\sum_{i=1}^{m} \sum_{j=1}^{m} [\mathbf{F}]_{ij} \|\mathbf{v}_i - \mathbf{u}_j\| \geq \left\| \frac{1}{m} \sum_{i=1}^{m} \mathbf{v}_i - \frac{1}{m} \sum_{j=1}^{m} \mathbf{u}_j \right\|$$
$$= \frac{1}{m} \|(\mathbf{v}_1 - \mathbf{u}_1) + (\mathbf{v}_2 - \mathbf{u}_2) + \ldots + (\mathbf{v}_m - \mathbf{u}_m)\|$$
$$= \frac{1}{m} \|\mathbf{v}_m - \mathbf{u}_m\|$$
$$= \frac{1}{m}$$

We can achieve the lower bound if we set the values of $\mathbf{F}$ as follows:

$$[\mathbf{F}^*]_{ij} = \begin{cases} \frac{1}{m} & \text{if } i = j \\ 0 & \text{if } i \neq j \end{cases}$$

Therefore, the EMD between $X$ and $Y$ is equal to $1/m$. Then, we have that

$$\left\| f_{\text{MAX}}(X) - f_{\text{MAX}}(Y) \right\| = \|\mathbf{v}_m - \mathbf{u}_m\|$$
$$= m \cdot \frac{1}{m}$$
$$= m \sum_{i=1}^{m} \sum_{j=1}^{m} [\mathbf{F}^*]_{ij} \|\mathbf{v}_i - \mathbf{u}_j\|$$
$$> L \sum_{i=1}^{m} \sum_{j=1}^{m} [\mathbf{F}^*]_{ij} \|\mathbf{v}_i - \mathbf{u}_j\|$$
$$= L\, d_{\text{EMD}}(X, Y)$$

Therefore, for any $L > 0$, there exist $X, Y \in \mathcal{S}(\mathbb{R}^d)$ such that $\|f_{\text{MAX}}(X) - f_{\text{MAX}}(Y)\| > L\, d_{\text{EMD}}(X, Y)$, which is a contradiction. Therefore, the MAX function is not Lipschitz continuous with respect to EMD.

### B.3.8 THE MAX FUNCTION IS NOT LIPSCHITZ CONTINUOUS WITH RESPECT TO THE MATCHING DISTANCE

Let $L > 0$ be given. Let also $\epsilon > 0$ and $c > L\epsilon$. Let $X = \{\mathbf{v}_1, \mathbf{v}_2\}, Y = \{\mathbf{u}_1\}$ be two multisets, consisting of 2 and 1 vectors of dimension $d$, respectively. Then, we set $\mathbf{v}_1 = \mathbf{u}_1 = (-c, -c, \ldots, -c)^\top$, and $\mathbf{v}_2 = (\epsilon, \epsilon, \ldots, \epsilon)^\top$. Clearly, we have that $d_M(X, Y) = \|\mathbf{v}_2\| = \sqrt{d}\epsilon$. Let also $\mathbf{v}_{\max}$ and $\mathbf{u}_{\max}$ denote the vectors that emerge after applying max pooling across all points of $X$ and $Y$, respectively. We also have that:

$$
\begin{aligned}
\left\| f_{\text{MAX}}(X) - f_{\text{MAX}}(Y) \right\| &= \|\mathbf{v}_{\max} - \mathbf{u}_{\max}\| \\
&= \|\mathbf{v}_{\max} - \mathbf{u}_1\| \\
&= \left\| \left( \max(-c, \epsilon), \max(-c, \epsilon), \ldots, \max(-c, \epsilon) \right)^\top - (-c, -c, \ldots, -c)^\top \right\| \\
&= \|(\epsilon, \epsilon, \ldots, \epsilon)^\top - (-c, -c, \ldots, -c)^\top\| \\
&= \sqrt{\underbrace{(\epsilon + c)^2 + (\epsilon + c)^2 + \ldots + (\epsilon + c)^2}_{d \text{ times}}} \\
&= \sqrt{d}(c + \epsilon) \\
&> \sqrt{d}(L\epsilon + \epsilon) \\
&> L\sqrt{d}\epsilon \\
&= L d_M(X, Y)
\end{aligned}
$$

For any $L > 0$, there exist $X, Y \in \mathcal{S}(\mathbb{R}^d)$ such that $\|f_{\text{MAX}}(X) - f_{\text{MAX}}(Y)\| > L\, d_M(X, Y)$. Based on the above inequality, the MAX function is not Lipschitz continuous with respect to the matching distance.

### B.3.9 THE MAX FUNCTION IS LIPSCHITZ CONTINUOUS WITH RESPECT TO THE HAUSDORFF DISTANCE

Let $X = \{\!\{\mathbf{v}_1, \mathbf{v}_2, \ldots, \mathbf{v}_m\}\!\}$ and $Y = \{\!\{\mathbf{u}_1, \mathbf{u}_2, \ldots, \mathbf{u}_n\}\!\}$ denote two multisets of vectors. Let also $\mathbf{v}_{\max}$ and $\mathbf{u}_{\max}$ denote the vectors that emerge after applying max pooling across all points of $X$ and $Y$, respectively. We will show that $\forall k \in [d]$, we have that $\left| [\mathbf{v}_{\max}]_k - [\mathbf{u}_{\max}]_k \right| \leq d_H(X, Y)$. By contradiction, we assume that that there is some $k \in [d]$ such that $\left| [\mathbf{v}_{\max}]_k - [\mathbf{u}_{\max}]_k \right| > d_H(X, Y)$. Without loss of generality, we also assume that $[\mathbf{v}_{\max}]_k \geq [\mathbf{u}_{\max}]_k$, and therefore $\left| [\mathbf{v}_{\max}]_k - [\mathbf{u}_{\max}]_k \right| = [\mathbf{v}_{\max}]_k - [\mathbf{u}_{\max}]_k$.

Since $[\mathbf{u}_{\max}]_k \geq [\mathbf{u}_j]_k, \forall j \in [n]$, we have that $[\mathbf{v}_{\max}]_k - [\mathbf{u}_{\max}]_k \leq [\mathbf{v}_{\max}]_k - [\mathbf{u}_j]_k, \forall j \in [n]$. Since by our assumption above, $\left| [\mathbf{v}_{\max}]_k - [\mathbf{u}_{\max}]_k \right| = [\mathbf{v}_{\max}]_k - [\mathbf{u}_{\max}]_k > d_H(X, Y)$, it follows that

$$[\mathbf{v}_{\max}]_k - [\mathbf{u}_j]_k > d_H(X, Y), \quad \forall j \in [n] \tag{1}$$

Note that there is at least one vector $\mathbf{v}_i \in X$ such that $[\mathbf{v}_i]_k = [\mathbf{v}_{\max}]_k$. From equation 1, we have for this vector that $[\mathbf{v}_i]_k - [\mathbf{u}_j]_k > d_H(X, Y), \forall j \in [n]$. Then, we have $\|\mathbf{v}_i - \mathbf{u}_j\| = \sqrt{\left([\mathbf{v}_i]_1 - [\mathbf{u}_j]_1\right)^2 + \ldots + \left([\mathbf{v}_i]_k - [\mathbf{u}_j]_k\right)^2 + \ldots + \left([\mathbf{v}_i]_d - [\mathbf{u}_j]_d\right)^2} \geq \sqrt{\left([\mathbf{v}_i]_k - [\mathbf{u}_j]_k\right)^2} = [\mathbf{v}_i]_k - [\mathbf{u}_j]_k > d_H(X, Y), \forall j \in [n]$. We thus have that $\min_{j \in [n]} \|\mathbf{v}_i - \mathbf{u}_j\| > d_H(X, Y)$ which is a contradiction since $\min_{j \in [n]} \|\mathbf{v}_i - \mathbf{u}_j\| \leq \max_{i \in [m]} \min_{j \in [n]} \|\mathbf{v}_i - \mathbf{u}_j\| = h(X, Y) \leq d_H(X, Y)$. Therefore, we have that $\left| [\mathbf{v}_{\max}]_k - [\mathbf{u}_{\max}]_k \right| \leq d_H(X, Y)$.

Since $k$ was arbitrary, the above inequality holds for all $k \in [d]$. We thus have

$$
\begin{aligned}
\left\| f_{\text{MAX}}(X) - f_{\text{MAX}}(Y) \right\| &= \|\mathbf{v}_{\max} - \mathbf{u}_{\max}\| \\
&= \sqrt{\left([\mathbf{v}_{\max}]_1 - [\mathbf{u}_{\max}]_1\right)^2 + \left([\mathbf{v}_{\max}]_2 - [\mathbf{u}_{\max}]_2\right)^2 + \ldots + \left([\mathbf{v}_{\max}]_d - [\mathbf{u}_{\max}]_d\right)^2} \\
&\leq \sqrt{\underbrace{\left(d_H(X, Y)\right)^2 + \left(d_H(X, Y)\right)^2 + \ldots + \left(d_H(X, Y)\right)^2}_{d \text{ times}}}
\end{aligned}
$$

$$= \sqrt{d\big(d_H(X,Y)\big)^2}$$
$$= \sqrt{d}\, d_H(X,Y)$$

which concludes the proof. Therefore, The MAX function is Lipschitz continuous with respect to the Hausdorff distance and the Lipschitz constant is equal to $\sqrt{d}$.

### B.4  PROOF OF LEMMA 3.2

#### B.4.1  THE MEAN FUNCTION IS LIPSCHITZ CONTINUOUS WITH RESPECT TO THE MATCHING DISTANCE

Let $\mathcal{X}$ denote a set that contains multisets of vectors of equal cardinalities, i.e., $|X| = M$ and $X \in \mathcal{S}_M(\mathbb{R}^d)$, $\forall X \in \mathcal{X}$ where $M \in \mathbb{N}$. Let $X, Y \in \mathcal{X}$ denote two multisets. By Proposition 2.3, we have that $d_M(X,Y) = M d_{\text{EMD}}(X,Y)$. By Theorem 3.1, we have that:

$$\left\| f_{\text{MEAN}}(X) - f_{\text{MEAN}}(Y) \right\| \le d_{\text{EMD}}(X,Y)$$
$$= \frac{1}{M} d_M(X,Y) \text{ (due to Proposition 2.3)}$$

The MEAN function restricted to inputs from set $\mathcal{X}$ is thus Lipschitz continuous with respect to the matching distance and the Lipschitz constant is equal to $\frac{1}{M}$.

#### B.4.2  THE MEAN FUNCTION IS NOT LIPSCHITZ CONTINUOUS WITH RESPECT TO THE HAUSDORFF DISTANCE

Let $\mathcal{X}$ denote a set that contains multisets of vectors of equal cardinalities, i.e., $|X| = M$ and $X \in \mathcal{S}_M(\mathbb{R}^d)$, $\forall X \in \mathcal{X}$ where $M \in \mathbb{N}$. Suppose that the MEAN function restricted to inputs from set $\mathcal{X}$ is Lipschitz continuous with respect to the Hausdorff distance. Let $L > 0$ be given. Let also $\epsilon > 0$ and $c > 3L\epsilon$. Let $X = \{\mathbf{v}_1, \mathbf{v}_2, \mathbf{v}_3\}$, $Y = \{\mathbf{u}_1, \mathbf{u}_2, \mathbf{u}_3\}$ be two multisets, consisting of 3 vectors of dimension $d$, respectively. Then, we set $\mathbf{v}_1 = \mathbf{u}_1 = (-\frac{c}{2}, -\frac{c}{2}, \ldots, -\frac{c}{2})^\top$, $\mathbf{v}_2 = \mathbf{u}_2 = \left(\frac{c}{2}, \frac{c}{2}, \ldots, \frac{c}{2}\right)^\top$, $\mathbf{v}_3 = \left(\frac{c+\epsilon}{2}, \frac{c+\epsilon}{2}, \ldots, \frac{c+\epsilon}{2}\right)^\top$ and $\mathbf{u}_3 = \left(-\frac{c+\epsilon}{2}, -\frac{c+\epsilon}{2}, \ldots, -\frac{c+\epsilon}{2}\right)^\top$ Clearly, we have that $d_H(X,Y) = \max_{i\in[3]} \min_{j\in[3]} \|\mathbf{v}_i - \mathbf{u}_j\| = \max_{j\in[3]} \min_{i\in[3]} \|\mathbf{v}_i - \mathbf{u}_j\| = \|\mathbf{v}_3 - \mathbf{u}_2\| = \|\mathbf{v}_1 - \mathbf{u}_3\| = \frac{\sqrt{d}\epsilon}{2}$. We also have that:

$$\left\| f_{\text{MEAN}}(X) - f_{\text{MEAN}}(Y) \right\| = \left\| \frac{1}{3} \sum_{i=1}^{3} \mathbf{v}_i - \frac{1}{3} \sum_{i=1}^{3} \mathbf{u}_1 \right\|$$
$$= \left\| \frac{1}{3}(\mathbf{v}_1 - \mathbf{u}_1 + \mathbf{v}_2 - \mathbf{u}_2 + \mathbf{v}_3 - \mathbf{u}_3) \right\|$$
$$= \frac{1}{3} \|\mathbf{v}_3 - \mathbf{u}_3\|$$
$$= \frac{1}{3} \left\| \left(\frac{c+\epsilon}{2}, \frac{c+\epsilon}{2}, \ldots, \frac{c+\epsilon}{2}\right)^\top - \left(-\frac{c+\epsilon}{2}, -\frac{c+\epsilon}{2}, \ldots, -\frac{c+\epsilon}{2}\right)^\top \right\|$$
$$= \frac{1}{3} \|(c+\epsilon, c+\epsilon, \ldots, c+\epsilon)^\top\|$$
$$= \frac{1}{3} \sqrt{(c+\epsilon)^2 + (c+\epsilon)^2 + \ldots + (c+\epsilon)^2}$$
$$= \frac{1}{3} \sqrt{d(c+\epsilon)^2}$$
$$= \frac{1}{3} \sqrt{d}(c+\epsilon)$$
$$> \frac{1}{3} \sqrt{d}(3L\epsilon + \epsilon)$$
$$> L\sqrt{d}\epsilon$$
$$> L\frac{\sqrt{d}\epsilon}{2}$$

$$= L\, d_H(X, Y)$$

Therefore, for any $L > 0$, there exist $X, Y \in \mathcal{X}$ such that $\|f_{\text{MEAN}}(X) - f_{\text{MEAN}}(Y)\| > L\, d_H(X, Y)$, which is a contradiction. Therefore, the MEAN function is not Lipschitz continuous with respect to the Hausdorff distance even when it is restricted to inputs from set $\mathcal{X}$.

### B.4.3 THE SUM FUNCTION IS LIPSCHITZ CONTINUOUS WITH RESPECT TO EMD

Let $\mathcal{X}$ denote a set that contains multisets of vectors of equal cardinalities, i.e., $|X| = M$ and $X \in \mathcal{S}_M(\mathbb{R}^d)$, $\forall X \in \mathcal{X}$ where $M \in \mathbb{N}$. Let $X, Y \in \mathcal{X}$ denote two multisets. By Proposition 2.3, we have that $d_M(X, Y) = M d_{\text{EMD}}(X, Y)$. By Theorem 3.1, we have that:

$$\left\| f_{\text{SUM}}(X) - f_{\text{SUM}}(Y) \right\| \leq d_M(X, Y)$$
$$= M d_{\text{EMD}}(X, Y) \text{ (due to Proposition 2.3)}$$

Therefore, the SUM function restricted to inputs from set $\mathcal{X}$ is Lipschitz continuous with respect to EMD and the Lipschitz constant is equal to $M$.

### B.4.4 THE SUM FUNCTION IS NOT LIPSCHITZ CONTINUOUS WITH RESPECT TO THE HAUSDORFF DISTANCE

Let $\mathcal{X}$ denote a set that contains multisets of vectors of equal cardinalities, i.e., $|X| = M$ and $X \in \mathcal{S}_M(\mathbb{R}^d)$, $\forall X \in \mathcal{X}$ where $M \in \mathbb{N}$. Suppose that the SUM function restricted to inputs from set $\mathcal{X}$ is Lipschitz continuous with respect to the Hausdorff distance. Let $L > 0$ be given. Let also $\epsilon > 0$ and $c > L\epsilon$. Let $X = \{\mathbf{v}_1, \mathbf{v}_2, \mathbf{v}_3\}$, $Y = \{\mathbf{u}_1, \mathbf{u}_2, \mathbf{u}_3\}$ be two multisets, consisting of 3 vectors of dimension $d$, respectively. Then, we set $\mathbf{v}_1 = \mathbf{u}_1 = \left(-\frac{c}{2}, -\frac{c}{2}, \ldots, -\frac{c}{2}\right)^\top$, $\mathbf{v}_2 = \mathbf{u}_2 = \left(\frac{c}{2}, \frac{c}{2}, \ldots, \frac{c}{2}\right)^\top$, $\mathbf{v}_3 = \left(\frac{c+\epsilon}{2}, \frac{c+\epsilon}{2}, \ldots, \frac{c+\epsilon}{2}\right)^\top$ and $\mathbf{u}_3 = \left(-\frac{c+\epsilon}{2}, -\frac{c+\epsilon}{2}, \ldots, -\frac{c+\epsilon}{2}\right)^\top$ Clearly, we have that $d_H(X, Y) = \max_{i \in [3]} \min_{j \in [3]} \|\mathbf{v}_i - \mathbf{u}_j\| = \max_{j \in [3]} \min_{i \in [3]} \|\mathbf{v}_i - \mathbf{u}_j\| = \|\mathbf{v}_3 - \mathbf{u}_2\| = \|\mathbf{v}_1 - \mathbf{u}_3\| = \frac{\sqrt{d}\epsilon}{2}$. We also have that:

$$
\begin{aligned}
\left\| f_{\text{SUM}}(X) - f_{\text{SUM}}(Y) \right\| &= \left\| \sum_{i=1}^{3} \mathbf{v}_i - \sum_{i=1}^{3} \mathbf{u}_1 \right\| \\
&= \|\mathbf{v}_1 - \mathbf{u}_1 + \mathbf{v}_2 - \mathbf{u}_2 + \mathbf{v}_3 - \mathbf{u}_3\| \\
&= \|\mathbf{v}_3 - \mathbf{u}_3\| \\
&= \left\| \left(\frac{c+\epsilon}{2}, \frac{c+\epsilon}{2}, \ldots, \frac{c+\epsilon}{2}\right)^\top - \left(-\frac{c+\epsilon}{2}, -\frac{c+\epsilon}{2}, \ldots, -\frac{c+\epsilon}{2}\right)^\top \right\| \\
&= \|(c+\epsilon, c+\epsilon, \ldots, c+\epsilon)^\top\| \\
&= \sqrt{(c+\epsilon)^2 + (c+\epsilon)^2 + \ldots + (c+\epsilon)^2} \\
&= \sqrt{d(c+\epsilon)^2} \\
&= \sqrt{d}(c+\epsilon) \\
&> \sqrt{d}(L\epsilon + \epsilon) \\
&> L\frac{\sqrt{d}\epsilon}{2} \\
&= L\, d_H(X, Y)
\end{aligned}
$$

Therefore, for any $L > 0$, there exist $X, Y \in \mathcal{X}$ such that $\|f_{\text{SUM}}(X) - f_{\text{SUM}}(Y)\| > L\, d_H(X, Y)$, which is a contradiction. Therefore, the SUM function is not Lipschitz continuous with respect to the Hausdorff distance even when it is restricted to inputs from set $\mathcal{X}$.

### B.4.5 THE MAX FUNCTION IS LIPSCHITZ CONTINUOUS WITH RESPECT TO THE MATCHING DISTANCE

Let $\mathcal{X}$ denote a set that contains multisets of vectors of equal cardinalities, i.e., $|X| = M$ and $X \in \mathcal{S}_M(\mathbb{R}^d)$, $\forall X \in \mathcal{X}$ where $M \in \mathbb{N}$. Let $X = \{\!\{\mathbf{v}_1, \mathbf{v}_2, \ldots, \mathbf{v}_M\}\!\} \in \mathcal{X}$, and

$Y = \{\!\{\mathbf{u}_1, \mathbf{u}_2, \ldots, \mathbf{u}_M\}\!\} \in \mathcal{X}$ denote two multisets. Then, we have that:

$$\left\|f_{\text{MAX}}(X) - f_{\text{MAX}}(Y)\right\| = \sqrt{\left([\mathbf{v}_{\max}]_1 - [\mathbf{u}_{\max}]_1\right)^2 + \left([\mathbf{v}_{\max}]_2 - [\mathbf{u}_{\max}]_2\right)^2 + \ldots + \left([\mathbf{v}_{\max}]_d - [\mathbf{u}_{\max}]_d\right)^2} \tag{2}$$

Note that $\forall k \in [d]$, there exists at least one $\mathbf{v}_i \in X$ such that $[\mathbf{v}_{\max}]_k = [\mathbf{v}_i]_k$, and also at least one $\mathbf{u}_j \in Y$ such that $[\mathbf{u}_{\max}]_k = [\mathbf{u}_j]_k$. Furthermore, we denote by $\pi^*$ the matching produced by the solution of the matching distance function:

$$\pi^* = \arg\min_{\pi \in \mathfrak{S}_M} \sum_{i=1}^{M} \|\mathbf{v}_{\pi(i)} - \mathbf{u}_i\|$$

Then, $\forall k \in [d]$ where $[\mathbf{v}_{\max}]_k \geq [\mathbf{u}_{\max}]_k$ and $[\mathbf{v}_{\max}]_k = [\mathbf{v}_{\pi(i)}]_k$, we have that:

$$\begin{aligned}
\left([\mathbf{v}_{\max}]_k - [\mathbf{u}_{\max}]_k\right)^2 &= \left([\mathbf{v}_{\pi^*(i)}]_k - [\mathbf{u}_{\max}]_k\right)^2 \\
&= \min_{j \in [M]} \left([\mathbf{v}_{\pi^*(i)}]_k - [\mathbf{u}_j]_k\right)^2 \\
&\leq \left([\mathbf{v}_{\pi^*(i)}]_k - [\mathbf{u}_i]_k\right)^2
\end{aligned} \tag{3}$$

Likewise, $\forall k \in [d]$ where $[\mathbf{u}_{\max}]_k > [\mathbf{v}_{\max}]_k$ and $[\mathbf{u}_{\max}]_k = [\mathbf{u}_i]_k$, we have that:

$$\begin{aligned}
\left([\mathbf{v}_{\max}]_k - [\mathbf{u}_{\max}]_k\right)^2 &= \left([\mathbf{v}_{\max}]_k - [\mathbf{u}_i]_k\right)^2 \\
&= \min_{j \in [M]} \left([\mathbf{v}_j]_k - [\mathbf{u}_i]_k\right)^2 \\
&\leq \left([\mathbf{v}_{\pi^*(i)}]_k - [\mathbf{u}_i]_k\right)^2
\end{aligned} \tag{4}$$

Then, for each index pair $(\pi^*(i), i)$, let $\mathcal{S}_{\pi^*(i)} = \{j\colon [\mathbf{v}_{\max}]_j = [\mathbf{v}_{\pi^*(i)}]_j, j \in [d]\}$ and $\mathcal{S}_i = \{j\colon [\mathbf{u}_{\max}]_j = [\mathbf{u}_i]_j, j \in [d]\}$ denote the sets of coordinates at which $\mathbf{v}_{\pi^*(i)}$ and $\mathbf{u}_i$, respectively, attain the coordinate-wise maximum over their corresponding multisets. Then, from equation 3, and equation 4, we have that:

$$\sqrt{\sum_{k \in \mathcal{S}_{\pi^*(i)}} \left([\mathbf{v}_{\max}]_k - [\mathbf{u}_{\max}]_k\right)^2 + \sum_{k \in \mathcal{S}_i \setminus \mathcal{S}_{\pi^*(i)}} \left([\mathbf{v}_{\max}]_k - [\mathbf{u}_{\max}]_k\right)^2} \leq \|\mathbf{v}_{\pi^*(i)} - \mathbf{u}_i\|$$

Based on the above, we have that:

$$\begin{aligned}
\left\|f_{\text{MAX}}(X) - f_{\text{MAX}}(Y)\right\| &= \sqrt{\left([\mathbf{v}_{\max}]_1 - [\mathbf{u}_{\max}]_1\right)^2 + \left([\mathbf{v}_{\max}]_2 - [\mathbf{u}_{\max}]_2\right)^2 + \ldots + \left([\mathbf{v}_{\max}]_d - [\mathbf{u}_{\max}]_d\right)^2} \\
&\leq \|\mathbf{v}_{\pi(1)} - \mathbf{u}_1\| + \|\mathbf{v}_{\pi(2)} - \mathbf{u}_2\| + \ldots + \|\mathbf{v}_{\pi(M)} - \mathbf{u}_M\| \\
&= d_M(X, Y)
\end{aligned}$$

The MAX function restricted to inputs from set $\mathcal{X}$ is thus Lipschitz continuous with respect to the matching distance and the Lipschitz constant is equal to 1.

### B.4.6 THE MAX FUNCTION IS LIPSCHITZ CONTINUOUS WITH RESPECT TO EMD

Let $\mathcal{X}$ denote a set that contains multisets of vectors of equal cardinalities, i.e., $|X| = M$ and $X \in \mathcal{S}_M(\mathbb{R}^d)$, $\forall X \in \mathcal{X}$ where $M \in \mathbb{N}$. Let $X, Y \in \mathcal{X}$ denote two multisets. We have shown in subsection B.4.5 above that:

$$\left\|f_{\text{MAX}}(X) - f_{\text{MAX}}(Y)\right\| \leq d_M(X, Y)$$

By Proposition 2.3, we have that $d_M(X, Y) = M d_{\text{EMD}}(X, Y)$. Therefore, we have that:

$$\begin{aligned}
\left\|f_{\text{MAX}}(X) - f_{\text{MAX}}(Y)\right\| &\leq d_M(X, Y) \\
&= M\, d_{\text{EMD}}(X, Y) \text{ (due to Proposition 2.3)}
\end{aligned}$$

The MAX function restricted to inputs from set $\mathcal{X}$ is thus Lipschitz continuous with respect to EMD and the Lipschitz constant is equal to $M$.

### B.5 PROOF OF PROPOSITION 3.3

#### B.5.1 THE ATT MECHANISM IS NOT LIPSCHITZ CONTINUOUS WITH RESPECT TO EMD

Suppose that the ATT mechanism is Lipschitz continuous with respect to EMD. Let $L > 0$ be given. Let also $\epsilon > 0$ and $c > \frac{2(1+\exp(d\epsilon))L\epsilon}{\exp(d\epsilon)-1}$. Let $X = \{\mathbf{v}_1, \mathbf{v}_2\}$, $Y = \{\mathbf{u}_1, \mathbf{u}_2\}$ be two multisets, each consisting of 2 vectors of dimension $d$. Then, suppose that $\mathbf{v}_1 = \mathbf{u}_1 = (c, c, \ldots, c)^\top$, $\mathbf{v}_2 = (\epsilon, \epsilon, \ldots, \epsilon)^\top$ and $\mathbf{u}_2 = (-\epsilon, -\epsilon, \ldots, -\epsilon)^\top$. Clearly, we have that $d_{EMD}(X, Y) = \frac{1}{2}\|\mathbf{v}_2 - \mathbf{u}_2\| = \sqrt{d}\epsilon$. Let also $\mathbf{W} = -\mathbf{I}$ and $\mathbf{q} = \mathbf{1}$ where $\mathbf{I}$ denotes the $d \times d$ identity matrix and $\mathbf{1}$ denotes the $d$-dimensional vector of all ones. We define $g$ as the ReLU function. The attention coefficients of the elements of $X$ are equal to:

$$\alpha_1^X = \frac{\exp\left(\mathbf{1}^\top \text{RELU}(-\mathbf{I}\,\mathbf{v}_1)\right)}{\sum_{j=1}^2 \exp\left(\mathbf{1}^\top \text{RELU}(-\mathbf{I}\,\mathbf{v}_j)\right)} = \frac{\exp(0)}{\exp(0) + \exp(0)} = \frac{1}{2}$$

$$\alpha_2^X = \frac{\exp\left(\mathbf{1}^\top \text{RELU}(-\mathbf{I}\,\mathbf{v}_2)\right)}{\sum_{j=1}^2 \exp\left(\mathbf{1}^\top \text{RELU}(-\mathbf{I}\,\mathbf{v}_j)\right)} = \frac{\exp(0)}{\exp(0) + \exp(0)} = \frac{1}{2}$$

The attention coefficients of the elements of $Y$ are equal to:

$$\alpha_1^Y = \frac{\exp\left(\mathbf{1}^\top \text{RELU}(-\mathbf{I}\,\mathbf{u}_1)\right)}{\sum_{j=1}^2 \exp\left(\mathbf{1}^\top \text{RELU}(-\mathbf{I}\,\mathbf{u}_j)\right)} = \frac{\exp(0)}{\exp(0) + \exp(d\epsilon)} = \frac{1}{1 + \exp(d\epsilon)}$$

$$\alpha_2^Y = \frac{\exp\left(\mathbf{1}^\top \text{RELU}(-\mathbf{I}\,\mathbf{u}_2)\right)}{\sum_{j=1}^2 \exp\left(\mathbf{1}^\top \text{RELU}(-\mathbf{I}\,\mathbf{u}_j)\right)} = \frac{\exp(d\epsilon)}{\exp(0) + \exp(d\epsilon)} = \frac{\exp(d\epsilon)}{1 + \exp(d\epsilon)}$$

We then have that:

$$
\begin{aligned}
\left\| f_{\text{ATT}}(X) - f_{\text{ATT}}(Y) \right\| &= \left\| \sum_{i=1}^2 \alpha_i^X \mathbf{v}_i - \sum_{j=1}^2 \alpha_j^Y \mathbf{u}_j \right\| \\
&= \left\| \frac{1}{2}\mathbf{v}_1 + \frac{1}{2}\mathbf{v}_2 - \frac{1}{1 + \exp(d\epsilon)}\mathbf{u}_1 - \frac{\exp(d\epsilon)}{1 + \exp(d\epsilon)}\mathbf{u}_2 \right\| \\
&= \left\| \frac{\exp(d\epsilon) - 1}{2(1 + \exp(d\epsilon))}(c, c, \ldots, c)^\top + \frac{3\exp(d\epsilon) + 1}{2(1 + \exp(d\epsilon))}(\epsilon, \epsilon, \ldots, \epsilon)^\top \right\| \\
&> \left\| \frac{\exp(d\epsilon) - 1}{2(1 + \exp(d\epsilon))}(c, c, \ldots, c)^\top \right\| \\
&> \left\| \frac{\exp(d\epsilon) - 1}{2(1 + \exp(d\epsilon))}\left(\frac{2(1 + \exp(d\epsilon))L\epsilon}{\exp(d\epsilon) - 1}, \frac{2(1 + \exp(d\epsilon))L\epsilon}{\exp(d\epsilon) - 1}, \ldots, \frac{2(1 + \exp(d\epsilon))L\epsilon}{\exp(d\epsilon) - 1}\right)^\top \right\| \\
&= \left\| L(\epsilon, \epsilon, \ldots, \epsilon)^\top \right\| \\
&= L\sqrt{\underbrace{\epsilon^2 + \epsilon^2 + \ldots + \epsilon^2}_{d \text{ times}}} \\
&= L\sqrt{d}\epsilon \\
&= L\, d_{EMD}(X, Y)
\end{aligned}
$$

We have now reached a contradiction. It turns out that for any $L > 0$, there exist $X, Y \in \mathcal{S}(\mathbb{R}^d)$, $\mathbf{W} \in \mathbb{R}^{d \times d}$ and $\mathbf{q} \in \mathbb{R}^d$ such that $\|f_{\text{ATT}}(X) - f_{\text{ATT}}(Y)\| > L\, d_{EMD}(X, Y)$. Thus, the ATT mechanism is not Lipschitz continuous with respect to EMD.

#### B.5.2 THE ATT MECHANISM IS NOT LIPSCHITZ CONTINUOUS WITH RESPECT TO THE MATCHING DISTANCE

Suppose that the ATT mechanism is Lipschitz continuous with respect to the matching distance. Let $L > 0$ be given. Let also $\epsilon > 0$ and $c > \frac{2(1+\exp(d\epsilon))2L\epsilon}{\exp(d\epsilon)-1}$. Let $X = \{\mathbf{v}_1, \mathbf{v}_2\}$, $Y = \{\mathbf{u}_1, \mathbf{u}_2\}$ be two

multisets, each consisting of 2 vectors of dimension $d$. Then, suppose that $\mathbf{v}_1 = \mathbf{u}_1 = (c, c, \ldots, c)^\top$, $\mathbf{v}_2 = (\epsilon, \epsilon, \ldots, \epsilon)^\top$ and $\mathbf{u}_2 = (-\epsilon, -\epsilon, \ldots, -\epsilon)^\top$. Clearly, we have that $d_M(X, Y) = \|\mathbf{v}_2 - \mathbf{u}_2\| = 2\sqrt{d}\epsilon$. Let also $\mathbf{W} = -\mathbf{I}$ and $\mathbf{q} = \mathbf{1}$ where $\mathbf{I}$ denotes the $d \times d$ identity matrix and $\mathbf{1}$ denotes the $d$-dimensional vector of all ones. We choose $g$ to be the ReLU activation function. The attention coefficients of the elements of $X$ are equal to:

$$\alpha_1^X = \frac{\exp\left(\mathbf{1}^\top \mathrm{RELU}(-\mathbf{I}\,\mathbf{v}_1)\right)}{\sum_{j=1}^2 \exp\left(\mathbf{1}^\top \mathrm{RELU}(-\mathbf{I}\,\mathbf{v}_j)\right)} = \frac{\exp(0)}{\exp(0) + \exp(0)} = \frac{1}{2}$$

$$\alpha_2^X = \frac{\exp\left(\mathbf{1}^\top \mathrm{RELU}(-\mathbf{I}\,\mathbf{v}_2)\right)}{\sum_{j=1}^2 \exp\left(\mathbf{1}^\top \mathrm{RELU}(-\mathbf{I}\,\mathbf{v}_j)\right)} = \frac{\exp(0)}{\exp(0) + \exp(0)} = \frac{1}{2}$$

The attention coefficients of the elements of $Y$ are equal to:

$$\alpha_1^Y = \frac{\exp\left(\mathbf{1}^\top \mathrm{RELU}(-\mathbf{I}\,\mathbf{u}_1)\right)}{\sum_{j=1}^2 \exp\left(\mathbf{1}^\top \mathrm{RELU}(-\mathbf{I}\,\mathbf{u}_j)\right)} = \frac{\exp(0)}{\exp(0) + \exp(d\epsilon)} = \frac{1}{1 + \exp(d\epsilon)}$$

$$\alpha_2^Y = \frac{\exp\left(\mathbf{1}^\top \mathrm{RELU}(-\mathbf{I}\,\mathbf{u}_2)\right)}{\sum_{j=1}^2 \exp\left(\mathbf{1}^\top \mathrm{RELU}(-\mathbf{I}\,\mathbf{u}_j)\right)} = \frac{\exp(d\epsilon)}{\exp(0) + \exp(d\epsilon)} = \frac{\exp(d\epsilon)}{1 + \exp(d\epsilon)}$$

We then have that:

$$\left\|f_{\mathrm{ATT}}(X) - f_{\mathrm{ATT}}(Y)\right\| = \left\|\sum_{i=1}^2 \alpha_i^X \mathbf{v}_i - \sum_{j=1}^2 \alpha_j^Y \mathbf{u}_j\right\|$$

$$= \left\|\frac{1}{2}\mathbf{v}_1 + \frac{1}{2}\mathbf{v}_2 - \frac{1}{1 + \exp(d\epsilon)}\mathbf{u}_1 - \frac{\exp(d\epsilon)}{1 + \exp(d\epsilon)}\mathbf{u}_2\right\|$$

$$= \left\|\frac{\exp(d\epsilon) - 1}{2(1 + \exp(d\epsilon))}(c, c, \ldots, c)^\top + \frac{3\exp(d\epsilon) + 1}{2(1 + \exp(d\epsilon))}(\epsilon, \epsilon, \ldots, \epsilon)^\top\right\|$$

$$> \left\|\frac{\exp(d\epsilon) - 1}{2(1 + \exp(d\epsilon))}(c, c, \ldots, c)^\top\right\|$$

$$> \left\|\frac{\exp(d\epsilon) - 1}{2(1 + \exp(d\epsilon))}\left(\frac{2(1 + \exp(d\epsilon))2L\epsilon}{\exp(d\epsilon) - 1}, \frac{2(1 + \exp(d\epsilon))2L\epsilon}{\exp(d\epsilon) - 1}, \ldots, \frac{2(1 + \exp(d\epsilon))2L\epsilon}{\exp(d\epsilon) - 1}\right)^\top\right\|$$

$$= \left\|2L(\epsilon, \epsilon, \ldots, \epsilon)^\top\right\|$$

$$= 2L\sqrt{\underbrace{\epsilon^2 + \epsilon^2 + \ldots + \epsilon^2}_{d \text{ times}}}$$

$$= 2L\sqrt{d}\epsilon$$

$$= L\,d_M(X, Y)$$

Therefore, for any $L > 0$, there exist $X, Y \in \mathcal{S}(\mathbb{R}^d)$, $\mathbf{W} \in \mathbb{R}^{d \times d}$ and $\mathbf{q} \in \mathbb{R}^d$ such that $\|f_{\mathrm{ATT}}(X) - f_{\mathrm{ATT}}(Y)\| > L\,d_M(X, Y)$. Thus, the ATT mechanism is not Lipschitz continuous with respect to the matching distance.

### B.5.3 THE ATT MECHANISM IS NOT LIPSCHITZ CONTINUOUS WITH RESPECT TO THE HAUSDORFF DISTANCE

Suppose that the ATT mechanism is Lipschitz continuous with respect to the Hausdorff distance. Let $L > 0$ be given. Let also $\epsilon > 0$ and $c > \frac{2(1 + \exp(d\epsilon))2L\epsilon}{\exp(d\epsilon) - 1}$. Let $X = \{\mathbf{v}_1, \mathbf{v}_2\}$, $Y = \{\mathbf{u}_1, \mathbf{u}_2\}$ be two multisets, each consisting of 2 vectors of dimension $d$. Then, we set $\mathbf{v}_1 = \mathbf{u}_1 = (c, c, \ldots, c)^\top$, $\mathbf{v}_2 = (\epsilon, \epsilon, \ldots, \epsilon)^\top$ and $\mathbf{u}_2 = (-\epsilon, -\epsilon, \ldots, -\epsilon)^\top$. Clearly, we have that $d_H(X, Y) \leq \|\mathbf{v}_2 - \mathbf{u}_2\| = 2\sqrt{d}\epsilon$. Let also $\mathbf{W} = -\mathbf{I}$ and $\mathbf{q} = \mathbf{1}$ where $\mathbf{I}$ denotes the $d \times d$ identity matrix and $\mathbf{1}$ denotes the

$d$-dimensional vector of all ones. We choose $g$ to be the ReLU activation function. The attention coefficients of the elements of $X$ are equal to:

$$\alpha_1^X = \frac{\exp\left(\mathbf{1}^\top \text{RELU}(-\mathbf{I}\,\mathbf{v}_1)\right)}{\sum_{j=1}^{2} \exp\left(\mathbf{1}^\top \text{RELU}(-\mathbf{I}\,\mathbf{v}_j)\right)} = \frac{\exp(0)}{\exp(0) + \exp(0)} = \frac{1}{2}$$

$$\alpha_2^X = \frac{\exp\left(\mathbf{1}^\top \text{RELU}(-\mathbf{I}\,\mathbf{v}_2)\right)}{\sum_{j=1}^{2} \exp\left(\mathbf{1}^\top \text{RELU}(-\mathbf{I}\,\mathbf{v}_j)\right)} = \frac{\exp(0)}{\exp(0) + \exp(0)} = \frac{1}{2}$$

The attention coefficients of the elements of $Y$ are equal to:

$$\alpha_1^Y = \frac{\exp\left(\mathbf{1}^\top \text{RELU}(-\mathbf{I}\,\mathbf{u}_1)\right)}{\sum_{j=1}^{2} \exp\left(\mathbf{1}^\top \text{RELU}(-\mathbf{I}\,\mathbf{u}_j)\right)} = \frac{\exp(0)}{\exp(0) + \exp(d\epsilon)} = \frac{1}{1 + \exp(d\epsilon)}$$

$$\alpha_2^Y = \frac{\exp\left(\mathbf{1}^\top \text{RELU}(-\mathbf{I}\,\mathbf{u}_2)\right)}{\sum_{j=1}^{2} \exp\left(\mathbf{1}^\top \text{RELU}(-\mathbf{I}\,\mathbf{u}_j)\right)} = \frac{\exp(d\epsilon)}{\exp(0) + \exp(d\epsilon)} = \frac{\exp(d\epsilon)}{1 + \exp(d\epsilon)}$$

We then have that:

$$\left\| f_{\text{ATT}}(X) - f_{\text{ATT}}(Y) \right\| = \left\| \sum_{i=1}^{2} \alpha_i^X \mathbf{v}_i - \sum_{j=1}^{2} \alpha_j^Y \mathbf{u}_j \right\|$$

$$= \left\| \frac{1}{2}\mathbf{v}_1 + \frac{1}{2}\mathbf{v}_2 - \frac{1}{1 + \exp(d\epsilon)}\mathbf{u}_1 - \frac{\exp(d\epsilon)}{1 + \exp(d\epsilon)}\mathbf{u}_2 \right\|$$

$$= \left\| \frac{\exp(d\epsilon) - 1}{2(1 + \exp(d\epsilon))}(c, c, \ldots, c)^\top + \frac{3\exp(d\epsilon) + 1}{2(1 + \exp(d\epsilon))}(\epsilon, \epsilon, \ldots, \epsilon)^\top \right\|$$

$$> \left\| \frac{\exp(d\epsilon) - 1}{2(1 + \exp(d\epsilon))}(c, c, \ldots, c)^\top \right\|$$

$$> \left\| \frac{\exp(d\epsilon) - 1}{2(1 + \exp(d\epsilon))}\left( \frac{2(1 + \exp(d\epsilon))2L\epsilon}{\exp(d\epsilon) - 1}, \frac{2(1 + \exp(d\epsilon))2L\epsilon}{\exp(d\epsilon) - 1}, \ldots, \frac{2(1 + \exp(d\epsilon))2L\epsilon}{\exp(d\epsilon) - 1} \right)^\top \right\|$$

$$= \left\| 2L(\epsilon, \epsilon, \ldots, \epsilon)^\top \right\|$$

$$= 2L\sqrt{\underbrace{\epsilon^2 + \epsilon^2 + \ldots + \epsilon^2}_{d \text{ times}}}$$

$$= 2L\sqrt{d}\epsilon$$

$$= L\, d_H(X, Y)$$

Therefore, for any $L > 0$, there exist $X, Y \in \mathcal{S}(\mathbb{R}^d)$, $\mathbf{W} \in \mathbb{R}^{d \times d}$ and $\mathbf{q} \in \mathbb{R}^d$ such that $\|f_{\text{ATT}}(X) - f_{\text{ATT}}(Y)\| > L\, d_M(X, Y)$, which is a contradiction. Therefore, the ATT mechanism is not Lipschitz continuous with respect to the Hausdorff distance.

## B.6 Lipschitz Continuity of $\text{ATT}_{\ell_2}$ Mechanism

We demonstrate here that the attention mechanism is not Lipschitz continuous with respect to the considered functions, even when $\ell_2$ attention is used. Given a multiset $X = \{\!\{\mathbf{v}_1, \ldots, \mathbf{v}_m\}\!\} \in \mathcal{S}(\mathbb{R}^d)$, the attention mechanism is defined as follows:

$$f_{\text{ATT}_{\ell_2}}(X) = \sum_{i=1}^{m} \alpha_i \mathbf{v}_i \quad \text{where} \quad \alpha_i = \frac{\exp\left(-\left\|\mathbf{q} - g(\mathbf{W}\,\mathbf{v}_i)\right\|\right)}{\sum_{j=1}^{m} \exp\left(-\left\|\mathbf{q} - g(\mathbf{W}\,\mathbf{v}_j)\right\|\right)}$$

where $\mathbf{W} \in \mathbb{R}^{d' \times d}$ and $\mathbf{q} \in \mathbb{R}^{d'}$ denote a trainable matrix and a trainable vector, respectively, while $g$ denotes some activation function.

### B.6.1 THE $\text{ATT}_{\ell_2}$ MECHANISM IS NOT LIPSCHITZ CONTINUOUS WITH RESPECT TO EMD

Suppose that the $\text{ATT}_{\ell_2}$ mechanism is Lipschitz continuous with respect to EMD. Let $L > 0$ be given. Let also $\epsilon > 0$ and $c > \frac{2(1+\exp(-\sqrt{d}\epsilon))L\epsilon}{1-\exp(-\sqrt{d}\epsilon)}$. Let $X = \{\mathbf{v}_1, \mathbf{v}_2\}$, $Y = \{\mathbf{u}_1, \mathbf{u}_2\}$ be two multisets, each consisting of 2 vectors of dimension $d$. Then, suppose that $\mathbf{v}_1 = \mathbf{u}_1 = (c, c, \ldots, c)^\top$, $\mathbf{v}_2 = (\epsilon, \epsilon, \ldots, \epsilon)^\top$ and $\mathbf{u}_2 = (-\epsilon, -\epsilon, \ldots, -\epsilon)^\top$. Clearly, we have that $d_{EMD}(X, Y) = \frac{1}{2}\|\mathbf{v}_2 - \mathbf{u}_2\| = \sqrt{d}\epsilon$. Let also $\mathbf{W} = -\mathbf{I}$ and $\mathbf{q} = (-\epsilon, -\epsilon, \ldots, -\epsilon)^\top$ where $\mathbf{I}$ denotes the $d \times d$ identity matrix. We define $g$ as the ReLU function. The attention coefficients of the elements of $X$ are equal to:

$$\alpha_1^X = \frac{\exp\left(-\|\mathbf{q} - \text{RELU}(-\mathbf{I}\,\mathbf{v}_1)\|\right)}{\sum_{j=1}^{2}\exp\left(-\|\mathbf{q}^\top - \text{RELU}(-\mathbf{I}\,\mathbf{v}_j)\|\right)} = \frac{\exp\left(-\|\mathbf{q}\|\right)}{\exp\left(-\|\mathbf{q}\|\right) + \exp\left(-\|\mathbf{q}\|\right)} = \frac{1}{2}$$

$$\alpha_2^X = \frac{\exp\left(-\|\mathbf{q} - \text{RELU}(-\mathbf{I}\,\mathbf{v}_2)\|\right)}{\sum_{j=1}^{2}\exp\left(-\|\mathbf{q} - \text{RELU}(-\mathbf{I}\,\mathbf{v}_j)\|\right)} = \frac{\exp\left(-\|\mathbf{q}\|\right)}{\exp\left(-\|\mathbf{q}\|\right) + \exp\left(-\|\mathbf{q}\|\right)} = \frac{1}{2}$$

The attention coefficients of the elements of $Y$ are equal to:

$$\alpha_1^Y = \frac{\exp\left(-\|\mathbf{q} - \text{RELU}(-\mathbf{I}\,\mathbf{u}_1)\|\right)}{\sum_{j=1}^{2}\exp\left(-\|\mathbf{q} - \text{RELU}(-\mathbf{I}\,\mathbf{u}_j)\|\right)} = \frac{\exp\left(-\|\mathbf{q}\|\right)}{\exp\left(-\|\mathbf{q}\|\right) + \exp(0)} = \frac{\exp(-\sqrt{d}\epsilon)}{1 + \exp(-\sqrt{d}\epsilon)}$$

$$\alpha_2^Y = \frac{\exp\left(-\|\mathbf{q} - \text{RELU}(-\mathbf{I}\,\mathbf{u}_2)\|\right)}{\sum_{j=1}^{2}\exp\left(-\|\mathbf{q} - \text{RELU}(-\mathbf{I}\,\mathbf{u}_j)\|\right)} = \frac{\exp(0)}{\exp(0) + \exp\left(-\|\mathbf{q}\|\right)} = \frac{1}{1 + \exp(-\sqrt{d}\epsilon)}$$

We then have that:

$$\left\|f_{\text{ATT}_{\ell_2}}(X) - f_{\text{ATT}_{\ell_2}}(Y)\right\| = \left\|\sum_{i=1}^{2}\alpha_i^X\mathbf{v}_i - \sum_{j=1}^{2}\alpha_j^Y\mathbf{u}_j\right\|$$

$$= \left\|\frac{1}{2}\mathbf{v}_1 + \frac{1}{2}\mathbf{v}_2 - \frac{\exp(-\sqrt{d}\epsilon)}{1 + \exp(-\sqrt{d}\epsilon)}\mathbf{u}_1 - \frac{1}{1 + \exp(-\sqrt{d}\epsilon)}\mathbf{u}_2\right\|$$

$$= \left\|\frac{1 - \exp(-\sqrt{d}\epsilon)}{2(1 + \exp(-\sqrt{d}\epsilon))}(c, c, \ldots, c)^\top + \frac{\exp(-\sqrt{d}\epsilon) + 3}{2(1 + \exp(-\sqrt{d}\epsilon))}(\epsilon, \epsilon, \ldots, \epsilon)^\top\right\|$$

$$> \left\|\frac{1 - \exp(-\sqrt{d}\epsilon)}{2(1 + \exp(-\sqrt{d}\epsilon))}(c, c, \ldots, c)^\top\right\|$$

$$> \left\|\frac{1 - \exp(-\sqrt{d}\epsilon)}{2(1 + \exp(-\sqrt{d}\epsilon))}\left(\frac{2(1 + \exp(-\sqrt{d}\epsilon))L\epsilon}{1 - \exp(-\sqrt{d}\epsilon)}, \ldots, \frac{2(1 + \exp(-\sqrt{d}\epsilon))L\epsilon}{1 - \exp(-\sqrt{d}\epsilon)}\right)^\top\right\|$$

$$= \left\|L(\epsilon, \epsilon, \ldots, \epsilon)^\top\right\|$$

$$= L\sqrt{\underbrace{\epsilon^2 + \epsilon^2 + \ldots + \epsilon^2}_{d \text{ times}}}$$

$$= L\sqrt{d}\epsilon$$

$$= L\,d_{EMD}(X, Y)$$

We have now reached a contradiction. It turns out that for any $L > 0$, there exist $X, Y \in \mathcal{S}(\mathbb{R}^d)$, $\mathbf{W} \in \mathbb{R}^{d \times d}$ and $\mathbf{q} \in \mathbb{R}^d$ such that $\|f_{\text{ATT}_{\ell_2}}(X) - f_{\text{ATT}_{\ell_2}}(Y)\| > L\,d_{EMD}(X, Y)$. Thus, the $\text{ATT}_{\ell_2}$ mechanism is not Lipschitz continuous with respect to EMD.

### B.6.2 THE $\text{ATT}_{\ell_2}$ MECHANISM IS NOT LIPSCHITZ CONTINUOUS WITH RESPECT TO THE MATCHING DISTANCE

Suppose that the $\text{ATT}_{\ell_2}$ mechanism is Lipschitz continuous with respect to the matching distance. Let $L > 0$ be given. Let also $\epsilon > 0$ and $c > \frac{2(1+\exp(-\sqrt{d}\epsilon))2L\epsilon}{1-\exp(-\sqrt{d}\epsilon)}$. Let $X = \{\mathbf{v}_1, \mathbf{v}_2\}$, $Y =$

$\{\mathbf{u}_1, \mathbf{u}_2\}$ be two multisets, each consisting of 2 vectors of dimension $d$. Then, suppose that $\mathbf{v}_1 = \mathbf{u}_1 = (c, c, \ldots, c)^\top$, $\mathbf{v}_2 = (\epsilon, \epsilon, \ldots, \epsilon)^\top$ and $\mathbf{u}_2 = (-\epsilon, -\epsilon, \ldots, -\epsilon)^\top$. Clearly, we have that $d_M(X, Y) = \|\mathbf{v}_2 - \mathbf{u}_2\| = 2\sqrt{d}\epsilon$. Let also $\mathbf{W} = -\mathbf{I}$ and $\mathbf{q} = (-\epsilon, -\epsilon, \ldots, -\epsilon)^\top$ where $\mathbf{I}$ denotes the $d \times d$ identity matrix. We choose $g$ to be the ReLU activation function. The attention coefficients of the elements of $X$ are equal to:

$$\alpha_1^X = \frac{\exp\left(-\|\mathbf{q} - \mathrm{RELU}(-\mathbf{I}\,\mathbf{v}_1)\|\right)}{\sum_{j=1}^2 \exp\left(-\|\mathbf{q}^\top - \mathrm{RELU}(-\mathbf{I}\,\mathbf{v}_j)\|\right)} = \frac{\exp\left(-\|\mathbf{q}\|\right)}{\exp\left(-\|\mathbf{q}\|\right) + \exp\left(-\|\mathbf{q}\|\right)} = \frac{1}{2}$$

$$\alpha_2^X = \frac{\exp\left(-\|\mathbf{q} - \mathrm{RELU}(-\mathbf{I}\,\mathbf{v}_2)\|\right)}{\sum_{j=1}^2 \exp\left(-\|\mathbf{q} - \mathrm{RELU}(-\mathbf{I}\,\mathbf{v}_j)\|\right)} = \frac{\exp\left(-\|\mathbf{q}\|\right)}{\exp\left(-\|\mathbf{q}\|\right) + \exp\left(-\|\mathbf{q}\|\right)} = \frac{1}{2}$$

The attention coefficients of the elements of $Y$ are equal to:

$$\alpha_1^Y = \frac{\exp\left(-\|\mathbf{q} - \mathrm{RELU}(-\mathbf{I}\,\mathbf{u}_1)\|\right)}{\sum_{j=1}^2 \exp\left(-\|\mathbf{q} - \mathrm{RELU}(-\mathbf{I}\,\mathbf{u}_j)\|\right)} = \frac{\exp\left(-\|\mathbf{q}\|\right)}{\exp\left(-\|\mathbf{q}\|\right) + \exp(0)} = \frac{\exp(-\sqrt{d}\epsilon)}{1 + \exp(-\sqrt{d}\epsilon)}$$

$$\alpha_2^Y = \frac{\exp\left(-\|\mathbf{q} - \mathrm{RELU}(-\mathbf{I}\,\mathbf{u}_2)\|\right)}{\sum_{j=1}^2 \exp\left(-\|\mathbf{q} - \mathrm{RELU}(-\mathbf{I}\,\mathbf{u}_j)\|\right)} = \frac{\exp(0)}{\exp(0) + \exp\left(-\|\mathbf{q}\|\right)} = \frac{1}{1 + \exp(-\sqrt{d}\epsilon)}$$

We then have that:

$$\left\|f_{\mathrm{ATT}_{\ell_2}}(X) - f_{\mathrm{ATT}_{\ell_2}}(Y)\right\| = \left\|\sum_{i=1}^2 \alpha_i^X \mathbf{v}_i - \sum_{j=1}^2 \alpha_j^Y \mathbf{u}_j\right\|$$

$$= \left\|\frac{1}{2}\mathbf{v}_1 + \frac{1}{2}\mathbf{v}_2 - \frac{\exp(-\sqrt{d}\epsilon)}{1 + \exp(-\sqrt{d}\epsilon)}\mathbf{u}_1 - \frac{1}{1 + \exp(-\sqrt{d}\epsilon)}\mathbf{u}_2\right\|$$

$$= \left\|\frac{1 - \exp(-\sqrt{d}\epsilon)}{2(1 + \exp(-\sqrt{d}\epsilon))}(c, c, \ldots, c)^\top + \frac{\exp(-\sqrt{d}\epsilon) + 3}{2(1 + \exp(-\sqrt{d}\epsilon))}(\epsilon, \epsilon, \ldots, \epsilon)^\top\right\|$$

$$> \left\|\frac{1 - \exp(-\sqrt{d}\epsilon)}{2(1 + \exp(-\sqrt{d}\epsilon))}(c, c, \ldots, c)^\top\right\|$$

$$> \left\|\frac{1 - \exp(-\sqrt{d}\epsilon)}{2(1 + \exp(-\sqrt{d}\epsilon))}\left(\frac{2(1 + \exp(-\sqrt{d}\epsilon))2L\epsilon}{1 - \exp(-\sqrt{d}\epsilon)}, \ldots, \frac{2(1 + \exp(-\sqrt{d}\epsilon))2L\epsilon}{1 - \exp(-\sqrt{d}\epsilon)}\right)^\top\right\|$$

$$= \left\|2L(\epsilon, \epsilon, \ldots, \epsilon)^\top\right\|$$

$$= 2L\sqrt{\underbrace{\epsilon^2 + \epsilon^2 + \ldots + \epsilon^2}_{d \text{ times}}}$$

$$= 2L\sqrt{d}\epsilon$$

$$= L\,d_M(X, Y)$$

Therefore, for any $L > 0$, there exist $X, Y \in \mathcal{S}(\mathbb{R}^d)$, $\mathbf{W} \in \mathbb{R}^{d \times d}$ and $\mathbf{q} \in \mathbb{R}^d$ such that $\|f_{\mathrm{ATT}_{\ell_2}}(X) - f_{\mathrm{ATT}_{\ell_2}}(Y)\| > L\,d_M(X, Y)$. Thus, the $\mathrm{ATT}_{\ell_2}$ mechanism is not Lipschitz continuous with respect to the matching distance.

### B.6.3 THE $\mathrm{ATT}_{\ell_2}$ MECHANISM IS NOT LIPSCHITZ CONTINUOUS WITH RESPECT TO THE HAUSDORFF DISTANCE

Suppose that the $\mathrm{ATT}_{\ell_2}$ mechanism is Lipschitz continuous with respect to the Hausdorff distance. Let $L > 0$ be given. Let also $\epsilon > 0$ and $c > \frac{2(1+\exp(-\sqrt{d}\epsilon))2L\epsilon}{1-\exp(-\sqrt{d}\epsilon)}$. Let $X = \{\mathbf{v}_1, \mathbf{v}_2\}$, $Y = \{\mathbf{u}_1, \mathbf{u}_2\}$ be two multisets, each consisting of 2 vectors of dimension $d$. Then, we set $\mathbf{v}_1 = \mathbf{u}_1 = (c, c, \ldots, c)^\top$, $\mathbf{v}_2 = (\epsilon, \epsilon, \ldots, \epsilon)^\top$ and $\mathbf{u}_2 = (-\epsilon, -\epsilon, \ldots, -\epsilon)^\top$. Clearly, we have that $d_H(X, Y) \leq$

$\|\mathbf{v}_2 - \mathbf{u}_2\| = 2\sqrt{d}\epsilon$. Let also $\mathbf{W} = -\mathbf{I}$ and $\mathbf{q} = \mathbf{1}$ where $\mathbf{I}$ denotes the $d \times d$ identity matrix and $\mathbf{1}$ denotes the $d$-dimensional vector of all ones. We choose $g$ to be the ReLU activation function. The attention coefficients of the elements of $X$ are equal to:

$$\alpha_1^X = \frac{\exp\left(-\|\mathbf{q} - \mathrm{RELU}(-\mathbf{I}\,\mathbf{v}_1)\|\right)}{\sum_{j=1}^2 \exp\left(-\|\mathbf{q}^\top - \mathrm{RELU}(-\mathbf{I}\,\mathbf{v}_j)\|\right)} = \frac{\exp\left(-\|\mathbf{q}\|\right)}{\exp\left(-\|\mathbf{q}\|\right) + \exp\left(-\|\mathbf{q}\|\right)} = \frac{1}{2}$$

$$\alpha_2^X = \frac{\exp\left(-\|\mathbf{q} - \mathrm{RELU}(-\mathbf{I}\,\mathbf{v}_2)\|\right)}{\sum_{j=1}^2 \exp\left(-\|\mathbf{q} - \mathrm{RELU}(-\mathbf{I}\,\mathbf{v}_j)\|\right)} = \frac{\exp\left(-\|\mathbf{q}\|\right)}{\exp\left(-\|\mathbf{q}\|\right) + \exp\left(-\|\mathbf{q}\|\right)} = \frac{1}{2}$$

The attention coefficients of the elements of $Y$ are equal to:

$$\alpha_1^Y = \frac{\exp\left(-\|\mathbf{q} - \mathrm{RELU}(-\mathbf{I}\,\mathbf{u}_1)\|\right)}{\sum_{j=1}^2 \exp\left(-\|\mathbf{q} - \mathrm{RELU}(-\mathbf{I}\,\mathbf{u}_j)\|\right)} = \frac{\exp\left(-\|\mathbf{q}\|\right)}{\exp\left(-\|\mathbf{q}\|\right) + \exp(0)} = \frac{\exp(-\sqrt{d}\epsilon)}{1 + \exp(-\sqrt{d}\epsilon)}$$

$$\alpha_2^Y = \frac{\exp\left(-\|\mathbf{q} - \mathrm{RELU}(-\mathbf{I}\,\mathbf{u}_2)\|\right)}{\sum_{j=1}^2 \exp\left(-\|\mathbf{q} - \mathrm{RELU}(-\mathbf{I}\,\mathbf{u}_j)\|\right)} = \frac{\exp(0)}{\exp(0) + \exp\left(-\|\mathbf{q}\|\right)} = \frac{1}{1 + \exp(-\sqrt{d}\epsilon)}$$

We then have that:

$$
\begin{aligned}
\left\| f_{\mathrm{ATT}_{\ell_2}}(X) - f_{\mathrm{ATT}_{\ell_2}}(Y) \right\| &= \left\| \sum_{i=1}^2 \alpha_i^X \mathbf{v}_i - \sum_{j=1}^2 \alpha_j^Y \mathbf{u}_j \right\| \\
&= \left\| \frac{1}{2}\mathbf{v}_1 + \frac{1}{2}\mathbf{v}_2 - \frac{\exp(-\sqrt{d}\epsilon)}{1 + \exp(-\sqrt{d}\epsilon)}\mathbf{u}_1 - \frac{1}{1 + \exp(-\sqrt{d}\epsilon)}\mathbf{u}_2 \right\| \\
&= \left\| \frac{1 - \exp(-\sqrt{d}\epsilon)}{2(1 + \exp(-\sqrt{d}\epsilon))}(c, c, \dots, c)^\top + \frac{\exp(-\sqrt{d}\epsilon) + 3}{2(1 + \exp(-\sqrt{d}\epsilon))}(\epsilon, \epsilon, \dots, \epsilon)^\top \right\| \\
&> \left\| \frac{1 - \exp(-\sqrt{d}\epsilon)}{2(1 + \exp(-\sqrt{d}\epsilon))}(c, c, \dots, c)^\top \right\| \\
&> \left\| \frac{1 - \exp(-\sqrt{d}\epsilon)}{2(1 + \exp(-\sqrt{d}\epsilon))}\left( \frac{2(1 + \exp(-\sqrt{d}\epsilon))2L\epsilon}{1 - \exp(-\sqrt{d}\epsilon)}, \dots, \frac{2(1 + \exp(-\sqrt{d}\epsilon))2L\epsilon}{1 - \exp(-\sqrt{d}\epsilon)} \right)^\top \right\| \\
&= \left\| 2L(\epsilon, \epsilon, \dots, \epsilon)^\top \right\| \\
&= 2L\sqrt{\underbrace{\epsilon^2 + \epsilon^2 + \dots + \epsilon^2}_{d \text{ times}}} \\
&= 2L\sqrt{d}\epsilon \\
&= L\, d_H(X, Y)
\end{aligned}
$$

Therefore, for any $L > 0$, there exist $X, Y \in \mathcal{S}(\mathbb{R}^d)$, $\mathbf{W} \in \mathbb{R}^{d \times d}$ and $\mathbf{q} \in \mathbb{R}^d$ such that $\|f_{\mathrm{ATT}_{\ell_2}}(X) - f_{\mathrm{ATT}_{\ell_2}}(Y)\| > L\, d_M(X, Y)$, which is a contradiction. Therefore, the $\mathrm{ATT}_{\ell_2}$ mechanism is not Lipschitz continuous with respect to the Hausdorff distance.

### B.7 PROOF OF THEOREM 3.4

#### B.7.1 THE NN$_{\mathrm{MEAN}}$ MODEL IS LIPSCHITZ CONTINUOUS WITH RESPECT TO EMD

Let $X = \{\!\{\mathbf{v}_1, \dots, \mathbf{v}_m\}\!\}$, $Y = \{\!\{\mathbf{u}_1, \dots, \mathbf{u}_n\}\!\}$ be two multisets of vectors. Let also $\mathrm{Lip}(f_{\mathrm{MLP}_1})$ and $\mathrm{Lip}(f_{\mathrm{MLP}_2})$ denote the Lipschitz constants of $f_{\mathrm{MLP}_1}$ and $f_{\mathrm{MLP}_2}$, respectively. Finally, let $\mathbf{F}^*$ denote the matrix that minimizes $d_{\mathrm{EMD}}(X, Y)$. Then, we have:

$$\left\| f_{\mathrm{MLP}_2}\left( \frac{1}{m}\sum_{i=1}^m f_{\mathrm{MLP}_1}(\mathbf{v}_i) \right) - f_{\mathrm{MLP}_2}\left( \frac{1}{n}\sum_{j=1}^n f_{\mathrm{MLP}_1}(\mathbf{u}_j) \right) \right\| \leq \mathrm{Lip}(f_{\mathrm{MLP}_2}) \left\| \frac{1}{m}\sum_{i=1}^m f_{\mathrm{MLP}_1}(\mathbf{v}_i) - \frac{1}{n}\sum_{j=1}^n f_{\mathrm{MLP}_1}(\mathbf{u}_j) \right\|$$

$$\leq \mathrm{Lip}(f_{\mathrm{MLP}_2}) \sum_{i=1}^{m} \sum_{j=1}^{n} [\mathbf{F}^*]_{ij} \| f_{\mathrm{MLP}_1}(\mathbf{v}_i) - f_{\mathrm{MLP}_1}(\mathbf{v}_j) \|$$
$$\text{(due to Theorem 3.1)}$$

$$\leq \mathrm{Lip}(f_{\mathrm{MLP}_2}) \sum_{i=1}^{m} \sum_{j=1}^{n} [\mathbf{F}^*]_{ij} \mathrm{Lip}(f_{\mathrm{MLP}_1}) \| \mathbf{v}_i - \mathbf{v}_j \|$$

$$= \mathrm{Lip}(f_{\mathrm{MLP}_1}) \mathrm{Lip}(f_{\mathrm{MLP}_2}) \sum_{i=1}^{m} \sum_{j=1}^{n} [\mathbf{F}^*]_{ij} \| \mathbf{v}_i - \mathbf{v}_j \|$$

$$= \mathrm{Lip}(f_{\mathrm{MLP}_1}) \mathrm{Lip}(f_{\mathrm{MLP}_2}) d_{\mathrm{EMD}}(X, Y)$$

Note that whether the above can hold as an equality depends on $f_{\mathrm{MLP}_1}$ and $f_{\mathrm{MLP}_2}$, and therefore, the Lipschitz constant of the $\mathrm{NN}_{\mathrm{MEAN}}$ model is upper bounded by $\mathrm{Lip}(f_{\mathrm{MLP}_1}) \mathrm{Lip}(f_{\mathrm{MLP}_2}) d_{\mathrm{EMD}}(X, Y)$.

### B.7.2 THERE EXIST $\mathrm{NN}_{\mathrm{SUM}}$ MODELS WHICH ARE NOT LIPSCHITZ CONTINUOUS WITH RESPECT TO THE MATCHING DISTANCE

Suppose that $\mathrm{NN}_{\mathrm{SUM}}$ is a neural network model which computes its output as follows:

$$\mathbf{v}_X = f_2 \Big( \mathrm{ReLU} \Big( \{\!\{ f_1(v_1), \dots, f_1(v_m) \}\!\} \Big) \Big)$$

where $X = \{\!\{ v_1, \dots, v_m \}\!\}$ is a multiset of scalars, and $f_1$ and $f_2$ are fully-connected layers, i. e., $f_1(x) = a_1 x + b_1$ and $f_2(x) = a_2 x + b_2$. Furthermore, suppose that $a_1 > 0$, $b_1 > 0$ and $a_2 > 0$. Note that the Lipschitz constants of $f_1$ and $f_2$ are $\mathrm{Lip}(f_1) = a_1$ and $\mathrm{Lip}(f_2) = a_2$, respectively.

Suppose that the $\mathrm{NN}_{\mathrm{SUM}}$ model is Lipschitz continuous with respect to the matching distance. Let $L > 0$ be given. Let also $X = \{\!\{ v_1, v_2 \}\!\}$, $Y = \{\!\{ u_1 \}\!\}$ be two multisets that contain real numbers. We construct the two sets such that $v_1 = v_2 = u_1 = c$ where $0 < c < \frac{b_1}{L a_1}$. Thus, we have that $d_M(X, Y) = |c| = c$. We also have that $b_1 > c L a_1$. Then, we have:

$$\left| f_2 \Big( \sum_{j=1}^{2} f_1(c) \Big) - f_2 \Big( f_1(c) \Big) \right| = \left| f_2 \Big( \sum_{j=1}^{2} \mathrm{ReLU}(a_1 c + b_1) \Big) - f_2 \Big( \mathrm{ReLU}(a_1 c + b_1) \Big) \right|$$

$$= \left| f_2 \Big( 2 (a_1 c + b_1) \Big) - f_2 \Big( a_1 c + b_1 \Big) \right|$$

$$= \left| a_2 \Big( 2 (a_1 c + b_1) \Big) + b_2 - a_2 \Big( a_1 c + b_1 \Big) - b_2 \right|$$

$$= |a_2 (a_1 c + b_1)|$$

$$> |a_2 (a_1 c + c L a_1)|$$

$$= |(L + 1) a_2 a_1 c|$$

$$= (L + 1) \mathrm{Lip}(f_2) \mathrm{Lip}(f_1) c$$

$$> L \mathrm{Lip}(f_2) \mathrm{Lip}(f_1) d_M(X, Y)$$

Therefore, for any $L > 0$, there exist $X, Y \in \mathcal{S}(\mathbb{R}^d)$ such that $\left\| f_2 \Big( \sum_{\mathbf{v} \in X} f_1(\mathbf{v}) \Big) - f_2 \Big( \sum_{\mathbf{u} \in Y} f_1(\mathbf{u}) \Big) \right\| > L \mathrm{Lip}(f_2) \mathrm{Lip}(f_1) d_M(X, Y)$. Based on the above, there exist $\mathrm{NN}_{\mathrm{SUM}}$ models which are not Lipschitz continuous with respect to the matching distance.

### B.7.3 THE $\mathrm{NN}_{\mathrm{MAX}}$ MODEL IS LIPSCHITZ CONTINUOUS WITH RESPECT TO THE HAUSDORFF DISTANCE

Let $X = \{\!\{ \mathbf{v}_1, \dots, \mathbf{v}_m \}\!\}$, $Y = \{\!\{ \mathbf{u}_1, \dots, \mathbf{u}_n \}\!\}$ be two multisets of vectors. Let also $\mathrm{Lip}(f_{\mathrm{MLP}_1})$ and $\mathrm{Lip}(f_{\mathrm{MLP}_2})$ denote the Lipschitz constants of $f_{\mathrm{MLP}_1}$ and $f_{\mathrm{MLP}_2}$, respectively. Let $f_{\mathrm{MLP}_1}(X) = \{\!\{ f_{\mathrm{MLP}_1}(\mathbf{v}_1), \dots, f_{\mathrm{MLP}_1}(\mathbf{v}_m) \}\!\}$ and $f_{\mathrm{MLP}_1}(Y) = \{\!\{ f_{\mathrm{MLP}_1}(\mathbf{u}_1), \dots, f_{\mathrm{MLP}_1}(\mathbf{u}_n) \}\!\}$. Let also $\big( f_{\mathrm{MLP}_1}(X) \big)_{\max}$ and $\big( f_{\mathrm{MLP}_1}(Y) \big)_{\max}$ denote the output of the max operator applied to multisets $f_{\mathrm{MLP}_1}(X)$ and $f_{\mathrm{MLP}_1}(Y)$, respectively. Without loss of generality, we also assume that $h \Big( f_{\mathrm{MLP}_1}(X), f_{\mathrm{MLP}_1}(Y) \Big) \geq h \Big( f_{\mathrm{MLP}_1}(Y), f_{\mathrm{MLP}_1}(X) \Big)$. Then, we have:

$$\left\| f_{\mathrm{MLP}_2} \Big( \big( f_{\mathrm{MLP}_1}(X) \big)_{\max} \Big) - f_{\mathrm{MLP}_2} \Big( \big( f_{\mathrm{MLP}_1}(Y) \big)_{\max} \Big) \right\| \leq \mathrm{Lip}(f_{\mathrm{MLP}_2}) \left\| \big( f_{\mathrm{MLP}_1}(\mathbf{v}_i) \big)_{\max} - \big( f_{\mathrm{MLP}_1}(\mathbf{u}_j) \big)_{\max} \right\|$$

$$\leq \text{Lip}(f_{\text{MLP}_2})\sqrt{d}\; d_H\Big(f_{\text{MLP}_1}(X), f_{\text{MLP}_1}(Y)\Big) \text{ (due to Theorem 3.1)}$$

$$= \sqrt{d}\text{Lip}(f_{\text{MLP}_2}) \max_{i\in[m]} \min_{j\in[n]} \left\|f_{\text{MLP}_1}(\mathbf{v}_i) - f_{\text{MLP}_1}(\mathbf{u}_j)\right\|$$

$$\leq \sqrt{d}\text{Lip}(f_{\text{MLP}_2})\text{Lip}(f_{\text{MLP}_1}) \max_{i\in[m]} \min_{j\in[n]} \|\mathbf{v}_i - \mathbf{u}_j\|$$

$$\leq \sqrt{d}\text{Lip}(f_{\text{MLP}_1})\text{Lip}(f_{\text{MLP}_2})d_H(X, Y)$$

We have thus shown that the $\text{NN}_{\text{MAX}}$ model is Lipschitz continuous with respect to the Hausdorff distance and its Lipschitz constant is upper bounded by $\sqrt{d}\text{Lip}(f_{\text{MLP}_1})\text{Lip}(f_{\text{MLP}_2})$.

## B.8 PROOF OF LEMMA 3.5

### B.8.1 THE $\text{NN}_{\text{MEAN}}$ MODEL IS LIPSCHITZ CONTINUOUS WITH RESPECT TO THE MATCHING DISTANCE

Let $\mathcal{X}$ denote a set that contains multisets of vectors of equal cardinalities, i.e., $|X| = M$ and $X \in \mathcal{S}(\mathbb{R}^d)$, $\forall X \in \mathcal{X}$ where $M \in \mathbb{N}$. Let $X, Y \in \mathcal{X}$ denote two multisets. We have shown in subsection B.7.1 above that:

$$\left\|f_{\text{MLP}_2}\left(\frac{1}{m}\sum_{i=1}^{m} f_{\text{MLP}_1}(\mathbf{v}_i)\right) - f_{\text{MLP}_2}\left(\frac{1}{n}\sum_{j=1}^{n} f_{\text{MLP}_1}(\mathbf{u}_j)\right)\right\| \leq \text{Lip}(f_{\text{MLP}_1})\text{Lip}(f_{\text{MLP}_2})d_{\text{EMD}}(X, Y)$$

By Proposition 2.3, we have that $d_M(X, Y) = Md_{\text{EMD}}(X, Y)$. Therefore, we have that:

$$\left\|f_{\text{MLP}_2}\left(\frac{1}{m}\sum_{i=1}^{m} f_{\text{MLP}_1}(\mathbf{v}_i)\right) - f_{\text{MLP}_2}\left(\frac{1}{n}\sum_{j=1}^{n} f_{\text{MLP}_1}(\mathbf{u}_j)\right)\right\| \leq \text{Lip}(f_{\text{MLP}_1})\text{Lip}(f_{\text{MLP}_2})d_{\text{EMD}}(X, Y)$$

$$= \frac{1}{M}\text{Lip}(f_{\text{MLP}_1})\text{Lip}(f_{\text{MLP}_2})d_M(X, Y) \text{ (due to Proposition 2.3)}$$

The $\text{NN}_{\text{MEAN}}$ model is Lipschitz continuous with respect to the matching distance and its Lipschitz constant is upper bounded by $\frac{1}{M}\text{Lip}(f_{\text{MLP}_1})\text{Lip}(f_{\text{MLP}_2})$.

### B.8.2 THE $\text{NN}_{\text{SUM}}$ MODEL IS LIPSCHITZ CONTINUOUS WITH RESPECT TO THE MATCHING DISTANCE

Let $\mathcal{X}$ denote a set that contains multisets of vectors of equal cardinalities, i.e., $|X| = M$ and $X \in \mathcal{S}(\mathbb{R}^d)$, $\forall X \in \mathcal{X}$ where $M \in \mathbb{N}$. Let $X, Y \in \mathcal{X}$ denote two multisets. Let $\pi^*$ denote the matching produced by the solution of the matching distance function:

$$\pi^* = \arg\min_{\pi\in\mathfrak{S}_M} \sum_{i=1}^{M} \|\mathbf{v}_{\pi(i)} - \mathbf{u}_i\|$$

Then, we have:

$$\left\|f_{\text{MLP}_2}\left(\sum_{i=1}^{M} f_{\text{MLP}_1}(\mathbf{v}_i)\right) - f_{\text{MLP}_2}\left(\sum_{j=1}^{M} f_{\text{MLP}_1}(\mathbf{u}_j)\right)\right\| \leq \text{Lip}(f_{\text{MLP}_2})\left\|\sum_{i=1}^{M} f_{\text{MLP}_1}(\mathbf{v}_i) - \sum_{j=1}^{M} f_{\text{MLP}_1}(\mathbf{u}_j)\right\|$$

$$= \text{Lip}(f_{\text{MLP}_2})\left\|\sum_{i=1}^{M}\left(f_{\text{MLP}_1}(\mathbf{v}_{\pi^*(i)}) - f_{\text{MLP}_1}(\mathbf{u}_i)\right)\right\|$$

$$\leq \text{Lip}(f_{\text{MLP}_2})\sum_{i=1}^{M}\left\|f_{\text{MLP}_1}(\mathbf{v}_{\pi^*(i)}) - f_{\text{MLP}_1}(\mathbf{u}_i)\right\|$$

$$\leq \text{Lip}(f_{\text{MLP}_2})\sum_{i=1}^{M}\text{Lip}(f_{\text{MLP}_1})\|\mathbf{v}_{\pi^*(i)} - \mathbf{u}_i\|$$

$$\leq \text{Lip}(f_{\text{MLP}_1})\text{Lip}(f_{\text{MLP}_2})\sum_{i=1}^{M}\|\mathbf{v}_{\pi^*(i)} - \mathbf{u}_i\|$$

$$= \mathrm{Lip}(f_{\mathrm{MLP}_1})\mathrm{Lip}(f_{\mathrm{MLP}_2})d_M(X,Y)$$

which concludes the proof. The $\mathrm{NN}_{\mathrm{SUM}}$ model is thus Lipschitz continuous with respect to the matching distance when the inputs are restricted to multisets of a fixed size, and its Lipschitz constant is upper bounded by $\mathrm{Lip}(f_{\mathrm{MLP}_1})\mathrm{Lip}(f_{\mathrm{MLP}_2})$.

### B.8.3 THE $\mathrm{NN}_{\mathrm{SUM}}$ MODEL IS LIPSCHITZ CONTINUOUS WITH RESPECT TO EMD

Let $\mathcal{X}$ denote a set that contains multisets of vectors of equal cardinalities, i.e., $|X| = M$ and $X \in \mathcal{S}(\mathbb{R}^d)$, $\forall X \in \mathcal{X}$ where $M \in \mathbb{N}$. Let $X, Y \in \mathcal{X}$ denote two multisets. We have shown in subsection B.8.2 above that:

$$\left\| f_{\mathrm{MLP2}}\left( \sum_{i=1}^{M} f_{\mathrm{MLP}_1}(\mathbf{v}_i) \right) - f_{\mathrm{MLP2}}\left( \sum_{j=1}^{M} f_{\mathrm{MLP}_1}(\mathbf{u}_j) \right) \right\| \leq \mathrm{Lip}(f_{\mathrm{MLP}_1})\mathrm{Lip}(f_{\mathrm{MLP}_2})d_M(X,Y)$$

By Proposition 2.3, we have that $d_M(X,Y) = Md_{\mathrm{EMD}}(X,Y)$. Therefore, we have that:

$$\left\| f_{\mathrm{MLP2}}\left( \sum_{i=1}^{M} f_{\mathrm{MLP}_1}(\mathbf{v}_i) \right) - f_{\mathrm{MLP2}}\left( \sum_{j=1}^{M} f_{\mathrm{MLP}_1}(\mathbf{u}_j) \right) \right\| \leq \mathrm{Lip}(f_{\mathrm{MLP}_1})\mathrm{Lip}(f_{\mathrm{MLP}_2})d_M(X,Y)$$

$$= M\mathrm{Lip}(f_{\mathrm{MLP}_1})\mathrm{Lip}(f_{\mathrm{MLP}_2})d_{\mathrm{EMD}}(X,Y) \text{ (due to Proposition 2.3)}$$

Therefore, the $\mathrm{NN}_{\mathrm{SUM}}$ model is Lipschitz continuous with respect to EMD and its Lipschitz constant is upper bounded by $M\mathrm{Lip}(f_{\mathrm{MLP}_1})\mathrm{Lip}(f_{\mathrm{MLP}_2})$.

### B.8.4 THE $\mathrm{NN}_{\mathrm{MAX}}$ MODEL IS LIPSCHITZ CONTINUOUS WITH RESPECT TO THE MATCHING DISTANCE

Let $\mathcal{X}$ denote a set that contains multisets of vectors of equal cardinalities, i.e., $|X| = M$ and $X \in \mathcal{S}(\mathbb{R}^d)$, $\forall X \in \mathcal{X}$ where $M \in \mathbb{N}$. Let $X, Y \in \mathcal{X}$ denote two multisets. Let $f_{\mathrm{MLP}_1}(X) = \left\{\!\!\left\{ f_{\mathrm{MLP}_1}(\mathbf{v}_1), \ldots, f_{\mathrm{MLP}_1}(\mathbf{v}_m) \right\}\!\!\right\}$ and $f_{\mathrm{MLP}_1}(Y) = \left\{\!\!\left\{ f_{\mathrm{MLP}_1}(\mathbf{u}_1), \ldots, f_{\mathrm{MLP}_1}(\mathbf{u}_n) \right\}\!\!\right\}$. Let also $\left(f_{\mathrm{MLP}_1}(X)\right)_{\max}$ and $\left(f_{\mathrm{MLP}_1}(Y)\right)_{\max}$ denote the output of the max operator applied to multisets $f_{\mathrm{MLP}_1}(X)$ and $f_{\mathrm{MLP}_1}(Y)$, respectively. Let $\pi^*$ denote the matching produced by the solution of the matching distance function:

$$\pi^* = \arg\min_{\pi \in \mathfrak{S}_M} \sum_{i=1}^{M} \|\mathbf{v}_{\pi(i)} - \mathbf{u}_i\|$$

Then, we have:

$$\left\| f_{\mathrm{MLP2}}\left( \left(f_{\mathrm{MLP}_1}(X)\right)_{\max} \right) - f_{\mathrm{MLP2}}\left( \left(f_{\mathrm{MLP}_1}(Y)\right)_{\max} \right) \right\| \leq \mathrm{Lip}(f_{\mathrm{MLP}_2}) \left\| \left(f_{\mathrm{MLP}_1}(X)\right)_{\max} - \left(f_{\mathrm{MLP}_1}(Y)\right)_{\max} \right\|$$

$$\leq \mathrm{Lip}(f_{\mathrm{MLP}_2})d_M\left( f_{\mathrm{MLP}_1}(X) - f_{\mathrm{MLP}_1}(Y) \right) \text{ (due to Lemma 3.2)}$$

$$\leq \mathrm{Lip}(f_{\mathrm{MLP}_2}) \sum_{i=1}^{M} \left\| f_{\mathrm{MLP}_1}(\mathbf{v}_{\pi^*(i)}) - f_{\mathrm{MLP}_1}(\mathbf{u}_i) \right\|$$

$$\leq \mathrm{Lip}(f_{\mathrm{MLP}_2}) \sum_{i=1}^{M} \mathrm{Lip}(f_{\mathrm{MLP}_1})\|\mathbf{v}_{\pi^*(i)} - \mathbf{u}_i\|$$

$$\leq \mathrm{Lip}(f_{\mathrm{MLP}_1})\mathrm{Lip}(f_{\mathrm{MLP}_2}) \sum_{i=1}^{M} \|\mathbf{v}_{\pi^*(i)} - \mathbf{u}_i\|$$

$$= \mathrm{Lip}(f_{\mathrm{MLP}_1})\mathrm{Lip}(f_{\mathrm{MLP}_2})d_M(X,Y)$$

We conclude that the $\mathrm{NN}_{\mathrm{MAX}}$ model is Lipschitz continuous with respect to the matching distance and its Lipschitz constant is upper bounded by $\mathrm{Lip}(f_{\mathrm{MLP}_1})\mathrm{Lip}(f_{\mathrm{MLP}_2})$.

### B.8.5 THE NN$_{\text{MAX}}$ MODEL IS LIPSCHITZ CONTINUOUS WITH RESPECT TO EMD

Let $\mathcal{X}$ denote a set that contains multisets of vectors of equal cardinalities, i.e., $|X| = M$ and $X \in \mathcal{S}(\mathbb{R}^d)$, $\forall X \in \mathcal{X}$ where $M \in \mathbb{N}$. Let $X, Y \in \mathcal{X}$ denote two multisets. We have shown in subsection B.8.4 above that:

$$\left\| f_{\text{MLP}_2}\left( \left( f_{\text{MLP}_1}(\mathbf{v}_i) \right)_{\max} \right) - f_{\text{MLP}_2}\left( \left( f_{\text{MLP}_1}(\mathbf{u}_j) \right)_{\max} \right) \right\| \leq \text{Lip}(f_{\text{MLP}_1})\text{Lip}(f_{\text{MLP}_2})d_M(X, Y)$$

By Proposition 2.3, we have that $d_M(X, Y) = M d_{\text{EMD}}(X, Y)$. Therefore, we have that:

$$\left\| f_{\text{MLP}_2}\left( \left( f_{\text{MLP}_1}(\mathbf{v}_i) \right)_{\max} \right) - f_{\text{MLP}_2}\left( \left( f_{\text{MLP}_1}(\mathbf{u}_j) \right)_{\max} \right) \right\| \leq \text{Lip}(f_{\text{MLP}_1})\text{Lip}(f_{\text{MLP}_2})d_M(X, Y)$$
$$= M\text{Lip}(f_{\text{MLP}_1})\text{Lip}(f_{\text{MLP}_2})d_{\text{EMD}}(X, Y) \text{ (due to Proposition 2.3)}$$

We thus have that the NN$_{\text{MAX}}$ model is Lipschitz continuous with respect to EMD and its Lipschitz constant is upper bounded by $M\,\text{Lip}(f_{\text{MLP}_1})\text{Lip}(f_{\text{MLP}_2})$.

### B.9 PROOF OF PROPOSITION 3.6

(1) Let $\mathbf{F} \in \mathbb{R}^{n \times (n+1)}$ be a matrix. We set the elements of $\mathbf{F}$ equal to the following values:

$$[\mathbf{F}]_{ij} = \begin{cases} \frac{1}{n(n+1)} & \text{if } j = n+1 \\ \frac{1}{n+1} & \text{if } i = j \\ 0 & \text{otherwise} \end{cases}$$

Then, we have that:

$$[\mathbf{F}]_{ij} > 0, \qquad 1 \leq i \leq n, \quad 1 \leq j \leq n+1$$

$$\sum_{j=1}^{n+1}[\mathbf{F}]_{ij} = \frac{1}{n}, \qquad 1 \leq i \leq n$$

$$\sum_{i=1}^{n}[\mathbf{F}]_{ij} = \frac{1}{n+1}, \qquad 1 \leq j \leq n+1$$

Therefore, $\mathbf{F}$ is a feasible solution of the EMD formulation and its value is equal to:

$$\sum_{i=1}^{n}\sum_{j=1}^{n+1}[\mathbf{F}]_{ij} \left\| \mathbf{v}_i - \mathbf{v}_j \right\| = \frac{1}{n(n+1)} \sum_{i=1}^{n} \left\| \mathbf{v}_i - \mathbf{v}_{n+1} \right\|$$

We thus have that:

$$d_{\text{EMD}}(X, X') \leq \frac{1}{n(n+1)} \sum_{i=1}^{n} \left\| \mathbf{v}_i - \mathbf{v}_{n+1} \right\|$$

A simple case where the inequality holds with equality is when $\mathbf{v}_1 = \mathbf{v}_2 = \ldots = \mathbf{v}_n = \mathbf{v}_{n+1}$.

(2) The bidirectional Hausdorff distance between $X$ and $X'$ is defined as:

$$d_H(X, X') = \max\left( h(X, X'), h(X', X) \right)$$

For each $i \in [n]$, we have that $\min_{j \in [n+1]} \left\| \mathbf{v}_i - \mathbf{v}_j \right\| = \left\| \mathbf{v}_i - \mathbf{v}_i \right\| = 0$. Therefore, we have that:

$$h(X, X') = \max_{i \in [n]} \min_{j \in [n+1]} \left\| \mathbf{v}_i - \mathbf{v}_j \right\| = 0$$

We thus obtain the following:

$$d_H(X, X') = h(X', X) = \max_{i \in [n+1]} \min_{j \in [n]} \left\| \mathbf{v}_i - \mathbf{v}_j \right\|$$

For each $i \in [n]$, we have that $\min_{j \in [n]} \left\| \mathbf{v}_i - \mathbf{v}_j \right\| = \left\| \mathbf{v}_i - \mathbf{v}_i \right\| = 0$. Therefore, we have that:

$$d_H(X, X') = h(X', X) = \min_{j \in [n]} \left\| \mathbf{v}_{n+1} - \mathbf{v}_j \right\|$$

## C  STABILITY OF NEURAL NETWORKS FOR SETS UNDER PERTURBATIONS

Lemma 3.5 implies that the output variation of $NN_{MEAN}$, $NN_{SUM}$ and $NN_{MAX}$ under perturbations of the elements of an input set can be bounded via the EMD, matching distance and Hausdorff distance between the input and perturbed sets, respectively. We next investigate what are the values of EMD, matching distance and Hausdorff distance when an element of a multiset is replaced by a new element.

**Proposition C.1.** *Given a multiset of vectors $X = \{\!\{\mathbf{v}_1, \ldots, \mathbf{v}_{i-1}, \mathbf{v}_i, \mathbf{v}_{i+1}, \ldots, \mathbf{v}_n\}\!\} \in \mathcal{S}(\mathbb{R}^d)$, let $X' = \{\!\{\mathbf{v}_1, \ldots, \mathbf{v}_{i-1}, \mathbf{v}_i', \mathbf{v}_{i+1}, \ldots, \mathbf{v}_n\}\!\} \in \mathcal{S}(\mathbb{R}^d)$ be the multiset where element $\mathbf{v}_i$ is perturbed to $\mathbf{v}_i'$. Then,*

1. *The EMD and matching distance between $X$ and $X'$ are equal to:*

$$d_{EMD}(X, X') = \frac{1}{n}\|\mathbf{v}_i - \mathbf{v}_i'\| \qquad and \qquad d_M(X, X') = \|\mathbf{v}_i - \mathbf{v}_i'\|$$

2. *The Hausdorff distance between $X$ and $X'$ is bounded as:*

$$d_H(X, X') \leq \|\mathbf{v}_i - \mathbf{v}_i'\|$$

*Proof.* (1) We set $\mathbf{u}_1 = \mathbf{v}_1$, $\mathbf{u}_2 = \mathbf{v}_2$, ..., $\mathbf{u}_i = \mathbf{v}_i'$, ..., $\mathbf{u}_n = \mathbf{v}_n$. For the matching distance, we have:

$$\begin{aligned}
d_M(X, X') &= \min_{\pi \in \mathfrak{S}_n}\left[\sum_{i=1}^{n}\|\mathbf{v}_{\pi(i)} - \mathbf{u}_i\|\right] \\
&\leq \|\mathbf{v}_1 - \mathbf{v}_1\| + \|\mathbf{v}_2 - \mathbf{v}_2\| + \ldots + \|\mathbf{v}_i - \mathbf{v}_i'\| + \ldots + \|\mathbf{v}_n - \mathbf{v}_n\| \\
&= \|\mathbf{v}_i - \mathbf{v}_i'\|
\end{aligned}$$

Suppose that $\pi^* \in \mathfrak{S}_n$ is the permutation associated with $d_M(X, X')$. Then, we have:

$$\begin{aligned}
\|\mathbf{v}_i - \mathbf{v}_i'\| &= \|\mathbf{v}_1 + \ldots + \mathbf{v}_i + \ldots + \mathbf{v}_n - \mathbf{v}_1 - \ldots - \mathbf{v}_i' - \ldots - \mathbf{v}_n\| \\
&= \|\mathbf{v}_1 + \ldots + \mathbf{v}_i + \ldots + \mathbf{v}_n - \mathbf{u}_1 - \ldots - \mathbf{u}_i - \ldots - \mathbf{u}_n\| \\
&\leq \|\mathbf{v}_{\pi^*(1)} - \mathbf{u}_1\| + \ldots + \|\mathbf{v}_{\pi^*(i)} - \mathbf{u}_i\| + \ldots + \|\mathbf{v}_{\pi^*(n)} - \mathbf{u}_n\| \\
&= d_M(X, X')
\end{aligned}$$

We showed that $d_M(X, X') \leq \|\mathbf{v}_i - \mathbf{v}_i'\|$ and that $\|\mathbf{v}_i - \mathbf{v}_i'\| \leq d_M(X, X')$. Therefore, $d_M(X, X') = \|\mathbf{v}_i - \mathbf{v}_i'\|$. Since $|X| = |X'| = n$, by Proposition 2.3, we have that $d_M(X, X') = n\, d_{EMD}(X, X')$. The following then holds:

$$d_{EMD}(X, X') = \frac{1}{n}\|\mathbf{v}_i - \mathbf{v}_i'\|$$

(2) We set $\mathbf{u}_1 = \mathbf{v}_1$, $\mathbf{u}_2 = \mathbf{v}_2$, ..., $\mathbf{u}_i = \mathbf{v}_i'$, ..., $\mathbf{u}_n = \mathbf{v}_n$. The Hausdorff distance is equal to:

$$d_H(X, X') = \max\big(h(X, X'), h(X', X)\big)$$

For each $j \in [i-1] \cup \{i+1, \ldots, n\}$, we have that $\min_{k \in [n]}\|\mathbf{v}_j - \mathbf{u}_k\| = \|\mathbf{v}_j - \mathbf{v}_j\| = 0$. For each $j \in [i-1] \cup \{i+1, \ldots, n\}$, we also have that $\min_{k \in [n]}\|\mathbf{u}_j - \mathbf{v}_k\| = \|\mathbf{v}_j - \mathbf{v}_j\| = 0$. Therefore, we have that:

$$h(X, X') = \min_{j \in [n]}\|\mathbf{v}_i - \mathbf{u}_j\| \leq \|\mathbf{v}_i - \mathbf{u}_i\| = \|\mathbf{v}_i - \mathbf{v}_i'\|$$

$$h(X', X) = \min_{j \in [n]}\|\mathbf{u}_i - \mathbf{v}_j\| \leq \|\mathbf{u}_i - \mathbf{v}_i\| = \|\mathbf{v}_i - \mathbf{v}_i'\|$$

Both $h(X, X')$ and $h(X', X)$ are thus no greater than $\|\mathbf{v}_i - \mathbf{v}_i'\|$. We then have that:

$$\begin{aligned}
d_H(X, X') &= \max\big(h(X, X'), h(X', X)\big) \\
&\leq \max\big(\|\mathbf{v}_i - \mathbf{v}_i'\|, \|\mathbf{v}_i - \mathbf{v}_i'\|\big) \\
&= \|\mathbf{v}_i - \mathbf{v}_i'\|
\end{aligned}$$

$\square$

We next investigate what are the values of EMD, Hausdorff distance and matching distance when a random vector sampled from $\mathcal{U}(0, k)^d$ is added to each element of a multiset.

**Proposition C.2.** *Given a multiset of vectors* $X = \{\!\{\mathbf{v}_1, \mathbf{v}_2, \ldots, \mathbf{v}_n\}\!\} \in \mathcal{S}(\mathbb{R}^d)$, *let* $X' = \{\!\{\mathbf{v}_1 + \mathbf{u}_1, \mathbf{v}_2 + \mathbf{u}_2, \ldots, \mathbf{v}_n + \mathbf{u}_n\}\!\} \in \mathcal{S}(\mathbb{R}^d)$ *where* $\mathbf{u}_i \sim \mathcal{U}(0, k)^d$ *for all* $i \in [n]$. *Then,*

1. *The EMD and matching distance between $X$ and $X'$ is bounded as:*

$$d_{EMD}(X, X') \leq k\sqrt{d} \qquad and \qquad d_M(X, X') \leq nk\sqrt{d}$$

2. *The Hausdorff distance between $X$ and $X'$ is bounded as:*

$$d_H(X, X') \leq k\sqrt{d}$$

*Proof.* (1) Note that

$$\|\mathbf{v}_i - \mathbf{v}_i - \mathbf{u}_i\| = \|\mathbf{u}_i\| \leq \sqrt{\underbrace{k^2 + k^2 + \ldots + k^2}_{d \text{ times}}} = k\sqrt{d}$$

For the matching distance, we have:

$$
\begin{aligned}
d_M(X, X') &= \min_{\pi \in \mathfrak{S}_n} \left[ \sum_{i=1}^n \|\mathbf{v}_{\pi(i)} - \mathbf{v}_i - \mathbf{u}_i\| \right] \\
&\leq \|\mathbf{v}_1 - \mathbf{v}_1 - \mathbf{u}_1\| + \|\mathbf{v}_2 - \mathbf{v}_2 - \mathbf{u}_2\| + \ldots + \|\mathbf{v}_n - \mathbf{v}_n - \mathbf{u}_n\| \\
&= \|-\mathbf{u}_1\| + \|-\mathbf{u}_2\| + \ldots + \|-\mathbf{u}_n\| \\
&\leq \underbrace{k\sqrt{d} + k\sqrt{d} + \ldots + k\sqrt{d}}_{n \text{ times}} \\
&= nk\sqrt{d}
\end{aligned}
$$

Since $|X| = |X'| = n$, by Proposition 2.3, we have that $d_M(X, X') = n\, d_{\text{EMD}}(X, X')$. The following then holds:

$$d_{\text{EMD}}(X, X') \leq \frac{1}{n} nk\sqrt{d} = k\sqrt{d}$$

(2) The Hausdorff distance is equal to:

$$d_H(X, X') = \max\left(h(X, X'), h(X', X)\right)$$

We have that:

$$h(X, X') = \max_{i \in [n]} \min_{j \in [n]} \|\mathbf{v}_i - \mathbf{v}_j - \mathbf{u}_j\| \leq \max_{i \in [n]} \|\mathbf{v}_i - \mathbf{v}_i - \mathbf{u}_i\| = \max_{i \in [n]} \|-\mathbf{u}_i\| \leq k\sqrt{d}$$

$$h(X', X) = \max_{i \in [n]} \min_{j \in [n]} \|\mathbf{v}_i + \mathbf{u}_i - \mathbf{v}_j\| \leq \max_{i \in [n]} \|\mathbf{v}_i + \mathbf{u}_i - \mathbf{v}_i\| = \max_{i \in [n]} \|\mathbf{u}_i\| \leq k\sqrt{d}$$

Both $h(X, X')$ and $h(X', X)$ are thus no greater than $k\sqrt{d}$. We then have that:

$$
\begin{aligned}
d_H(X, X') &= \max\left(h(X, X'), h(X', X)\right) \\
&\leq \max\left(k\sqrt{d}, k\sqrt{d}\right) \\
&= k\sqrt{d}
\end{aligned}
$$

$\square$

# D  EXPERIMENTAL SETUP

We next provide details about the experimental setup in the different experiments we conducted.

## D.1  LIPSCHITZ CONSTANT OF AGGREGATION FUNCTIONS

**ModelNet40.**  We produce point clouds consisting of 100 particles ($x, y, z$-coordinates) from the mesh representation of objects. Each set is normalized by the initial layer of the deep network to have zero mean (along individual axes) and unit (global) variance. Each neural network consists of an MLP, followed by an aggregation function (i. e., SUM, MEAN or MAX) which in turn is followed by another MLP. The first MLP transforms the representations of the particles, while the aggregation function produces a single vector representation for each point cloud. The two MLPs consist of two layers and the ReLU function is applied to the output of the first layer. The output of the second layer of the second MLP is followed by the softmax function which outputs class probabilities. The hidden dimension size of all layers is set to 64. The model is trained by minimizing the cross-entropy loss. The minimization is performed using Adam with a learning rate equal to 0.001. The number of epochs is set to 200 and the batch size to 64. At the end of each epoch, we compute the performance of the model on the validation set, and we choose as our final model the one that achieved the smallest loss on the validation set. Note that in this set of experiments we do not compute the EMD, Hausdorff and matching distance between the input multisets, but we compute the distance of the "latent" multisets that emerge at the output of the first MLP (just before the aggregation function is applied) of each chosen model.

**Polarity.**  We represent each document as a multiset of the embeddings of its words. The word embeddings are obtained from a pre-trained model which contains 300-dimensional vectors (Mikolov et al., 2013). The embeddings are first fed to an MLP, and then an aggregation function (i. e., SUM, MEAN or MAX) is applied which produces a single vector representation for each document. This vector representation is passed onto another MLP. The two MLPs consist of two layers and the ReLU function is applied to the output of the first layer of each MLP. The output of the second MLP is followed by the softmax function. The hidden dimension size of all layers is set to 64. The model is trained by minimizing the cross-entropy loss. The minimization is performed using Adam with a learning rate equal to 0.001. The number of epochs is set to 200 and the batch size to 64. At the end of each epoch, we compute the performance of the model on the validation set, and we choose as our final model the one that achieved the smallest loss on the validation set. As discussed above, in each experiment, we compute the distance of the "latent" multisets that emerge at the output of the first MLP (just before the aggregation function is applied) of the model that achieves the lowest validation loss.

## D.2  LIPSCHITZ CONSTANT OF NEURAL NETWORKS FOR SETS

**ModelNet40.**  We produce point clouds with 100 particles ($x, y, z$-coordinates) from the mesh representation of objects. The data points of the point clouds are first fed to a fully-connected layer. Then, the data points of each point clouds are aggregated (i. e., SUM, MEAN or MAX function is utilized) and this results into a single vector representation for each point cloud. This vector representation is then fed to another fully-connected layer. The output of this layer passes through the ReLU function and is finally fed to a fully-connected layer which is followed by the softmax function and produces class probabilities. The output dimension of the first two fully-connected layers is set to 64. The model is trained by minimizing the cross-entropy loss. The Adam optimizer is employed with a learning rate of 0.001. The number of epochs is set to 200 and the batch size to 64. At the end of each epoch, we compute the performance of the model on the validation set, and we choose as our final model the one that achieved the smallest loss on the validation set. Note that the Lipschitz constant of an affine function $f \colon \mathbf{v} \mapsto \mathbf{W}\mathbf{v} + \mathbf{b}$ where $\mathbf{W} \in \mathbb{R}^{m \times n}$ and $\mathbf{b} \in \mathbb{R}^m$ is the largest singular value of matrix $\mathbf{W}$. Therefore, in this experiment we can exactly compute the Lipschitz constants of the two fully-connected layers.

**Polarity.**  We represent each document as a multiset of the embeddings of its words. The word embeddings are obtained from a publicly available pre-trained model (Mikolov et al., 2013). We randomly split the dataset into training, validation, and test sets with a $80 : 10 : 10$ split ratio. The

embeddings are first fed to a fully-connected layer, and then an aggregation function (i. e., SUM, MEAN or MAX) is applied which produces a single vector representation for each document. This vector representation is passed onto another fully-connected layer. The ReLU function is applied to the emerging vector and then a final fully-connected layer followed by the softmax function outputs class probabilities. The output dimension of the first two fully-connected layers is set to $64$. The model is trained by minimizing the cross-entropy loss. The minimization is performed using Adam with a learning rate equal to $0.001$. The number of epochs is set to $200$ and the batch size to $64$. At the end of each epoch, we compute the performance of the model on the validation set, and we choose as our final model the one that achieved the smallest loss on the validation set. As discussed above, we can exactly compute the Lipschitz constants of the two fully-connected layers.

### D.3 STABILITY UNDER PERTURBATIONS OF INPUT MULTISETS

**ModelNet40.** We produce point clouds with $100$ particles ($x, y, z$-coordinates) from the mesh representation of objects. The $\text{NN}_{\text{MEAN}}$ and $\text{NN}_{\text{MAX}}$ models consist of an MLP which transforms the representations of the particles, an aggregation function (MEAN and MAX, respectively) and a second MLP which produces the output (i. e., class probabilities). Both MLPs consist of two hidden layers. The ReLU function is applied to the outputs of the first layer. The hidden dimension size is set to $64$ for all hidden layers. The model is trained by minimizing the cross-entropy loss. The Adam optimizer is employed with a learning rate of $0.001$. The number of epochs is set to $200$ and the batch size to $64$. At the end of each epoch, we compute the performance of the model on the validation set, and we choose as our final model the one that achieved the smallest loss on the validation set.

**Polarity.** Each document of the Polarity dataset is represented as a multiset of word vectors. The word vectors are obtained from a publicly available pre-trained model (Mikolov et al., 2013). We randomly split the dataset into training, validation, and test sets with a $80 : 10 : 10$ split ratio. The $\text{NN}_{\text{MEAN}}$ and $\text{NN}_{\text{MAX}}$ models consist of an MLP which transforms the representations of the words, an aggregation function (MEAN and MAX, respectively) and a second MLP which produces the output. Both MLPs consist of two hidden layers. The ReLU function is applied to the outputs of the first layer. The hidden dimension size is set to $64$ for all hidden layers. The model is trained for $20$ epochs by minimizing the cross-entropy loss function with the Adam optimizer and a learning rate of $0.001$. At the end of each epoch, we compute the performance of the model on the validation set, and we choose as our final model the one that achieved the smallest loss on the validation set.

### D.4 GENERALIZATION UNDER DISTRIBUTION SHIFTS

This set of experiments is conducted on the Polarity dataset. The architecture of $\text{NN}_{\text{MEAN}}$ and $\text{NN}_{\text{MAX}}$, and the training details are same as in subsection D.3 above.

## E ADDITIONAL RESULTS

### E.1 LIPSCHITZ CONSTANT OF AGGREGATION FUNCTIONS

We next provide some additional empirical results that validate the findings of Theorem 3.1 (i. e., Lipschitz constants of aggregation functions). We experiment with the Polarity dataset. Figure 4 visualizes the relationship between the output of the three considered distance functions and the Euclidean distance of the aggregated representations of multisets of documents from the Polarity dataset. Since the Polarity dataset consists of documents which might differ from each other in the number of terms, by Theorem 3.1 we can derive upper bounds only for 3 out of the 9 combinations of distance functions for multisets and aggregation functions. As expected, the Lipschitz bounds (dash lines) upper bound the Euclidean distance of the outputs of the aggregation functions. With regards to the tightness of the bounds, we observe that the bounds that are associated with the MEAN and SUM functions are tighter than the one associated with the MAX function. The correlations between the distances of multisets and the Euclidean distances of the aggregated representations of multisets are relatively low in most of the cases. All three distance functions are mostly correlated with the aggregation functions with which they are related via Theorem 3.1. The highest correlation is achieved between EMD and the MEAN function ($r = 0.89$).

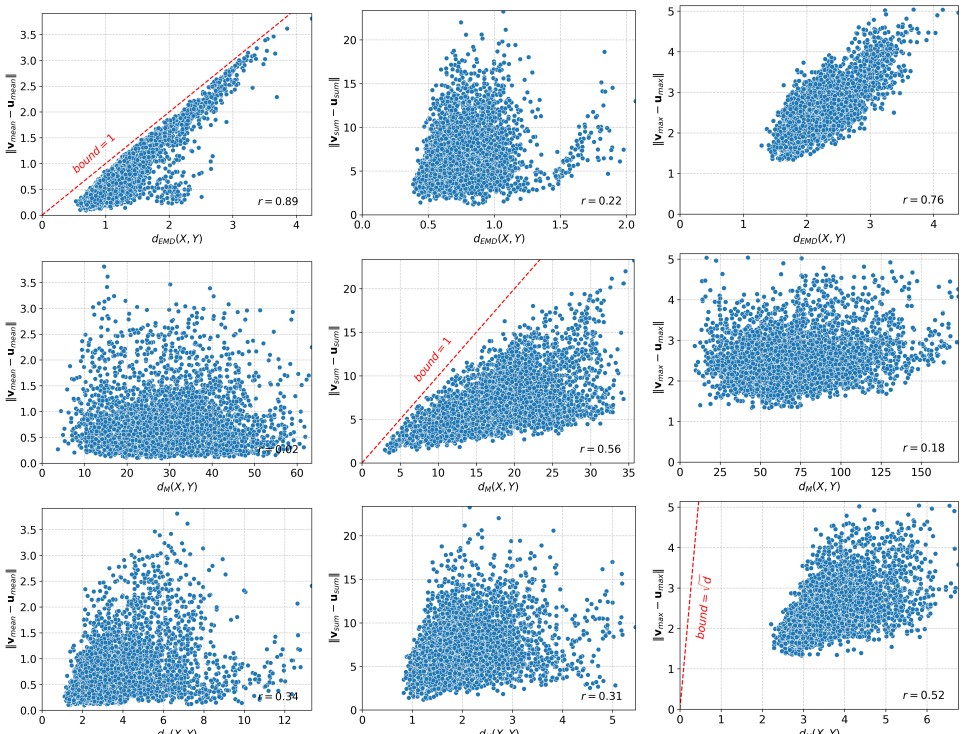

Figure 4: Each dot corresponds to a pair of documents from the test set of Polarity that is represented as a multiset of word vectors. Each subfigure compares the distance between the latent representations of pairs of documents computed by a distance function for multisets (i.e., EMD, Hausdorff distance or matching distance) with the Euclidean distance between the representations of the pairs obtained after applying an aggregation function (i.e., MEAN, SUM or MAX). The correlation between the two distances is also computed and visualized. The Lipschitz bounds are illustrated with dashed lines.

### E.2    UPPER BOUNDS OF LIPSCHITZ CONSTANTS OF NEURAL NETWORKS FOR SETS

We next provide some additional empirical results that validate the findings of Theorem 3.4 (i.e., upper bounds of Lipschitz constants of neural networks for sets). The results are shown in Figure 5. Since the Polarity dataset consists of documents which might differ from each other in the number of terms, by Theorem 3.4 we can derive upper bounds only for 2 out of the 9 combinations of distance functions for multisets and aggregation functions. While the dash lines upper bound the Euclidean distance of the outputs of the aggregation functions, both bounds are relatively loose on the Polarity dataset. The correlations between the distances of the representations produced by the neural network for the multisets and the distances produced by distance functions for multisets are much lower than those of the previous experiments. The highest correlation is equal to $0.55$ (between the neural network that utilizes the MEAN aggregation function and EMD), while there are even negative correlations. This is not surprising since for most of the combinations, there are no upper bounds on the Lipschitz constant of the corresponding neural networks.

We also visualize the relationship between the output of the three considered distance functions and the Euclidean distance of the multiset representations that are produced by a neural network that utilizes the attention mechanism that is presented in subsection 3.1. Specifically, the neural network is identical to $\text{NN}_{\text{MEAN}}$, $\text{NN}_{\text{SUM}}$ and $\text{NN}_{\text{MAX}}$, but instead of the standard aggregation functions, it employs the aforementioned attention mechanism. The experiments are conducted on the ModelNet40 dataset and the results are shown in Figure 6. We have shown that the attention mechanism is not Lipschitz continuous with respect to any of the three considered distance functions, and therefore the neural network models that employ this mechanism are also not Lipschitz continuous. We observe in Figure 6 that the correlations are indeed much lower than those illustrated in Figure 2 which confirms our theoretical result.

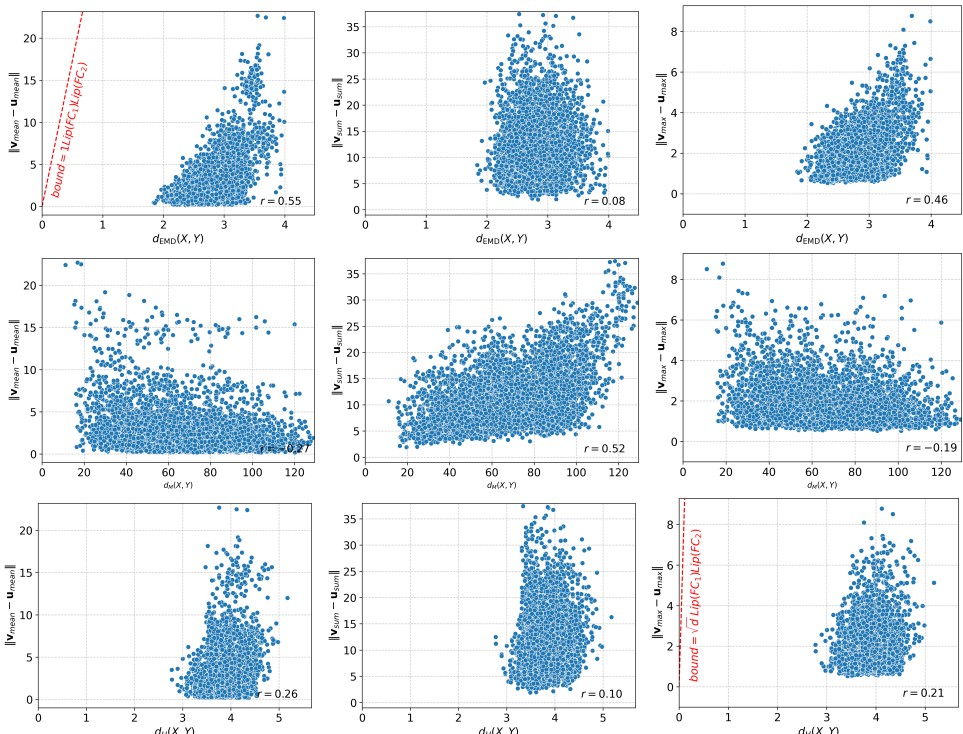

Figure 5: Each dot corresponds to a pair of documents from the test set of Polarity that is represented as a multiset of word vectors. Each subfigure compares the distance between the pairs of documents computed by a distance function for multisets (i. e., EMD, Hausdorff distance or matching distance) with the Euclidean distance between the representations of the pairs that emerge at the second-to-last layer of $\text{NN}_{\text{MEAN}}$, $\text{NN}_{\text{SUM}}$ or $\text{NN}_{\text{MAX}}$. The correlation between the two distances is also computed and visualized. The Lipschitz bounds are illustrated with dashed lines.

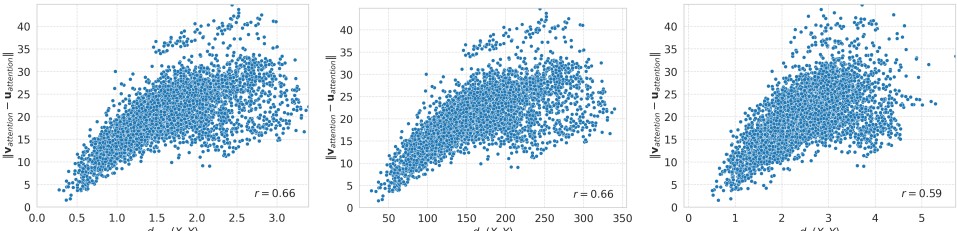

Figure 6: Each dot corresponds to a pair of point clouds from the test set of ModelNet40. Each subfigure compares the distance between the pairs of point clouds computed by EMD, Hausdorff distance or matching distance with the Euclidean distance between the representations of the pairs that emerge at the second-to-last layer of a neural network that consists of a fully-connected layer, an attention mechanism that aggregates the representations of the elements and two more fully-connected layers. The correlation between the two distances is also computed and visualized.

### E.3    STABILITY OF NEURAL NETWORKS FOR SETS UNDER PERTURBATIONS

Here we provide some further details about the experiments presented in subsection 4.3. Specifically, for the experiments conducted on ModelNet40, we attribute the drop in performance of $\text{NN}_{\text{MAX}}$ to the large Hausdorff distances between each test sample and its perturbed version. For each test sample $X_i$ (where $i \in [2468]$), let $X_i'$ denote the multiset that emerges from the application of Pert. #1 to $X_i$. Let also $y_i$ denote the class label of sample $X_i$. We compute the Hausdorff distance between $X_i$ and $X_i'$ (i. e., $d_H(X_i, X_i')$). We then compute the average Hausdorff distance between $X_i$ and the rest of the test samples that belong to the same class as $X_i$. Let $\mathcal{S}_i = \{X_j : j \in [2468], y_i = y_j\}$

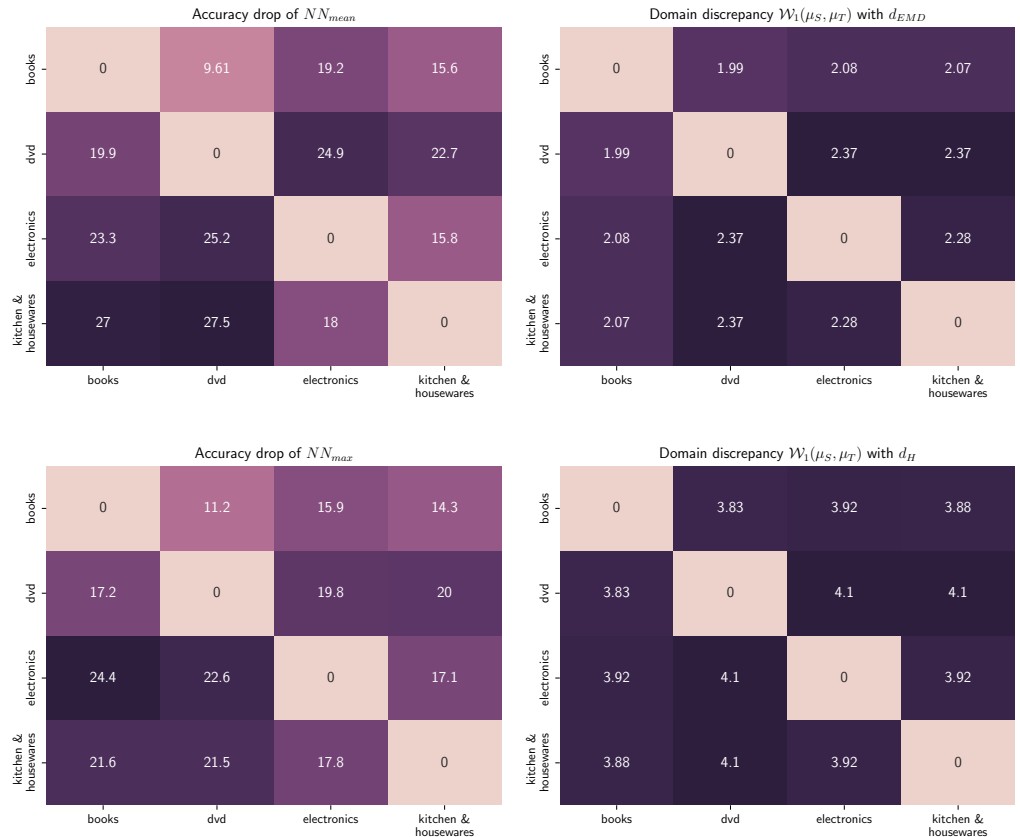

Figure 7: Accuracy drop of the $NN_{\text{MEAN}}$ and $NN_{\text{MAX}}$ models, and Wasserstein distance with $p = 1$ between groups when the EMD and the Hausdorff distance are used as ground metrics.

denote the set of all test samples that belong to the same class as $X_i$. Then, we compute $\bar{d}_H(X_i, \mathcal{S}_i)$ as follows:

$$\bar{d}_H(X_i, \mathcal{S}_i) = \frac{1}{|\mathcal{S}_i| - 1} \sum_{Y \in \mathcal{S}_i \setminus \{X_i\}} d_H(X_i, Y)$$

We then compute $d_H(X_i, X_i') - \bar{d}_H(X_i, \mathcal{S}_i)$. If this value is positive, then the distance from $X_i$ to $X_i'$ is greater than the average distance of $X_i$ to the other multisets that belong to the same class as $X_i$. We calculated this value for all test samples and then computed the average value which was found to be $2.63(\pm 1.10)$. In general, the perturbation increases the upper bound of the Lipschitz constant of $NN_{\text{MAX}}$ compared to the upper bound for samples that belong to the same class, and thus $X_i$ and $X_i'$ might end up having dissimilar representations. On the other hand, for EMD, the average distance is equal to $-1.18(\pm 0.72)$. For EMD the upper bound is in general tighter than the bound for pairs of multisets that belong to the same class, and this explains why $NN_{\text{MEAN}}$ is robust to Pert. #1.

### E.4 GENERALIZATION UNDER DISTRIBUTION SHIFTS

To evaluate the generalization of the two Lipschitz continuous models ($NN_{\text{MEAN}}$ and $NN_{\text{MAX}}$) under distribution shifts, we also experiment with the Amazon review dataset (Blitzer et al., 2007). The dataset consists of product reviews from Amazon for four different types of products (domains), namely books, DVDs, electronics and kitchen appliances. For each domain, there exist $2,000$ labeled reviews (positive or negative) and the classes are balanced. We construct $4$ adaptation tasks. In each task, the $NN_{\text{MEAN}}$ and $NN_{\text{MAX}}$ models are trained on reviews for a single type of products and evaluated on all domains.

Table 3: Average performance (accuracy or root mean square error) of the $NN_{SUM}$, $NN_{MEAN}$ and $NN_{MAX}$ on the four benchmark datasets.

|  | MODELNET40 ($\uparrow$) | POLARITY ($\uparrow$) | IMDB ($\downarrow$) | IMDB-BINARY ($\uparrow$) |
|---|---|---|---|---|
| $NN_{SUM}$ | $60.07 \pm 2.12$ | $76.93 \pm 2.42$ | $0.2210 \pm 0.0085$ | $70.20 \pm 1.72$ |
| $NN_{MEAN}$ | $63.41 \pm 0.98$ | $77.11 \pm 2.34$ | $0.2159 \pm 0.0033$ | $67.00 \pm 2.52$ |
| $NN_{MAX}$ | $77.21 \pm 1.09$ | $78.14 \pm 1.94$ | $0.2259 \pm 0.0017$ | $61.40 \pm 2.57$ |

Each review is represented as a multiset of word vectors. The word vectors are obtained from a publicly available pre-trained model (Mikolov et al., 2013). The $NN_{MEAN}$ and $NN_{MAX}$ models consist of an MLP which transforms the representations of the words, an aggregation function (MEAN and MAX, respectively) and a second MLP which produces the output. Both MLPs consist of two hidden layers. The ReLU function is applied to the outputs of the first layer and also dropout is applied between the two layers with $p = 0.2$. The hidden dimension size is set to $64$ for all hidden layers. The model is trained for $50$ epochs by minimizing the cross-entropy loss function with the Adam optimizer and a learning rate of $0.001$. At the end of each epoch, we compute the performance of the model on the validation set, and we choose as our final model the one that achieved the smallest loss on the validation set.

Figure 7 illustrates the Wasserstein distance with $p = 1$ between groups when the EMD (Top Right) and the Hausdorff distance (Bottom Right) are used as ground metrics. It also shows the drop in accuracy when the $NN_{MEAN}$ (Top Left) and $NN_{MAX}$ (Bottom Left) models are trained on one domain and evaluated on the others. Each row corresponds to one specific model, e. g., the first row represents the model trained on the reviews for books. We observe that in general the drop in accuracy follows a similar pattern with the distance between the source and target domains, i. e., the higher the distance, the higher the drop in accuracy. We also computed the Pearson correlation between the drop in accuracy and the domain dicrepancies. We found that the Wasserstein distance based on EMD highly correlates with the accuracy drop of $NN_{MEAN}$ ($r = 0.917$), while there is an even higher correlation between the Wasserstein distance based on Hausdorff distance and the accuracy drop of $NN_{MAX}$ ($r = 0.941$).

### E.5 PREDICTIVE PERFORMANCE OF NEURAL NETWORKS FOR SETS

We next evaluate the $NN_{SUM}$, $NN_{MEAN}$ and $NN_{MAX}$ models on four classification and regression datasets, namely ModelNet40, Polarity, IMDB and IMDB-BINARY. The first two datasets are described in section 4. IMDB contains movie reviews from the IMDb database (Maas et al., 2011). The targets are the ratings that accompany the reviews (10 different values). We treat this task as a regression problem. IMDB-BINARY is a standard graph classification dataset (Yanardag & Vishwanathan, 2015), commonly used for evaluating graph kernels and graph neural networks. Each graph of the IMDB-BINARY dataset was represented as a multiset of the degrees of its nodes. The results are illustrated in Table 3. Each experiment was repeated 5 times with different random seeds, and for each dataset we report average accuracy (for ModelNet40, Polarity and IMDB-BINARY) or average root mean square error (for IMDB) on the dataset's test set and the corresponding standard deviation.

We can see that $NN_{MAX}$ outperforms the other models on ModelNet40. A possible explanation is that all input multisets have the same size, and in such a setting the max aggregator is Lipschitz continuous with respect to all three considered distance functions. Therefore, $NN_{MAX}$ can effectively capture the distances between the point clouds in ModelNet40.

Polarity consists of short reviews (average number of terms = 20). Therefore, whether a review is positive or negative depends primarily on the presence of one or a few terms that indicate sentiment. These terms can be considered extreme elements, and the Hausdorff distance relies on such extreme elements when comparing its inputs. This distance function thus seems to be suitable for this task and potentially explains why $NN_{MAX}$ is the best-performing model on this dataset.

The reviews contained in the IMDB dataset are much longer than those in the Polarity dataset (average number of terms = 254). Due to the potential presence of outlier terms, the Hausdorff distance

may not accurately capture document similarity. The matching distance can also be sensitive to document length. In contrast, EMD captures the overall semantic alignment between documents, and compares documents based on their overall meaning. This makes EMD more suitable for this task, which is empirically confirmed by the superior performance of $\text{NN}_{\text{MEAN}}$ compared to the other two models.

$\text{NN}_{\text{SUM}}$ is the best-performing method on the IMDB-BINARY dataset. In this dataset, capturing both the number of nodes and their degrees is essential. The matching distance is well suited for this task, and the stronger performance of the $\text{NN}_{\text{SUM}}$ model, which is Lipschitz continuous with respect to this distance (under certain conditions), supports this intuition.

