# OpenReview forum: "On the Lipschitz Continuity of Set Aggregation Functions and Neural Networks for Sets"
_ICLR.cc/2026/Conference — ICLR 2026 Poster_

### Official Review · Reviewer_AMSr · 2025-10-20

**Soundness:** 4
**Presentation:** 3
**Contribution:** 3
**Rating:** 8
**Confidence:** 5

**Summary:**

This paper discusses the Lipschitz properties of aggregation functions for multisets. The main results is that each of the three common aggregations: (a) mean (b) sum and (c) max are Lipschitz with respect to only one of the three multiset metrics

mean--> EMD

max--> Hausdorff

sum--> matching

The explicit constants are also computed. In addition, results for attention based aggregations are provided, implications for perturbations and distribution shifts are discussed and empirical evidence for the theorems is provided.

**Strengths:**

The paper is well written, and from what I checked the theoretical results seem correct. The main result should be of some impact to the 'permutation invariant learning' community. Perhaps, the take home message is this: while from the lens of pure injectivity, the well known xu et al. paper argued that sum>mean>max, from the lens of Lipschitzness, each has its own distinct advantage.

**Weaknesses:**

While the paper does conduct some nice experiments, there is no clear bottom line to the paper from a practical point of view. E.g., one could hope this paper would substantiate a claim that for some type of tasks, sum pooling is better, and for others, max pooling is more accurate, or more robust.

**Questions:**

Honestly, these are not questions but suggestions for the next version of the paper (hopefully, the camera ready version). Feel free to ignore some or all of these points for the rebuttal. But please do take this into account when editing the next version of the paper.

**Bigger comments*
* EMD and the Hausdorff distance are not a metric on multisets of $\leq M$ elements because for X={a,a}, Y={a}  we have EMD(X,Y)=0=Hausdorff(X,Y). This should be mentioned when these metrics are introduced.
*  Related to this: some of your results follow from the arguments in xu et al., which are nicely summarized in figures 2-3 and in the schematic statement sum>mean>max. Mamely, it is established that mean cannot separate certain multisets (e.g., X and Y above) if they are equal as distributions, and that max cannot separate multisets which mean can, namely multisets which are different as distributions but are the same if multiplicities are ignored. It would be good to explain this, and this would also emphasize that the other direction, namely that sum is not Lipschitz with respect to the other metrics, is not known and is a truly new result.
* It would be good to add references to support your choice of distances on multisets. Namely, to show these choices are natural and not something you invented. For example, the matching metric, perhaps the least known of the three, is used in the Chuang paper you cited in a different context, and in [1].
* It seems to me that an easy corollary of Theorem 3.1 is that none of these three metrics are bi-Lipschitz equivalence. Thus, no aggregation can be simultaneously Lipschtiz with respect to all of them
* In Lemma 3.2, why don't you discuss the Hausdorff metric?
* Bullets 1-2 of Lemma 3.2 follow immediately from proposition 3.2, I would comment on that.
* I am missing some discussion of Theorem 3.3. Namely, what you get for mean and max is more or less the extension from theorem 3.1 which you might expect. For sum pooling you don't get this, which initially seems like a surprise. Why is this? After looking in the proof my understanding is that this is because the inner function can map non-zero elements to zero (in your proof, near zero). This too should be explained. This also seems a bit superficial to me: I would suggest to define the matching metric so that if X has $n_1$ elements and Y has $n_2$ elements, then the term $|n_2-n_1|$ would be added for the unmatched elements. Wouldn't this solve the issue?
* What do we learn from Proposition 3.6? Could you add some discussion?
* Section 3.4: I understand that Lipschitz results automatically lead to theoretical results on generalizaiton, still, I couldn't understand this section I think it should be rewritten.
* Line 459-461: I think when you write "insensitive" you actually mean "sensitive". In my understanding insesnsitive=robust.



**Minor comments**
* In line 51, the sentence "input data might correspond..." is strange. I would say, sometimes, data is permutation invariant and should be treated as a multiset.
* Line 119, for the sake of formal correctness, you might want to mention that sum pooling in injective when coupled with a  function applied to  each point.
* Proposition 2.3 seems obvious to me, if you agree I would add some sort of remark about that
* Theorem 3.3 starts with "Let X,Y" but X and Y do not appear elsewhere in the theorem.
* Line 404: "Therefore, the results of both Theorem 3.1 and Lemma 3.2 hold". I would rephrase, the experiment corraborates the theorem, but it does not prove it is true. What proves the theoremsare true, is, well, the proofs of the theorems.
* Line 418-420: You discuss Lipschitz constants of the affine layer, what about the activation?
* A.3.1: In the proof, say that this if for the F for which the EMD is realized.
*Line 771: "Then we have" doesn't make sense, should be "where we choose" or something like that

[1] On the Holder Stability Of Multiset and Graph Neural Network, Davidson and Dym, ICLR 2025

---

> ### Author Response · Authors · 2025-11-20
> **Response to Reviewer AMSr**
>
> We thank the reviewer for the positive assessment and the very detailed review. Below, we provide detailed responses to the major points raised.
>
> > While the paper does conduct some nice experiments, there is no clear bottom line to the paper from a practical point of view.
>
> Note that the different distance functions compare the input multisets according to different criteria, highlighting **different forms of variation**. As a general guideline, one should choose the function that is Lipschitz continuous with respect to the distance function that **best captures** the distances between the multisets in the considered dataset or problem. For example, in problems where the shape of the input object matters (e.g., shapes extracted from medical images or 3D scans), Hausdorff distance is preferable to EMD and the matching distance since we would like to detect whether any part of one shape is far away from the other shape, even if the rest of the shapes are well-aligned. On the other hand, EMD is better suited than the Hausdorff distance for text embedding tasks since it captures the overall semantic alignment between texts. That said, it is clear that choosing an aggregation function typically requires some domain knowledge. If such knowledge is not available, choosing an aggregation function is difficult, except in special cases, for example when all multisets have the same cardinality, in which case the max function is Lipschitz continuous for all distance functions. In short, choosing a model that is Lipschitz continuous with respect to a distance function that aligns well with the task at hand can lead to improved robustness and generalization performance, and practitioners should consider these aspects when developing models for their applications.
>
> > EMD and the Hausdorff distance are not a metric on $\leq M$ multisets of elements because for X={a,a}, Y={a} we have EMD(X,Y)=0=Hausdorff(X,Y). This should be mentioned when these metrics are introduced.
>
> We appreciate the reviewer bringing this to our attention. While these distances are metrics for sets, they are pseudometrics for multisets. We will fix this in the revised version (to be uploaded by December 3).
>
> > It would be good to add references to support your choice of distances on multisets. Namely, to show these choices are natural and not something you invented...
>
> EMD and Hausdorff distance are both standard distance functions for sets and multisets and have been widely used in various problem settings in the past. We will clarify this point in the revised manuscript and add some references as suggested by the reviewer. The reviewer is also correct that the matching distance is used in [1], where it is described as the unnormalized version of EMD. We also thank the reviewer for pointing us to [2], where this distance is also introduced. The definition in both papers is slightly different, as the authors assume that the input multisets have the same number of elements. For multisets of unequal cardinalities, they propose augmenting the smaller multiset by repeatedly adding a zero vector until its size matches that of the larger multiset. Importantly, the resulting distance in both cases remains the same.
>
> > It seems to me that an easy corollary of Theorem 3.1 is that none of these three metrics are bi-Lipschitz equivalence. Thus, no aggregation can be simultaneously Lipschtiz with respect to all of them
>
> Indeed, if two metrics $d_1$ and $d_2$ on a set $X$ are bi-Lipschitz equivalent, then a function $f \colon X \rightarrow Y$ is Lipschitz continuous with respect to $d_1$ if and only if it is Lipschitz continuous with respect to $d_2$. In other words, bi-Lipschitz equivalent metrics induce exactly the same class of Lipschitz functions. For the metrics under consideration, this property fails: for each pair, there exists a function that is Lipschitz with respect to one metric but not with respect to the other. Therefore, no pair of these metrics can be bi-Lipschitz equivalent. We thank the reviewer for pointing out this interesting result.
>
> > In Lemma 3.2, why don't you discuss the Hausdorff metric?
>
> This is a valid point. Neither the sum aggregator nor the mean aggregator is Lipschitz continuous with respect to the Hausdorff distance, even when they are restricted to input multisets of the same cardinality. While we had originally derived these results, due to an oversight they were not included in the initial submission. We will provide the two proofs in the revised manuscript (to be uploaded by December 3).
>
> > Bullets 1-2 of Lemma 3.2 follow immediately from proposition 3.2, I would comment on that.
>
> Indeed, in the proof of Lemma 3.2 (Appendix A.4), we use Proposition 2.3 to establish bullets 1 and 2. We will clarify this more explicitly in the revised manuscript.

---

> ### Author Response · Authors · 2025-11-20
> **Response to Reviewer AMSr**
>
> > I am missing some discussion of Theorem 3.3. Namely, what you get for mean and max is more or less the extension from theorem 3.1 which you might expect. For sum pooling you don't get this, which initially seems like a surprise...
>
> We thank the reviewer for this insightful comment. This actually happens **because of the bias** of the fully-connected layer (or MLP) that updates the multisets' elements. We next provide an example to make this clear. Let $X=${ $x_1,x_2$ } and $Y=${ $x_1$ } be two sets where $x_1,x_2 \in \mathbb{R}$ and $x_1,x_2 > 0$. The matching distance between the two sets is $d_M(X,Y) = \| x_2 \|$. Suppose that the fully-connected layer that updates the elements is $f_1(x) = a x + b$ where $a,b >0$, while the second fully-connected layer is the identity function, i.e., $f_2(x) = x$. Then, the distance of the representations of $X$ and $Y$ produced by $NN_{SUM}$ are equal to:
>
> $\| NN_{SUM}(X) - NN_{SUM}(Y) \| = \| f_2( f_1(x_1) + f_1(x_2) ) -  f_2( f_1(x_1) ) \| = \| f_1(x_1) + f_1(x_2) - f_1(x_1) \| = \| f_1(x_2) \| = \| a x_2 + b \|$
>
> Note that $a$ is $Lip(f_1)$. By increasing $b$, we can make $\| a x_2 + b \|$ **arbitrarily large**, regardless of the values of $Lip(f_1)$ and $\| x_2 \|$ are equal to.
>
> The reviewer's suggestion (adding the term $|n_2 - n_1|$ for the unmatched elements) would not solve this issue. The issue can be resolved by **omitting the bias** in the fully-connected layer (or MLP) that updates the multiset elements. Note also that even if we do not omit the bias, $NN_{SUM}$ is still Lipschitz continuous with respect to the matching distance if we assume that both the bias $b$ and the cardinality of the input multisets are bounded.
>
> > What do we learn from Proposition 3.6? Could you add some discussion?
>
> This Proposition provides the exact value of the Hausdorff distance along with an upper bound for the EMD between an input multiset and the same multiset with one additional element. The Hausdorff distance between the original multiset and the new multiset is equal to the minimum distance between the added element and the elements of the original multiset. The EMD, in turn, is upper bounded by the normalized sum of the distances between all elements of the original multiset and the newly added element. The normalization factor is $1/(n(n+1))$. Equivalently, this is the mean divided by $n+1$. If the newly added element **lies close to an existing element**, the Hausdorff distance will be small, whereas the upper bound on EMD can still be large (consider the case where the remaining elements of the multiset are very far from the new element). In such a setting, $NN_{MAX}$, which is Lipschitz continuous with respect to the Hausdorff distance, is likely to be very stable under the addition of the new element and therefore to produce multiset representations that remain very close to each other. In contrast, $NN_{MEAN}$ could be less stable, since EMD may take large values in this case. On the other hand, if the newly added element is an **outlier far from all existing elements**, the Hausdorff distance can become large, while the upper bound on EMD may remain relatively small due to the normalization factor (especially if the cardinality of the multiset is large). In this case, $NN_{MAX}$ is likely to be less stable than $NN_{MEAN}$. Overall, the stability of the different models **depends on the added element**. Due to space constraints, we were unable to include extensive discussion in the original submission. Note also that, in the case of acceptance, we will make use of the additional page available in the camera-ready version to improve the organization and clarity of the paper.

---

> ### Author Response · Authors · 2025-11-20
> **Response to Reviewer AMSr**
>
> > Section 3.4: I understand that Lipschitz results automatically lead to theoretical results on generalizaiton, still, I couldn't understand this section I think it should be rewritten.
>
> The main idea is that we consider two data distributions: $\mu_S$ is the original (source) distribution from which the model is trained, and $\mu_T$ is the new (target) distribution that the model will encounter at deployment. Typically, these two distributions are different from each other. Theorem 3.7 provides a bound on the expected error when evaluating the model on data sampled from the target distribution. The bound has the following form:
>
> $\epsilon_T(h) \leq \epsilon_S(h) + 2  L  W_1(\mu_S, \mu_T) + \lambda$
>
> The bound depends on the Lipschitz constant $L$ of the model, as well as on the Wasserstein distance between the two data distributions $W_1(\mu_S, \mu_T)$. To compute $W_1(\mu_S, \mu_T)$ in our setting, we first need a distance between individual samples (i.e., multisets) from the source and target distributions. For $NN_{MEAN}$, the distance between two such multisets is computed using EMD, whereas for $NN_{MAX}$, it is computed using the Hausdorff distance. Once these pairwise sample distances are defined, we can compute the Wasserstein distance $W_1(\mu_S, \mu_T)$. The smaller this distance is (i.e., the closer the two distributions), the tighter the resulting generalization bound. Intuitively, if the multisets of the test set are "close" (in EMD or Hausdorff distance) to the training multisets, and the $NN_{MEAN}$ or $NN_{MAX}$ model is stable (Lipschitz continuous), then the test error can be bounded.
>
> > Line 459-461: I think when you write "insensitive" you actually mean "sensitive". In my understanding insesnsitive=robust.
>
> The reviewer is correct. Thank you for pointing this out. $NN_{MEAN}$ is more robust to Pert. #1, whereas $NN_{MAX}$ is more robust to Pert. #2. We will correct this sentence in the upcoming revision of the paper.
>
> [1] Chuang, C. Y., & Jegelka, S., "Tree mover's distance: Bridging graph metrics and stability of graph neural networks", In NeurIPS'22.\
> [2] Davidson, Y., & Dym, N., "On the Hölder Stability Of Multiset and Graph Neural Network", In ICLR'25.

---

> > ### Comment · Reviewer_AMSr · 2025-11-22
> >
> > I thank the reviewers for the detailed response. I would ask that they take these comments into consideration when revising the paper. I will maintain my positive score.

---

> > > ### Author Response · Authors · 2025-11-23
> > >
> > > We thank the reviewer once again for carefully reading our paper and for the very constructive feedback that will help improve its quality. All comments will be taken into account in the next version of the paper.

---

### Official Review · Reviewer_y3oZ · 2025-10-28

**Soundness:** 3
**Presentation:** 2
**Contribution:** 2
**Rating:** 4
**Confidence:** 2

**Summary:**

This paper systematically analyzes the Lipschitz continuity of three representative aggregation functions for multiset inputs (SUM, MEAN, MAX) with respect to three order-invariant multiset distances (EMD, Hausdorff, Matching), and provides exact constants. Based on these results, the authors derive a Lipschitz upper bound for set neural networks composed of “MLP → Aggregation → MLP”, and further discuss the applicability of existing W₁-based generalization bounds under distribution shift. Numerical experiments are conducted on ModelNet40 and Polarity, visualizing the practical validity of the derived bounds through scatter plots and correlation coefficients.

**Strengths:**

- Presents a clear overview of which combinations of {SUM/MEAN/MAX} × {EMD, Hausdorff, Matching} are Lipschitz continuous, and additionally provides strengthened results for fixed cardinalities as lemmas.
- Proposes a simple yet useful composition rule that allows direct derivation of Lipschitz upper bounds for set neural networks.
- Demonstrates the effectiveness of the analysis across both image processing and natural language domains.

**Weaknesses:**

- The main results focus on correlation plots, but lack comparisons of task performance (accuracy) and ablation studies (e.g., classification accuracy differences among SUM/MEAN/MAX).
- It is unclear what practical benefits this work brings to neural networks for set functions.
- The Matching distance is a metric only when “no zero vector is included” (Proposition 2.2), which may be inconsistent with real-world preprocessing (e.g., padding).

**Questions:**

- When Proposition 2.2’s assumption is violated in practice (e.g., zero vectors introduced by padding), how can the proposed theory be applied or extended?
- In what kinds of practical situations would it be appropriate to use Mean, Max, or Sum, respectively? For instance, depending on conditions such as (a) a few outliers vs. widespread small noise, (b) fixed vs. variable set cardinality, or (c) small-element sets (e.g., fashion) vs. large-element sets (e.g., point clouds, documents).
- This study explores suitable distance functions for each aggregation type (Sum, Mean, Max) under different scenarios. As a potential future direction, could this approach be further generalized by incorporating the Hölder mean (see arXiv:2403.17410)?

(Other Comments)
- Overall: It would be preferable to end mathematical expressions with a comma or period.
- Line 41: might loose → might lose
- Line 114: The table should include a title.
- Line 147: Hausdorff distance → Hausdorff Distance
- Line 356: Haudorff → Hausdorff
- Figures 1–2: The axes are too small to read. It would also help to indicate within the figure which columns correspond to sum, mean, and max.

---

> ### Author Response · Authors · 2025-11-20
> **Response to Reviewer y3oZ**
>
> We would like to thank the reviewer for their thoughtful evaluation and the constructive comments. We provide detailed responses to the reviewer's major comments below.
>
> > The main results focus on correlation plots, but lack comparisons of task performance (accuracy) and ablation studies (e.g., classification accuracy differences among SUM/MEAN/MAX).
>
> Following the reviewer's suggestion, we report in the following Table the classification accuracies and RMSEs achieved by $NN_{SUM}$, $NN_{MEAN}$ and $NN_{MAX}$ on the test sets of ModelNet40, Polarity, IMDB and IMDB-binary. The first two datasets are already discussed in the submitted manuscript. IMDB contains movie reviews from the IMDb database [1]. The targets are the ratings that accompany the reviews (10 different values). We treat this task as a regression problem. IMDB-binary is a standard graph classification dataset [2], commonly used for evaluating graph kernels and graph neural networks. Each graph of IMDB-binary was represented as a multiset of the degrees of its nodes. Each experiment was repeated 5 times with different random seeds and we report average accuracy/RMSE and the corresponding standard deviation.
>
> |             | ModelNet40 $(\uparrow)$  | Polarity $(\uparrow)$ | IMDB $(\downarrow)$ | IMDB-binary $(\uparrow)$  |
> |-------------|------------------|------------------|---------------------|------------------|
> | $NN_{SUM}$  | 60.07 $\pm$ 2.12 | 76.93 $\pm$ 2.42 | 0.2210 $\pm$ 0.0085 | 70.20 $\pm$ 1.72 |
> | $NN_{MEAN}$ | 63.41 $\pm$ 0.98 | 77.11 $\pm$ 2.34 | 0.2159 $\pm$ 0.0033 | 67.00 $\pm$ 2.52 |
> | $NN_{MAX}$  | 77.21 $\pm$ 1.09 | 78.14 $\pm$ 1.94 | 0.2259 $\pm$ 0.0017 | 61.40 $\pm$ 2.57 |
>
> Next, we discuss and interpret the obtained results.
>
> We can see that $NN_{MAX}$ outperforms the other models on ModelNet40. A possible explanation is that **all input multisets have the same size**, and in such a setting the max aggregator is Lipschitz continuous with respect to all three considered distance functions. Therefore, $NN_{MAX}$ can effectively capture the distances between the point clouds in ModelNet40.
>
> Polarity consists of short reviews (average number of terms = 20). Therefore, whether a review is positive or negative depends primarily on the **presence of one or a few terms** that indicate sentiment. These terms can be considered extreme elements, and the Hausdorff distance relies on such extreme elements when comparing its inputs. This distance function thus seems to be suitable for this task and potentially explains why $NN_{MAX}$ is the best-performing model on this dataset.
>
> The reviews contained in the IMDB dataset are much longer than those in the Polarity dataset (average number of terms = 254). Due to the potential presence of outlier terms, the Hausdorff distance may not accurately capture document similarity. The matching distance can also be sensitive to document length. In contrast, EMD captures the overall semantic alignment between documents, and compares documents based on their **overall meaning**. This makes EMD more suitable for this task, which is empirically confirmed by the superior performance of $NN_{MEAN}$ compared to the other two models.
>
> $NN_{SUM}$ is the best-performing method on the IMDB-binary dataset. In this dataset, capturing **both the number of nodes and their degrees** is essential. The matching distance is well suited for this task, and the stronger performance of the $NN_{SUM}$ model, which is Lipschitz continuous with respect to this distance (under certain conditions), supports this intuition.
>
> > It is unclear what practical benefits this work brings to neural networks for set functions.
>
> To the best of our knowledge, our work constitutes the **first systematic study** of the Lipschitz continuity of neural networks for sets under different distance functions for multisets. The work mainly provides insights into the stability, the learned multiset representations and the generalization performance of those models. For example, in a multiset classification problem, if a distance function fails to capture the variability between samples from different classes (i.e., samples from different classes are close under this distance), then a model that is Lipschitz continuous with respect to this distance does not guarantee good performance. In such a case, samples from different classes will receive similar representations, making them difficult to distinguish. Therefore, if a practitioner has domain knowledge, they can determine which distance function is better suited to their data/problem and choose the appropriate neural network architecture. In this way, our work can also **support informed architecture design**.

---

> > ### Author Response · Authors · 2025-11-20
> > **Response to Reviewer y3oZ**
> >
> > > The Matching distance is a metric only when “no zero vector is included” (Proposition 2.2), which may be inconsistent with real-world preprocessing (e.g., padding).
> >
> > In terms of implementation, we can use scatter operations and **avoid zero padding**. For instance, the [torch_scatter](https://pytorch-scatter.readthedocs.io/en/latest/functions/scatter.html) package provides such kind of operations that allow aggregating variable-sized multisets of elements. This is actually how we implemented our neural network models for input multisets of variable sizes. Note also that such an implementation is less memory-intensive and ensures that no zero vectors are appended to the input multisets.
> >
> > In addition to the point above, we note that, in order for the matching distance to be a metric, the multisets should not contain zero vectors **as actual elements**. Zero vectors can instead be used for padding without affecting the derived results. The presence of such vectors due to padding could pose problems for the other considered distance functions (e.g., the Hausdorff distance when all vectors in a multiset are negative along every dimension).
> >
> > > When Proposition 2.2’s assumption is violated in practice (e.g., zero vectors introduced by padding), how can the proposed theory be applied or extended?
> >
> > As discussed above, **zero padding is not required** because scatter operations can be used instead. Even if zero padding is applied, it does not violate the assumption of Proposition 2.2, since the padded zeros are not actual elements of the multisets.
> >
> > The assumption would only be violated if a multiset contained genuine elements equal to the zero vector. However, this situation is highly unlikely in real-world scenarios. The zero vector typically represents the **absence of information**. For instance, a word embedding of all zeros carries no semantic content. Thus, in practice, this condition is never violated.
> >
> > > In what kinds of practical situations would it be appropriate to use Mean, Max, or Sum, respectively? For instance, depending on conditions such as (a) a few outliers vs. widespread small noise, (b) fixed vs. variable set cardinality, or (c) small-element sets (e.g., fashion) vs. large-element sets (e.g., point clouds, documents).
> >
> > As a general guideline, one should choose the function that is Lipschitz continuous with respect to the distance function that **best captures** the distances between the multisets in the considered dataset or problem. For example, in problems where the shape of the input object matters (e.g., shapes extracted from medical images or 3D scans), Hausdorff distance is preferable to EMD and the matching distance since we would like to detect whether any part of one shape is far away from the other shape, even if the rest of the shapes are well-aligned. However, in some cases, **not a single** distance function is suitable for a single problem. For instance, consider the problem of text categorization, where documents are represented as multisets of word vectors. If two documents are considered similar when they contain similar terms, regardless of their length, the EMD is likely to best capture the distance between them. On the other hand, if similarity is determined by the presence of just one or a few extreme shared words, the Hausdorff distance is more appropriate. This illustrates that selecting an aggregation function typically requires some domain knowledge. In the absence of such knowledge, choosing an aggregation function can be challenging, except in special cases, such as when multisets have the same cardinality where our results indicate that the max function is Lipschitz continuous with respect to all distance functions.
> >
> > With regards to the conditions listed by the reviewer, case (a) corresponds exactly to the setting of the experiment presented in subsection 4.3. For datasets with one or few outliers, the mean and sum functions are more suitable than the max aggregator since it is likely that they are less sensitive to the outliers, especially when the cardinalities of the multisets are large. On the other hand, in case of widespread small noise, the max function is likely to be more robust. In case (b), when the multiset sizes are fixed, our theoretical results suggest that the max function is preferable to the other functions. This is also empirically verified on ModelNet40, as discussed above. For variable cardinalities, there is no clear answer, as it depends on the input data and the specific task at hand. In case (c), for large multisets, it is likely that EMD and the matching distance better capture the distances between samples than the Hausdorff distance since two multisets can have similar extreme values but there elements can be very different from each other. Therefore, in this scenario, the mean and sum functions may be more suitable choices.

---

> > > ### Author Response · Authors · 2025-11-20
> > > **Response to Reviewer y3oZ**
> > >
> > > > This study explores suitable distance functions for each aggregation type (Sum, Mean, Max) under different scenarios. As a potential future direction, could this approach be further generalized by incorporating the Hölder mean (see arXiv:2403.17410)?
> > >
> > > This is an interesting direction. The mean and max functions considered in our work can in fact be viewed as special cases of the Hölder mean for $p = 1$ and $p = +\infty$ [3], respectively. Our results indicate that the Hölder mean is not Lipschitz continuous with respect to a single distance function for any value of $p$. Therefore, each special case of the Hölder mean will likely require a separate, dedicated analysis. Thank you for bringing this aggregation function to our attention. This is indeed a meaningful direction for future work.
> > >
> > > [1] Maas, A., Daly, R. E., Pham, P. T., Huang, D., Ng, A. Y., & Potts, C., "Learning word vectors for sentiment analysis", In ACL'11.\
> > > [2] Yanardag, P., & Vishwanathan, S. V. N, "Deep graph kernels", In KDD'15.\
> > > [3] Kimura, M., Shimizu, R., Hirakawa, Y., Goto, R., & Saito, Y.,s "On permutation-invariant neural networks", arXiv:2403.17410, 2024.

---

> ### Comment · Reviewer_y3oZ · 2025-11-25
> **Thanks for the clarifications!**
>
> Thank you for addressing all points. Many of my concerns have been resolved, so I am increasing my score. Please incorporate these updates into the camera-ready version.

---

> ### Author Response · Authors · 2025-11-25
> **Response to Reviewer y3oZ**
>
> Thank you for your positive feedback and for taking the time to re-evaluate the submission. We are glad that many of your concerns have been resolved, and we appreciate you raising your score. We will incorporate all suggested updates and clarifications into the next version to ensure the final paper reflects these improvements.

---

### Official Review · Reviewer_gLAF · 2025-10-29

**Soundness:** 3
**Presentation:** 4
**Contribution:** 4
**Rating:** 8
**Confidence:** 4

**Summary:**

It has been demonstrated that some of the key properties of neural network models (such as their generalisation or robustness to corruption) can be controlled by (a bound on) their Lipschitz constant. This article addresses the question of whether specific neural networks, designed to process collections of vectors (sets or multisets) whose order is irrelevant to the task at hand, are Lipschitz continuous with respect to a given distance. The authors examine three known possible distances between unordered multisets, the Haussdorf distance, the Earth mover distance (EMD) and the matching distance, as well as three order-independent aggregation functions at the heart of common neural architectures for multisets, namely the sum, mean and max functions. The authors then thoroughly investigate all combinations of distances and aggregation functions, determine the possible Lipschitz continuity and, if continuous, the limits of the Lipschitz constants of these aggregation functions and the neural network constructed from them. Finally, the article analyses the theoretical limits of two proof-of-concept applications, namely the classification of a point cloud representing shapes (the ModelNet40 dataset) and the classification of film reviews from a set of unordered words (the Polarity dataset).

**Strengths:**

-   The authors study the key properties of neural networks for unordered multisets, a context that appears to have been little studied to date.
-   They provide a comprehensive study of the Lipschitz continuity of these networks and their aggregation functions with respect to three known distances (EMD, Haussdorf, matching distance), and provide new bounds on the Lipschitz constant when available. I'm not expert in this topic but the results seem novel.
-   Where available, they show theoretically and numerically how the bounds of the Lipschitz constant can characterise the robustness of these neural networks in the face of certain data corruptions (addition of elements, perturbation of elements).
-   Based on a result from (Shen et al., 2018), they also explain how these limits can be used to characterise the generalisation error of such a network in the event of a change in distribution.
-   They provide a convincing numerical analysis of aggregation functions and the neural networks that rely on them. This analysis is performed on two datasets, and a comparison between the theoretical Lipschitz upper bounds and the numerical upper bounds is provided.

**Weaknesses:**

Overall, the article is interesting and easy to read despite its technical nature and the diversity of all the aggregation function<->distance associations considered. I have listed the following errors that the authors should correct. True weaknesses are labeled as such in the list below.

-   (weakness) In Sec. 3.1, while the numerical treatment is clear and legit, I didn't get why the authors need 3 trained different neural networks to simply test the Lipschitz continuity of the aggregation function. This part should be better motivated. Is it a way to artificially generate a lot of multiset examples? Or to generate vector multiset of a fixed dimension? A priori, I don't see any connection between the way these vectors are generated and the Lipschitz continuity: these vectors could be generated by any means, the Lipschitz continuity of the aggregation function that is fed with them is independent, except if the continuity is restricted to a family of specific vector but this is nowhere considered in this work.
-   (weakness) L1102: "f1 and f2 are fully-connected layers" This is part is a bit obscure. First the f1 is applied to vectors in the definition of vX and then in this line, it is defined by its application to scalars. If it is applied componentwise to vectors it must be explained. But then, the affine transformation is a bit special and seems to involve a diagonal matrix diag(a1, ..., an). This part needs improvements.
-   (weakness/error) Proof of Prop 3.6: Section A.8, L1433-1424: The proposed matrix F does not respect the constraints in L1439 and L1442. The only matrix I see that could respect it is $F_{ij} = 1/(n\*(n+1))$ whatever the value of i and j; but maybe I'm wrong.
-   The 3 claims of Thm 3.1 do not use explicitly the multisets X and Y introduce at the beginning of the theorem statement; Same remark for Thm 3.3
-   Page 4, sentence "As discussed above, the SUM function is theoretically more powerful than the rest of the functions.": the word "powerful" is a bit vague in this sentence. What was introduced earlier is rather the possibility for the aggregation to be "injective"; this is not directly related to a concept of powerfulness or efficiency.
-   Page 5, line 239: "g" does not strictly operate on a multiset with the double-brace notation, it is rather that g is invariant under any permutation of its input and I guess the notation g({{..}}) rather means that invariance. Such a meaning could be inserted in Sec. 2.1 "Notation"
-   I guess this is a bit too implicit but it should be (re)said the Lipschitz constant over $f_{MLP1}$ and $f_{MLP2}$ are associated to the standard L2-L2 continuity of these functions
-   Page 6: while interesting, the analysis of the attention mechanism (and the negative result regarding its continuity) would deserve to be announced with a few words in the intro. As it, it looks like an extra element that comes out of the blue.
-   L297-298: "Incorporating l2 attention into the definition of $f_{ATT}(X), unfortunately, does not make it Lipschitz." -> Is it a fact or something that has been proved by the authors? Explain
-   Proposition 3.6: there is an error in the proof of point (1), see below
-   L356-357: "Haudorff"
-   L458-459: how the value 0.2 has been determined in the width of the uniform noise? That is a magical value.
-   L459-462: Regarding the sensitivity of NNmax vs NNmean in the different considered scenarios, could this sensitivity be backed by the previous theoretical analysis?
-   L617: "denote the permutation" -> "denote one of permutation"; there could be several minimizing permutations with the same score.
-   L770: "Then, we have" -> The fact that v1 = u1 ...; is not a consequence, but a choice to find a counterexample; adapt the sentence accordingly, here and in the other 5 other similar proofs later.
-   L835: define $d_{ij}$ (I guess it's just $\|v_i - u_j\|$)
-   L835: the inequality is in the wrong direction; the sum with the entries $F_{ij}$ is bigger than the norm of the difference of means; same error later in another proof, L926
-   L1135: You should remove this line or place a minimum over all possible matrix F here since the next bound using F\* is then bigger if F\* is the matrix minimizing the EMD rather than this minimization
-   L1458-1460: The statement and equation "We thus obtain dH(W,X') = h(X',X) = ..." is wrong, but it seems anyway unnecessary for the proof of point (2).

Minor/cosmetic:
-   if not already the case, everywhere avoid denoting norms with the double bar, use rather "backslash bars" in latex

**Questions:**

-   Can you provide a correction to the proof of Prop. 3.6?
-   Can you better explain the need of neural networks to generate multisets in Sec. 3.1?

---

> ### Author Response · Authors · 2025-11-20
> **Response to Reviewer gLAF**
>
> We sincerely thank the reviewer for the positive evaluation and the constructive comments. We also thank the reviewer for carefully reading the manuscript and for highlighting the typos and notation issues. Below, we address the major comments raised by the reviewer.
>
> > (weakness) In Sec. 3.1, while the numerical treatment is clear and legit, I didn't get why the authors need 3 trained different neural networks to simply test the Lipschitz continuity of the aggregation function...
>
> We believe the reviewer is referring to subsection 4.1 and not subsection 3.1. We agree with the reviewer that the multisets used to verify the Lipschitz constants of the aggregation functions could, in principle, be generated by any means. However, our goal is to investigate how these functions behave in comparison to the derived bounds when the inputs are sampled from **real distributions**, rather than artificially generated data that do not occur in practice. For this reason, in our experiments we use neural networks to generate multiset examples, and the inputs to these networks are real samples, such as point clouds and movie reviews. We would be happy to provide additional clarification if needed.
>
> > (weakness) L1102: "f1 and f2 are fully-connected layers" This is part is a bit obscure. First the f1 is applied to vectors in the definition of vX and then in this line, it is defined by its application to scalars...
>
> In the general definition of $NN_{SUM}$, $f_1$ and $f_2$ are affine functions of the form $f_1 \colon \mathbf{v} \mapsto \mathbf{W}_1 \mathbf{v} + \mathbf{b}_1$ and $f_2 \colon \mathbf{v} \mapsto \mathbf{W}_2 \mathbf{v} + \mathbf{b}_2$. However, multisets $X$ and $Y$ defined in the proof consist of scalars. Therefore, we provide in L1102 the explicit forms of these functions when their inputs are scalars. We apologize for the misunderstanding. We will clarify this in the revised manuscript.
>
> > (weakness/error) Proof of Prop 3.6: Section A.8, L1433-1424: The proposed matrix F does not respect the constraints in L1439 and L1442. The only matrix I see that could respect it is $F_{ij} = 1/(n*(n+1))$ whatever the value of i and j; but maybe I'm wrong
>
> The reviewer is correct. This was an oversight on our part. Matrix $\mathbf{F}$ needs to be constructed as follows:
> - if $i=j$, then $[\mathbf{F}]_{ij} = \frac{1}{n+1}$
> - if $j=n+1$, then $[\mathbf{F}]_{ij} = \frac{1}{n(n+1)}$
> - if $i \neq j$ and $j \neq n+1$, then $[\mathbf{F}]_{ij} = 0$
>
> This matrix is thus obtained by horizontally concatenating a diagonal matrix with entries $\frac{1}{n+1}$ on the main diagonal and a column vector whose entries are all $\frac{1}{n(n+1)}$, and is a feasible solution of the EMD formulation. Then, the equality of L1446 holds. We thank the reviewer for noticing this issue.
>
> > L297-298: "Incorporating l2 attention into the definition of $f_{ATT}(X), unfortunately, does not make it Lipschitz." -> Is it a fact or something that has been proved by the authors? Explain
>
> This is a fair point. To our knowledge, this result has not been reported in prior work. In fact, we have proven it ourselves, but did not include it in the original submission because this type of attention is not commonly used in practice. We will provide a formal proof in the revised manuscript (to be uploaded by December 3).
>
> > L458-459: how the value 0.2 has been determined in the width of the uniform noise? That is a magical value.
>
> This value was determined as follows. We first extracted the embeddings of all words present in the Polarity dataset from the pre-trained GoogleNews word2vec model and computed the mean and standard deviation across the 300 dimensions. We found that the means of the 300 dimensions are very close to zero (mean of means = −0.0013), while the standard deviations mostly range between 0.15 and 0.18 (mean of standard deviations = 0.1651). Since our objective was to perturb the word vectors without altering them too strongly, we chose to add noise sampled from $\mathcal{U}(0,0.2)^{300}$ to each vector.

---

> > ### Author Response · Authors · 2025-11-20
> > **Response to Reviewer gLAF**
> >
> > > L459-462: Regarding the sensitivity of NNmax vs NNmean in the different considered scenarios, could this sensitivity be backed by the previous theoretical analysis?
> >
> > In the case of Pert. #1, a single element is added to each multiset in the test set. This element acts as an **outlier**, so the Hausdorff distance between the perturbed multisets and the original multisets of the same class is likely to be large. In contrast, the EMD between the perturbed multisets and the original multisets of the same class is likely to remain small, since EMD considers all elements and the multisets in ModelNet40 have large cardinalities. Therefore, this perturbation has a greater impact on the Hausdorff distance than on EMD, which explains why $NN_{MEAN}$ (Lipschitz continuous with respect to EMD), is more robust to this perturbation than $NN_{MAX}$ (Lipschitz continuous with respect to the Hausdorff distance).
> >
> > Pert. #2 applies a **small perturbation to each element** of the test multisets. Because the perturbations are small, the Hausdorff distance between the perturbed multisets and the original multisets of the same class is likely to be small, whereas the EMD may increase more noticeably since the perturbation is applied to all elements. Thus, this perturbation affects EMD more than Hausdorff distance, which explains why $NN_{MAX}$ is more robust to this perturbation than $NN_{MEAN}$.

---

### Official Review · Reviewer_ZjJJ · 2025-10-30

**Soundness:** 4
**Presentation:** 3
**Contribution:** 3
**Rating:** 8
**Confidence:** 3

**Summary:**

The paper analyses the Lipschitz continuity of the set aggregation functions MEAN, SUM, and MAX. The EMD, Hausdorff distance, and matching distance are analysed, and Lipschitz bounds are derived for the cases of multisets having the same and different cardinalities. These results are then extended to the Lipschitz continuity of a neural network based on MLPs and an aggregation function. The authors show that a modified attention mechanism is not Lipschitz-bounded and use their results to derive bounds on stability under perturbations and distribution shifts. Numerical experiments are provided for two tasks and linked to the theory.

**Strengths:**

The authors provide a significant contribution to the analysis of Lipschitz continuity of aggregation functions. The paper is well structured, and the results build on each other, from the fundamental results in Table 1 to the bounds on input perturbations. A small set of benchmarks nicely accompanies the theoretical results.

**Weaknesses:**

The paper would benefit from explaining related literature and the connections to it better. Based on the theoretical and experimental results, the conclusions for practitioners should be better spelled out (details see below).

**Questions:**

- Most theorems exclude sets that contain the zero vector. Explain why, whether it could be mitigated, and its effect on practice.
- Please better explain the usage of the analysed network structures for state of the art (SOTA) or practice in the context of transformers or GNNs (not just one sentence in line 63 and two citations from 2017).
- Please relate your work more to the cited Fourier Sliced-Wasserstein embedding (line 70) and the importance of bi-Lipschitz continuity and injectivity of aggregation functions.
- Please describe the theoretical advantages of the sum aggregator better (line 119).
- Why were those three distance metrics chosen?
- The reader would benefit from a summary of when each metric is relevant and which aggregation function should be chosen in practice. Please relate it to the initial motivation of robustness and include experimental results, such as loose bounds.
- Please relate the attention aggregation to the mean or the max.
- Please unify notation, e.g., activation functions in line 234 and line 284.

---

> ### Author Response · Authors · 2025-11-20
> **Response to Reviewer ZjJJ**
>
> We thank the reviewer for the positive evaluation and the thoughtful comments. We provide responses to the comments and questions below.
>
> > The paper would benefit from explaining related literature and the connections to it better.
>
> This is a valid point. Due to space constraints, we were unable to include a related work section in the main body of the submitted manuscript. In the revised version (to be uploaded by December 3), we will add a related work section in the Appendix discussing the connections between our contributions and the existing literature.
>
> > Most theorems exclude sets that contain the zero vector. Explain why, whether it could be mitigated, and its effect on practice.
>
> This concern only affects results involving the matching distance, since this function is not necessarily a metric when input multisets contain elements equal to the zero vector. We would like to emphasize that it is highly unlikely that multisets constructed from real-world data contain the zero vector as one of their elements. This vector typically corresponds to an **absence of information**. For example, a word embedding consisting entirely of zeros carries no semantic content.
>
> Note that certain multiset elements may be mapped to the zero vector after they pass through neural network layers with ReLU activations applied to their outputs. However, this becomes unlikely for sufficiently **large hidden dimension sizes**, and it can also be avoided by using **non-vanishing activations** such as LeakyReLU.
>
> Therefore, in practice it is very unlikely that input multisets or multisets whose elements have been updated by neural network layers contain any zero vectors.
>
> > Please better explain the usage of the analysed network structures for state of the art (SOTA) or practice in the context of transformers or GNNs (not just one sentence in line 63 and two citations from 2017).
>
> Architectures similar to those studied in this paper have been applied across diverse domains, including biology [1], chemistry [2] and materials science [3], while they have also been further extended in subsequent works [4,5,6]. The aggregation functions considered here are among the most widely used in the literature of graph neural networks [7,8], and also appear in Transformer architectures [9]. In addition, several works represent complex objects as multisets of vectors and employ models similar to those studied in this paper, often incorporating domain knowledge. For example, [10] introduces neural architectures specifically designed for eigenvector-based inputs, which can be viewed as variants of the models studied here, while explicitly accounting for the symmetries inherent in eigenvectors. We will provide more details in the revised manuscript.
>
> > Please relate your work more to the cited Fourier Sliced-Wasserstein embedding (line 70) and the importance of bi-Lipschitz continuity and injectivity of aggregation functions.
>
> The Fourier Sliced-Wasserstein embedding **does not rely on sum, mean, or max aggregators** to construct multiset representations. Instead, it is based on sort embeddings, where information about the distribution is captured using the project-and-sort operation. Therefore, the Fourier Sliced-Wasserstein embedding follows a fundamentally different approach to multiset representation and is not directly comparable to the aggregation functions studied in our paper.
>
> Bi-Lipschitz continuity is often a desirable property for multiset embeddings, as it ensures stability, robustness, and improved generalization. While the three aggregation functions we consider are not bi-Lipschitz, sort-based embeddings, such as those used in the Fourier Sliced-Wasserstein embedding, give rise to bi-Lipschitz multiset representations by leveraging the geometric properties of sorting random projections.
>
> > Please describe the theoretical advantages of the sum aggregator better (line 119).
>
> The sum aggregator can represent a strictly larger class of functions over multisets than the mean and max aggregators. It was shown in [11] that if the elements of the input sets come from a countable set $X$, then for an appropriate $g \colon X \rightarrow \mathbb{R}$, the function defined as $g(${  $v_1, \ldots, v_n$ } $) = \sum_{i=1}^n f(v_i)$ maps the input sets injectively to $\mathbb{R}$. Notably, it is also shown that injectivity is sufficient for approximation. On the other hand, the mean and max functions are not injective set functions. These results were generalized to multisets in [7]. Due to space constraints, we could not include all relevant details in the current version. We will provide more details in the revised manuscript.

---

> > ### Author Response · Authors · 2025-11-20
> > **Response to Reviewer ZjJJ**
> >
> > > Why were those three distance metrics chosen?
> >
> > EMD and Hausdorff distance are both **standard distance functions** for sets and multisets and have been widely used in different problem settings in the past. Since the sum function is not Lipschitz continuous with respect to either of these distances, we designed the matching distance to serve this purpose. Reviewer AMSr pointed out in their review that this distance has been adopted in prior work, which provides additional justification for the soundness of using this function. Importantly, the three considered distance functions are sensitive to **different types of variability** present in the input data.
> >
> > > The reader would benefit from a summary of when each metric is relevant and which aggregation function should be chosen in practice. Please relate it to the initial motivation of robustness and include experimental results, such as loose bounds.
> >
> > The selection of an aggregation function should be guided by the type of variation one intends to model in the input multisets. Different distance functions capture **different types of variation** in the input multisets. A distance measure can be more relevant in a given setting than the rest of the measures, but determining the most appropriate one usually requires domain knowledge. For instance, in applications where the geometric shape of the input is critical (such as shapes derived from medical imaging or 3D scans), the Hausdorff distance is often more appropriate than EMD and the matching distance, as it is sensitive to outlier points and can detect whether any part of one shape deviates significantly from another, even when the remainder is well-aligned. Conversely, for tasks involving text embeddings, EMD is generally preferable to the rest of the distance functions, as it effectively captures the overall semantic alignment between sets of word vectors. These insights can **inform empirical design choices** in aggregation layers. Specifically, choosing a model that is Lipschitz continuous with respect to a distance function that aligns well with the task at hand can lead to improved robustness, and practitioners should consider these aspects when developing models for their applications.
> >
> > > Please relate the attention aggregation to the mean or the max.
> >
> > The attention aggregation boils down to the mean aggregation when the attention coefficients are uniformly distributed across all multiset's elements. It also approximates the max function when each multiset includes a single element achieving the maximum value in every vector dimension and this element is assigned an attention coefficient close to 1. However, as far as we can tell, no connection can be derived between the attention aggregation and the rest of the aggregation functions if no assumptions are made about the input multisets and the trainable vector $\mathbf{q}$.
> >
> > [1] Clarke, B., et al., "Integration of variant annotations using deep set networks boosts rare variant association testing", Nature Genetics, 56(10), 2024.\
> > [2] Boulougouri, M., Vandergheynst, P., & Probst, D., "Molecular set representation learning", Nature Machine Intelligence, 6(7), 2024.\
> > [3] Zhang, J., Cai, C., Kim, G., Wang, Y., & Chen, W., "Composition design of high-entropy alloys with deep sets learning", npj Computational Materials, 8(1), 2022.\
> > [4] Fei, J., Zhu, Z., Liu, W., Deng, Z., Li, M., Deng, H., & Zhang, S., "DuMLP-Pin: A Dual-MLP-dot-product Permutation-invariant Network", In AAAI'22.\
> > [5] Ou, Z., Xu, T., Su, Q., Li, Y., Zhao, P., & Bian, Y., "Learning neural set functions under the optimal subset oracle", In NeurIPS'22.\
> > [6] Ma, X., Qin, C., You, H., Ran, H., & Fu, Y., "Rethinking Network Design and Local Geometry in Point Cloud: A Simple Residual MLP Framework", In ICLR'22.\
> > [7] Xu, K., Hu, W., Leskovec, J., & Jegelka, S., "How powerful are graph neural networks?", In ICLR'19.\
> > [8] Morris, C., Ritzert, M., Fey, M., Hamilton, W. L., Lenssen, J. E., Rattan, G., & Grohe, M., "Weisfeiler and leman go neural: Higher-order graph neural networks", In AAAI'19.\
> > [9] Wu, X., Lao, Y., Jiang, L., Liu, X., & Zhao, H., "Point Transformer V2: Grouped Vector Attention and Partition-based Pooling", In NeurIPS'22.\
> > [10] Lim, D., Robinson, J. D., Zhao, L., Smidt, T., Sra, S., Maron, H., & Jegelka, S., "Sign and Basis Invariant Networks for Spectral Graph Representation Learning", In ICLR'23.\
> > [11] Zaheer, M., Kottur, S., Ravanbakhsh, S., Poczos, B., Salakhutdinov, R. R., & Smola, A. J., "Deep Sets", In NIPS'17.

---

### Author Response · Authors · 2025-12-03
**Note to AC - Summary of Changes in Revised Manuscript**

Dear AC,

We appreciate your time and effort in managing the review process for our submission.

We have revised the manuscript to address most of the reviewers' comments. Below, we summarize the main points that have been addressed in the revision:

- We more clearly described the theoretical advantages of the sum aggregator (Subsection 2.3).

- We added references to support the choice of distances functions on multisets and mentioned that EMD and the Hausdorff distance are pseudometrics on multisets of vectors (Subsection 2.4).

- We mentioned that a direct consequence of the Theorem 3.1 is that no two of the considered distance functions are bi-Lipschitz equivalent (Subsection 3.1).

- We extended Lemma 3.2 to include the result that the sum and mean aggregators are not Lipschitz continuous with respect to the Hausdorff distance (Subsection 3.1).

- We explained that $NN_{SUM}$ is not necessarily Lipschitz continuous with respect to the matching distance because of the bias terms of the MLP that updates the set representations (Subsection 3.2).

- We justified the use of neural networks for generating multiset data instead of generating synthetic multisets (Subsection 4.1).

- We explained why the $\mathcal{U}(0,0.2)^{300}$ distribution was used to inject noise (Subsection 4.3).

- We added a section discussing related work (Appendix A).

- We demonstrated that the $\ell_2$-based attention mechanism is not Lipschitz continuous with respect to any of the three considered distance functions (Appendix B.6).

- We fixed the proof of Proposition 3.6 (Appendix B.9).

- We evaluated the $NN_{SUM}$, $NN_{MEAN}$ and $NN_{MAX}$ models on four datasets from different domains and provided interpretations of the obtained results (Appendix E.5).

- We discussed how the theoretical results can inform architectural choices and we provided practical recommendations for choosing the most appropriate aggregation function for a given problem (Appendix E.6)

- We fixed all typos and addressed minor issues.

Note that in case of acceptance, we will utilize the additional page provided in the camera-ready version to discuss how the theoretical results can inform architectural decisions and which aggregation function should be chosen in practical settings. We will also further improve the paper's clarity and flow.

We greatly appreciate the AC's careful consideration of our submission, and we hope that our responses and the revised manuscript provide a clear explanation of the contributions and importance of our work. As a final remark, we note that the two reviewers who responded to our rebuttal confirmed that their concerns were addressed and that they were satisfied with our response, with one reviewer also raising their rating.

@Reviewers: We would like to thank the reviewers once again for carefully reading our manuscript and for providing insightful, high-quality reviews.

---

### Meta-Review · Area_Chair_18km · 2026-01-14

**Summary:**

All reviewers find the papers setting and contributions meaningful. Main concerns were around limitation of the results to theoretical setting and lack of real world impact and limited experiments. Authors provided some promising directions towards addressing these. Overall I think the paper is in a good shape and recommend acceptance.

**Reviewer Concerns:**

Main concerns were around limitation of the results to theoretical setting and lack of real world impact and limited experiments.

**Reviewer Scores:**

y3oZ 4 -> 6

---

### Decision · Program_Chairs · 2026-01-26

Accept (Poster)